# Distributional Reduction: Unifying Dimensionality Reduction and Clustering with Gromov-Wasserstein

**Hugues Van Assel**  *hugues.van__assel@ens-lyon.fr*
*ENS de Lyon, CNRS, UMPA UMR 5669*

**Cédric Vincent-Cuaz**  *cedric.vincent-cuaz@epfl.ch*
*EPFL, LTS4*

**Nicolas Courty**  *courty@univ-ubs.fr*
*Université Bretagne Sud, IRISA UMR 6074*

**Rémi Flamary**  *remi.flamary@polytechnique.edu*
*École polytechnique, Institut Polytechnique de Paris, CMAP UMR 7641*

**Pascal Frossard**  *pascal.frossard@epfl.ch*
*EPFL, LTS4*

**Titouan Vayer**  *titouan.vayer@inria.fr*
*Inria, ENS de Lyon, CNRS, Université Claude Bernard Lyon 1, LIP UMR 5668*

**Reviewed on OpenReview:** *https://openreview.net/forum?id=cllm6SS354*

## Abstract

Unsupervised learning aims to capture the underlying structure of potentially large and high-dimensional datasets. Traditionally, this involves using dimensionality reduction (DR) methods to project data onto lower-dimensional spaces or organizing points into meaningful clusters (clustering). In this work, we revisit these approaches under the lens of optimal transport and exhibit relationships with the Gromov-Wasserstein problem. This unveils a new general framework, called distributional reduction, that recovers DR and clustering as special cases and allows addressing them jointly within a single optimization problem. We empirically demonstrate its relevance to the identification of low-dimensional prototypes representing data at different scales, across multiple image and genomic datasets.

## 1 Introduction

One major objective of unsupervised learning (Hastie et al., 2009) is to provide interpretable and meaningful approximate representations of the data that best preserve its structure *i.e.* the underlying geometric relationships between the data samples. Similar in essence to Occam's principle frequently employed in supervised learning, the preference for unsupervised data representation often aligns with the pursuit of simplicity, interpretability or visualizability in the associated model. These aspects are determinant in many real-world applications where the interaction with domain experts is paramount for interpreting the results and extracting meaningful insights from the model. For instance, the design of tissue atlases or the inference of single-cell trajectories, which are essential for addressing various biological challenges (Rao et al., 2021; Saelens et al., 2019), depends on the analysis of metacells, i.e. granular and interpretable representations of cells that enable separating sampling effect from biological variance (Baran et al., 2019).

**Dimensionality reduction and clustering.** When faced with the question of extracting interpretable representations, from a dataset $\mathbf{X} = (\mathbf{x}_1, ..., \mathbf{x}_N)^\top \in \mathbb{R}^{N \times p}$ of $N$ samples in $\mathbb{R}^p$, the machine learning community has proposed a variety of methods. Among them, dimensionality reduction (DR) algorithms

Figure 1: **Illustration of our DistR method** on two toy examples. For each example (left and right), the data pairs $\mathbf{X}$ and $\mathbf{Z}$ represent the original data and their embedding as found by our method DistR. These pairs are associated to similarity matrices $\mathbf{C}_X(\mathbf{X}) \in \mathbb{R}^{N \times N}$ and $\mathbf{C}_Z(\mathbf{Z}) \in \mathbb{R}^{n \times n}$ ($n < N$), which encode the pairwise similarities in their respective spaces. In the left example, we consider a Gaussian similarity for $\mathbf{Z}$ and Symmetric Entropic Affinity (SEA) (Van Assel et al., 2023) for $\mathbf{X}$. In the right example, we consider SEA for $\mathbf{X}$ and a similarity in the Lorentz model for $\mathbf{Z}$ (Nickel and Kiela, 2018). DistR also provides a coupling $\mathbf{T}$ between the points in $\mathbf{X}$ and $\mathbf{Z}$, illustrated in purple, with its marginals.

have been widely used to summarize data in a low-dimensional space $\mathbf{Z} = (\mathbf{z}_1, ..., \mathbf{z}_N)^\top \in \mathbb{R}^{N \times d}$ with $d \ll p$, allowing for visualization of every individual points for small enough $d$ (Lee et al., 2007; Van Der Maaten et al., 2009). Another major approach is to cluster the data into $n$ groups, with $n$ typically much smaller than $N$, and to summarize these groups through their centroids (Bishop, 2006; Von Luxburg, 2007). Clustering is particularly interpretable since it provides a smaller number of representative points that can be more easily inspected. The cluster assignments can also be analyzed. Both DR and clustering follow a similar philosophy of summarization and reduction of the dataset using a smaller size representation.

**Two sides of the same coin.** As a matter of fact, methods from both families share many similitudes, including the construction of a similarity graph between input samples. In clustering, many popular approaches design a reduced or coarsened version of the initial similarity graph while preserving some of its spectral properties (Von Luxburg, 2007; Schaeffer, 2007). In DR, the goal is to solve the inverse problem of finding low-dimensional embeddings that generate a similarity graph close to the one computed from input data points (Ham et al., 2004; Hinton and Roweis, 2002). Our work builds on these converging viewpoints and addresses the following question: *can DR and clustering be expressed in a common and unified framework ?*

**A distributional perspective.** To answer this question, we propose to look at both problems from a distributional point of view, treating the data as an empirical probability distribution $\mu = \frac{1}{N} \sum_i \delta_{\mathbf{x}_i}$. This enables to consider statistical measures of similarity such as Optimal Transport (OT), which is at the core of our work. On the one hand, OT and clustering are strongly related. The celebrated k-means algorithm can be seen as a particular case of minimal Wasserstein estimator where a distribution of $n$ Diracs is optimized *w.r.t* their weights and positions (Canas and Rosasco, 2012). Other connections between spectral clustering and the OT-based Gromov-Wasserstein (GW) distance have been recently developed in Chowdhury and Needham (2021); Chen et al. (2023); Vincent-Cuaz et al. (2022a). On the other hand, the link between DR and OT is less explored. DR methods, when modeling data as distributions, usually focus on joint distribution between samples within each space separately, see *e.g.* Van Assel et al. (2023) or Lu et al. (2019). Consequently, they do not consider couplings to transport samples across spaces of varying dimensions. At the time of this paper's submission, we note that other authors have independently developed a similar line of work, as detailed in Clark et al. (2024); Murray and Pickarski (2024). Their studies focus on "multidimensional scaling" problems and establish connections to the semi-relaxed GW framework but do not explore alternative DR approaches or clustering aspects.

**Contributions.** In this paper, we develop a general distributional framework that encompasses both DR and clustering as special cases. We notably cast those problems as finding a reduced distribution that minimizes the GW divergence from the original empirical data distribution. Our method proceeds by first constructing an input similarity matrix $\mathbf{C}_X(\mathbf{X})$ that is matched with the embedding similarity $\mathbf{C}_Z(\mathbf{Z})$ through an OT coupling matrix $\mathbf{T}$. The latter establishes correspondences between input and embedding samples. We illustrate this principle in Figure 1 where one can notice that $\mathbf{C}_Z(\mathbf{Z})$ preserves the topology of $\mathbf{C}_X(\mathbf{X})$ with a reduced number of nodes. The adaptivity of our model that can select an effective number of cluster $< n$,

is visible in the bottom plot, where only the exact number of clusters in the original data (9 out of the 12 initially proposed) is automatically recovered. Our method can operate in any embedding space, which is illustrated by projecting in either a 2D Euclidean plane or a Poincaré ball.

**Outline.** We show that this framework is versatile and allows recovering as special cases many popular DR methods such as the kernel PCA and neighbor embedding algorithms, but also clustering algorithms such as weighted k-means and its kernel counterpart including spectral clustering (Chan et al., 2004; Dhillon et al., 2007). We first prove in Section 3 that DR can be formulated as a GW projection problem under some conditions on the loss and similarity functions. We then propose in Section 4 a novel formulation of data summarization as a minimal GW estimator that allows selecting both the dimensionality of the embedding $d$ (DR) and the cardinality of the support $n$ (Clustering). Finally, we show in section 5 the practical interest of our approach on images and genomics datasets.

**Notations.** The $i^{th}$ entry of a vector $\mathbf{v}$ is denoted as either $v_i$ or $[\mathbf{v}]_i$. Similarly, for a matrix $\mathbf{M}$, $M_{ij}$ and $[\mathbf{M}]_{ij}$ both denote its entry $(i, j)$. $S_N$ is the set of permutations of $[\![N]\!]$. $P_N(\mathbb{R}^d)$ refers to the set of discrete probability measures composed of N points of $\mathbb{R}^d$. $\Sigma_N$ stands for the probability simplex of size $N$ that is $\Sigma_N := \{\mathbf{h} \in \mathbb{R}_+^N \text{ s.t. } \sum_i h_i = 1\}$. $\log(\mathbf{M})$ and $\exp(\mathbf{M})$ are to be understood element-wise. For $\mathbf{x} \in \mathbb{R}^N$, $\mathrm{diag}(\mathbf{x})$ denotes the diagonal matrix whose elements are the $x_i$.

## 2 Background on Dimensionality Reduction and Optimal Transport

We first review the most popular DR approaches and introduce the Gromov-Wasserstein problem.

### 2.1 Unified View of Dimensionality Reduction

Let $\mathbf{X} = (\mathbf{x}_1, ..., \mathbf{x}_N)^\top \in \mathbb{R}^{N \times p}$ be an input dataset. Dimensionality reduction focuses on constructing a low-dimensional representation or *embedding* $\mathbf{Z} = (\mathbf{z}_1, ..., \mathbf{z}_N)^\top \in \mathbb{R}^{N \times d}$, where $d < p$. The latter should preserve a prescribed geometry for the dataset encoded via a symmetric pairwise similarity matrix $\mathbf{C_X} \in \mathbb{R}_+^{N \times N}$ obtained from $\mathbf{X}$. To this end, most popular DR methods optimize $\mathbf{Z}$ such that a certain pairwise similarity matrix in the output space matches $\mathbf{C_X}$ according to some criteria. We subsequently introduce the functions

$$\mathbf{C}_X : \mathbb{R}^{N \times p} \to \mathbb{R}^{N \times N}, \mathbf{C}_Z : \mathbb{R}^{N \times d} \to \mathbb{R}^{N \times N}, \tag{1}$$

which define pairwise similarity matrices in the input and output space, from the dataset $\mathbf{X}$ and the embedding $\mathbf{Z}$ respectively. The DR problem can be formulated quite generally as the optimization problem

$$\min_{\mathbf{Z} \in \mathbb{R}^{N \times d}} \sum_{(i,j) \in [\![N]\!]^2} L\big([\mathbf{C}_X(\mathbf{X})]_{ij}, [\mathbf{C}_Z(\mathbf{Z})]_{ij}\big). \tag{DR}$$

where $L : \mathbb{R} \times \mathbb{R} \to \mathbb{R}$ is a loss that quantifies the discrepancy between similarities between points in the input space $\mathbb{R}^p$ and in the output space $\mathbb{R}^d$. Various losses are used, such as the quadratic loss $L_2(x, y) := (x - y)^2$ or the Kullback-Leibler divergence $L_{KL}(x, y) := x \log(x/y) - x + y$. Below, we recall several popular methods that can be placed within this framework.

**Spectral methods.** When $\mathbf{C}_X(\mathbf{X})$ is a positive semi-definite matrix, eq. (DR) recovers spectral methods by choosing the quadratic loss $L = L_2$ and $\mathbf{C}_Z(\mathbf{Z}) = (\langle \mathbf{z}_i, \mathbf{z}_j \rangle)_{(i,j) \in [\![N]\!]^2}$ the matrix of inner products in the embedding space. Indeed, in this case, the objective value of eq. (DR) reduces to

$$\sum_{(i,j) \in [\![N]\!]^2} L_2([\mathbf{C}_X(\mathbf{X})]_{ij}, \langle \mathbf{z}_i, \mathbf{z}_j \rangle) = \|\mathbf{C}_X(\mathbf{X}) - \mathbf{Z}\mathbf{Z}^\top\|_F^2$$

where $\|\cdot\|_F$ is the Frobenius norm. This problem is commonly known as kernel Principal Component Analysis (PCA) (Schölkopf et al., 1997) and an optimal solution is given by $\mathbf{Z}^\star = (\sqrt{\lambda_1}\mathbf{v}_1, ..., \sqrt{\lambda_d}\mathbf{v}_d)^\top$ where $\lambda_i$ is the $i$-th largest eigenvalue of $\mathbf{C}_X(\mathbf{X})$ with corresponding eigenvector $\mathbf{v}_i$ (Eckart and Young, 1936). As shown by Ham et al. (2004); Ghojogh et al. (2021), numerous dimension reduction methods can be categorized in this manner. This includes PCA when $\mathbf{C}_X(\mathbf{X}) = \mathbf{X}\mathbf{X}^\top$ is the matrix of inner products in the input

space; (classical) multidimensional scaling (Borg and Groenen, 2005), when $\mathbf{C}_X(\mathbf{X}) = -\frac{1}{2}\mathbf{H}\mathbf{D}_\mathbf{X}\mathbf{H}$ with $\mathbf{D}_\mathbf{X}$ the matrix of squared euclidean distance between the points in $\mathbb{R}^p$ and $\mathbf{H} = \mathbf{I}_N - \frac{1}{N}\mathbf{1}_N\mathbf{1}_N^\top$ is the centering matrix; Isomap (Tenenbaum et al., 2000), with $\mathbf{C}_X(\mathbf{X}) = -\frac{1}{2}\mathbf{H}\mathbf{D}_\mathbf{X}^{(g)}\mathbf{H}$ with $\mathbf{D}_\mathbf{X}^{(g)}$ the geodesic distance matrix; Laplacian Eigenmap (Belkin and Niyogi, 2003), with $\mathbf{C}_X(\mathbf{X}) = \mathbf{L}_\mathbf{X}^\dagger$ the pseudo-inverse of the Laplacian associated to some adjacency matrix $\mathbf{W}_\mathbf{X}$; but also Locally Linear Embedding (Roweis and Saul, 2000), and Diffusion Map (Coifman and Lafon, 2006) (for all of these examples we refer to Ghojogh et al. 2021, Table 1).

**Neighbor embedding methods.** An alternative group of methods relies on neighbor embedding techniques which consists in minimizing in $\mathbf{Z}$ the quantity

$$\sum_{(i,j)\in\llbracket N\rrbracket^2} L_{\mathrm{KL}}([\mathbf{C}_X(\mathbf{X})]_{ij}, [\mathbf{C}_Z(\mathbf{Z})]_{ij}) \,. \tag{NE}$$

Within our framework, this corresponds to eq. (DR) with $L = L_{\mathrm{KL}}$. The objective function of popular methods such as stochastic neighbor embedding (SNE) (Hinton and Roweis, 2002) or t-SNE (Van der Maaten and Hinton, 2008) can be derived from eq. (NE) with a particular choice of $\mathbf{C}_X, \mathbf{C}_Z$. For instance SNE and t-SNE both consider in the input space a symmetrized version of the entropic affinity (Vladymyrov and Carreira-Perpinan, 2013; Van Assel et al., 2023). In the embedding space, $\mathbf{C}_Z(\mathbf{Z})$ is usually constructed from a "kernel" matrix $\mathbf{K}_\mathbf{Z}$ which undergoes a scalar (Van der Maaten and Hinton, 2008), row-stochastic (Hinton and Roweis, 2002) or doubly stochastic (Lu et al., 2019; Van Assel et al., 2023) normalization. Gaussian kernel $[\mathbf{K}_\mathbf{Z}]_{ij} = \exp(-\|\mathbf{z}_i - \mathbf{z}_j\|_2^2)$, or heavy-tailed Student-t kernel $[\mathbf{K}_\mathbf{Z}]_{ij} = (1 + \|\mathbf{z}_i - \mathbf{z}_j\|_2^2)^{-1}$, are typical choices (Van der Maaten and Hinton, 2008). We also emphasize that one can retrieve the UMAP objective (McInnes et al., 2018) from eq. (DR) using the binary cross-entropy loss $L_{BCE}(x,y) = x\log\frac{x}{y} + (1-x)\log\frac{1-x}{1-y}$. A comprehensive overview and probabilistic analysis of these methods can be found in Van Assel et al. (2022).

*Remark* 2.1. The usual formulations of neighbor embedding methods rely on the loss $L(x,y) = x\log(x/y)$ instead of $L_{\mathrm{KL}}$. However, due to the normalization, the total mass $\sum_{ij}[\mathbf{C}_Z(\mathbf{Z})]_{ij}$ is constant (often equal to 1) in all of the cases mentioned above. Thus the minimization in $\mathbf{Z}$ with the $L_{\mathrm{KL}}$ formulation is equivalent to the usual formulations of neighbor embedding objectives.

**Non-Euclidean geometries**. Most DR methods can also be extended to incorporate non-Euclidean geometries. For instance, Hyperbolic spaces (Chami et al., 2021; Fan et al., 2022; Guo et al., 2022; Lin et al., 2023) are of particular interest as they can capture hierarchical structures more effectively than Euclidean spaces and further mitigate the curse of dimensionality. These methods introduce additional hyperparameters and optimization difficulties that can limit their applicability. For this reason, we report some numerical experiments with such kernels in Appendix F.10 as proofs of concept supporting the versatility of our method.

## 2.2 Optimal Transport Across Spaces

Optimal Transport (OT) (Villani et al., 2009; Peyré et al., 2019) is a popular framework for comparing probability distributions and is at the core of our contributions. We review in this section the Gromov-Wasserstein formulation of OT aiming at comparing distributions across spaces.

**Gromov-Wasserstein (GW).** The GW framework (Mémoli, 2011; Sturm, 2012) comprises a collection of OT methods designed to compare distributions by examining the pairwise relations *within each space*. For two matrices $\mathbf{C} \in \mathbb{R}^{N\times N}$, $\overline{\mathbf{C}} \in \mathbb{R}^{n\times n}$, and weights $\mathbf{h} \in \Sigma_N, \overline{\mathbf{h}} \in \Sigma_n$, the GW discrepancy with inner loss $L$ (Peyré et al., 2016) is defined as

$$\mathrm{GW}_L(\mathbf{C}, \overline{\mathbf{C}}, \mathbf{h}, \overline{\mathbf{h}}) = \min_{\mathbf{T}\in\mathcal{U}(\mathbf{h},\overline{\mathbf{h}})} E_L(\mathbf{C}, \overline{\mathbf{C}}, \mathbf{T})$$
$$\text{with } E_L(\mathbf{C}, \overline{\mathbf{C}}, \mathbf{T}) = \sum_{ijkl} L(C_{ij}, \overline{C}_{kl})T_{ik}T_{jl} \,, \tag{GW}$$

and $\mathcal{U}(\mathbf{h}, \overline{\mathbf{h}}) = \left\{ \mathbf{T} \in \mathbb{R}_+^{N\times n} : \mathbf{T}\mathbf{1}_n = \mathbf{h}, \mathbf{T}^\top\mathbf{1}_N = \overline{\mathbf{h}} \right\}$ is the set of couplings between $\mathbf{h}$ and $\overline{\mathbf{h}}$. In this formulation, both pairs $(\mathbf{C}, \mathbf{h})$ and $(\overline{\mathbf{C}}, \overline{\mathbf{h}})$ can be interpreted as graphs with corresponding connectivity matrices $\mathbf{C}, \overline{\mathbf{C}}$, and where nodes are weighted by histograms $\mathbf{h}, \overline{\mathbf{h}}$ (with implicit supports). Equation (GW) is thus a *quadratic problem* (in $\mathbf{T}$) which consists in finding a soft-assignment matrix $\mathbf{T}$ that aligns the nodes of the two graphs in a way that preserves their pairwise connectivities.

From a distributional perspective, GW defines a distance between distributions that do not belong to the same metric space. For two discrete probability distributions $\mu_X = \sum_{i=1}^{N}[\mathbf{h}_X]_i \delta_{\mathbf{x}_i} \in \mathcal{P}_N(\mathbb{R}^p), \mu_Z = \sum_{i=1}^{n}[\mathbf{h}_Z]_i \delta_{\mathbf{z}_i} \in \mathcal{P}_n(\mathbb{R}^d)$ and pairwise similarity matrices $\mathbf{C}_X(\mathbf{X})$ and $\mathbf{C}_Z(\mathbf{Z})$ associated with the explicit supports $\mathbf{X} = (\mathbf{x}_1, \cdots, \mathbf{x}_n)^\top$ and $\mathbf{Z} = (\mathbf{z}_1, \cdots, \mathbf{z}_n)^\top$, the quantity $\mathrm{GW}_L(\mathbf{C}_X(\mathbf{X}), \mathbf{C}_Z(\mathbf{Z}), \mathbf{h}_X, \mathbf{h}_Z)$ is a measure of dissimilarity or discrepancy between $\mu_X, \mu_Z$. Specifically, when $L = L_2$, and $\mathbf{C}_X(\mathbf{X}), \mathbf{C}_Z(\mathbf{Z})$ are pairwise distance matrices, GW defines a proper distance between $\mu_X$ and $\mu_Z$ with respect to measure preserving isometries (with weaker assumptions on $\mathbf{C}_X, \mathbf{C}_Z$, GW defines a pseudo-metric *w.r.t.* a different notion of isomorphism (Chowdhury and Mémoli, 2019), detailed in Appendix C).

Due to its versatile properties, notably in comparing distributions over different domains, the GW problem has found many applications in machine learning, *e.g.,* for 3D meshes alignment (Solomon et al., 2016; Ezuz et al., 2017), NLP (Alvarez-Melis and Jaakkola, 2018), (co-)clustering (Peyré et al., 2016; Redko et al., 2020), single-cell analysis (Demetci et al., 2020), neuroimaging (Thual et al., 2022), graph representation learning (Xu, 2020; Vincent-Cuaz et al., 2021; Liu et al., 2022b; Vincent-Cuaz et al., 2022b; Zeng et al., 2023) and partitioning (Xu et al., 2019; Chowdhury and Needham, 2021). In this work, we leverage the GW discrepancy to extend classical DR approaches, framing them as the projection of a distribution onto a space of lower dimensionality.

## 3 OT Formulation of DR

In this section, we outline the strong connections between the classical eq. (DR) and the GW problem.

**Gromov-Monge interpretation of DR.** As suggested by eq. (DR), dimension reduction seeks to find embeddings $\mathbf{Z}$ so that the similarity between the $(i, j)$ samples of the input data is as close as possible to the similarity between the $(i, j)$ samples of the embeddings. Under equivariant assumptions on $\mathbf{C}_Z$, this also amounts to identifying the embedding $\mathbf{Z}$ *and* the best permutation that realigns the two similarity matrices $\mathbf{C}_X(\mathbf{X})$ and $\mathbf{C}_Z(\mathbf{Z})$. Recall that the function $\mathbf{C}_Z$ is equivariant by permutation, if, for any $N \times N$ permutation matrix $\mathbf{P}$ and any $\mathbf{Z}$, $\mathbf{C}_Z(\mathbf{PZ}) = \mathbf{PC}_Z(\mathbf{Z})\mathbf{P}^\top$ (Bronstein et al., 2021). This type of assumption is natural for $\mathbf{C}_Z$: if we rearrange the order of samples (*i.e.* the rows of $\mathbf{Z}$), we expect the similarity matrix between the samples to undergo the same rearrangement.

**Lemma 3.1.** *Let* $\mathbf{C}_Z$ *be a permutation equivariant function and* $L$ *any loss. The minimum of eq.* (DR) *is equal to*

$$\min_{\mathbf{Z} \in \mathbb{R}^{N \times d}} \min_{\sigma \in S_N} \sum_{ij} L([\mathbf{C}_X(\mathbf{X})]_{ij}, [\mathbf{C}_Z(\mathbf{Z})]_{\sigma(i)\sigma(j)}) . \tag{2}$$

*Also, any sol.* $\mathbf{Z}$ *of eq.* (DR) *is such that* $(\mathbf{Z}, \mathrm{id})$ *is solution of eq.* (2). *And conversely any* $(\mathbf{Z}, \sigma)$ *sol. of eq.* (2) *is such that* $\mathbf{Z}$ *is a solution of eq.* (DR) *up to* $\sigma$.

Lemma 3.1, proven in Appendix B.1, establishes an equivalence between eq. (DR) and the optimization of the embedding $\mathbf{Z}$ w.r.t a Gromov-Monge discrepancy (Mémoli and Needham, 2018) given in eq. (2), which seeks to identify the permutation $\sigma$ that best aligns two similarity matrices, by solving a combinatorial quadratic assignment problem (Cela, 2013). We can delve deeper into these comparisons and demonstrate that the general formulation of dimension reduction is also equivalent to minimizing the Gromov-Wasserstein objective, which serves as a relaxation of the Gromov-Monge problem (Mémoli and Needham, 2022).

**DR as GW Minimization.** We suppose that the distributions have the same number of points $(N = n)$ and uniform weights $(\mathbf{h}_Z = \mathbf{h}_X = \frac{1}{N}\mathbf{1}_N)$. We recall that a matrix $\mathbf{C} \in \mathbb{R}^{N \times N}$ is conditionally positive definite (CPD), *resp.* conditionally negative definite (CND), if it is symmetric and $\forall \mathbf{x} \in \mathbb{R}^N$ s.t. $\mathbf{x}^\top \mathbf{1}_N = 0$ we have $\mathbf{x}^\top \mathbf{C} \mathbf{x} \geq 0$, *resp.* $\leq 0$.

We thus extend Lemma 3.1 to the GW problem in the following (proof in Appendix B.2):

**Theorem 3.2.** *The minimum eq.* (DR) *is equal to* $\min_{\mathbf{Z}} \mathrm{GW}_L(\mathbf{C}_X(\mathbf{X}), \mathbf{C}_Z(\mathbf{Z}), \frac{1}{N}\mathbf{1}_N, \frac{1}{N}\mathbf{1}_N)$ *in the following cases:*

  (i) *(spectral methods) When* $\mathbf{C}_X(\mathbf{X})$ *is any matrix,* $L = L_2$ *and* $\mathbf{C}_Z(\mathbf{Z}) = \mathbf{Z}\mathbf{Z}^\top$.

(ii) (neighbor embedding methods) When $\mathrm{Im}(\mathbf{C}_X) \subseteq \mathbb{R}_{>0}^{N \times N}$, $L = L_{\mathrm{KL}}$, the matrix $\mathbf{C}_X(\mathbf{X})$ is CPD and, for any $\mathbf{Z}$,

$$\mathbf{C}_Z(\mathbf{Z}) = \mathrm{diag}(\boldsymbol{\alpha}_{\mathbf{Z}})\mathbf{K}_{\mathbf{Z}}\,\mathrm{diag}(\boldsymbol{\beta}_{\mathbf{Z}}), \tag{3}$$

for some $\boldsymbol{\alpha}_{\mathbf{Z}}, \boldsymbol{\beta}_{\mathbf{Z}} \in \mathbb{R}_{>0}^N$ and $\mathbf{K}_{\mathbf{Z}} \in \mathbb{R}_{>0}^{N \times N}$ such that $\log(\mathbf{K}_{\mathbf{Z}})$ is CPD.

Remarkably, the first item of Theorem 3.2 shows that *all spectral DR methods* can be seen as OT problems in disguise, as they all equivalently minimize a GW problem. The second item also provides some insights into this equivalence in the case of neighbor embedding methods that require more assumptions. For instance, the Gaussian kernel $\mathbf{K}_{\mathbf{Z}}$, used extensively in DR (Section 2.1), satisfies the hypothesis as $\log(\mathbf{K}_{\mathbf{Z}}) = (-\|\mathbf{z}_i - \mathbf{z}_j\|_2^2)_{ij}$ is CPD (see *e.g.* Maron and Lipman 2018). The terms $\boldsymbol{\alpha}_{\mathbf{Z}}, \boldsymbol{\beta}_{\mathbf{Z}}$ also allow for considering all the usual normalizations of $\mathbf{K}_{\mathbf{Z}}$: by a scalar so as to have $\sum_{ij}[\mathbf{C}_Z(\mathbf{Z})]_{ij} = 1$, but also any row or doubly stochastic normalizations (with the Sinkhorn-Knopp algorithm Sinkhorn and Knopp 1967).

Matrices satisfying $\log(\mathbf{K}_{\mathbf{Z}})$ being CPD are well-studied in the literature and are known as infinitely divisible matrices (Bhatia, 2006). It is noteworthy that the t-Student kernel does not fall into this category. Moreover, in the aforementioned neighbor embedding methods, the matrix $\mathbf{C}_X(\mathbf{X})$ is generally not CPD. The intriguing question of generalizing this result with weaker assumptions on $\mathbf{C}_Z$ and $\mathbf{C}_X$ remains open for future research. Interestingly, we have observed in the numerical experiments performed in Section 5 that the symmetric entropic affinity of Van Assel et al. (2023) was systematically CPD.

*Remark* 3.3. In Corollary B.3 of the appendix we also provide other sufficient conditions for neighbor embedding methods with the cross-entropy loss $L(x, y) = x \log(x/y)$. They rely on specific structures for $\mathbf{C}_Z$ but do not impose any assumptions on $\mathbf{C}_X$. Additionally, in Appendix B.3, we provide a *necessary* condition based on a bilinear relaxation of the GW problem. Although its applicability is limited due to challenges in proving it in full generality, it requires minimal assumptions on $\mathbf{C}_X, \mathbf{C}_Z$ and $L$.

In essence, both Lemma 3.1 and Theorem 3.2 indicate that dimensionality reduction can be reframed from a distributional perspective, with the search for an empirical distribution that aligns with the data distribution in the sense of optimal transport, through the GW framework. In other words, DR is informally solving $\min_{\mathbf{z}_1, \cdots, \mathbf{z}_N} \mathrm{GW}(\frac{1}{n}\sum_{i=1}^N \delta_{\mathbf{x}_i}, \frac{1}{n}\sum_{i=1}^N \delta_{\mathbf{z}_i})$.

## 4 Distributional Reduction

The above perspective on DR allows for the two following generalizations. Firstly, beyond solely determining the positions $\mathbf{z}_i$ of Diracs (as in classical DR) we can now optimize *the mass* of the distribution $\mu_Z$. This is interpreted as finding the relative importance of each point in the embedding $\mathbf{Z}$. More importantly, due to the flexibility of GW, we can also seek a distribution in the embedding with a smaller number of points $n < N$. This will result in both reducing the dimension *and* clustering the points in the embedding space through the optimal coupling. Informally, we thus propose a new *Distributional Reduction* (DistR) framework that aims at solving $\min_{\mu_Z \in \mathcal{P}_n(\mathbb{R}^d)} \mathrm{GW}(\frac{1}{n}\sum_{i=1}^N \delta_{\mathbf{x}_i}, \mu_Z)$.

### 4.1 Distributional Reduction Problem

Precisely, the novel optimization problem that we tackle in this paper can be formulated as follows

$$\min_{\substack{\mathbf{Z} \in \mathbb{R}^{n \times d} \\ \mathbf{h}_Z \in \Sigma_n}} \mathrm{GW}_L(\mathbf{C}_X(\mathbf{X}), \mathbf{C}_Z(\mathbf{Z}), \mathbf{h}_X, \mathbf{h}_Z) \tag{DistR}$$

This problem comes down to learning the graph $(\mathbf{C}_Z(\mathbf{Z}), \mathbf{h}_Z)$ parametrized by $\mathbf{Z}$ that is the closest from $(\mathbf{C}_X(\mathbf{X}), \mathbf{h}_X)$ in the GW sense. When $n < N$, the embeddings then act as *low-dimensional prototypical examples* of input samples, whose learned relative importance $\mathbf{h}_Z$ accommodates clusters of varying proportions in the input data $\mathbf{X}$ (see Section 1). We refer to them as *prototypes*. The weight vector $\mathbf{h}_X$ is typically assumed to be uniform, that is $\mathbf{h}_X = \frac{1}{N}\mathbf{1}_N$, in the absence of prior knowledge. As discussed in Section 3, traditional DR amounts to setting $n = N, \mathbf{h}_Z = \frac{1}{N}\mathbf{1}_N$.

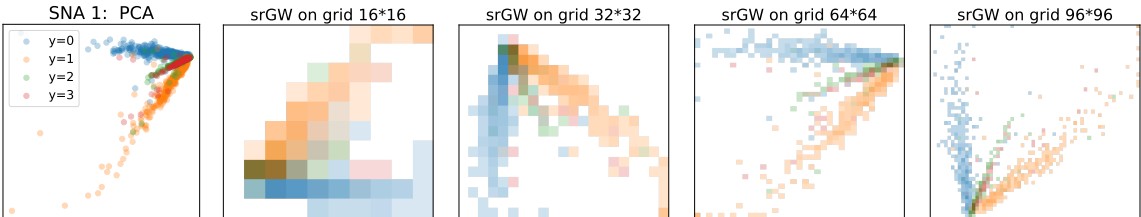

Figure 2: GW projections of a genomics dataset (Chen et al., 2019) on regular grids with increasing resolutions, respectively encoded as $C_X(\mathbf{X}) = \mathbf{X}\mathbf{X}^\top$ and $C_Z(\mathbf{Z}) = \mathbf{Z}\mathbf{Z}^\top$. Pixels on cropped grids are colored by interpolating samples' colors according to the transport plan and their intensity is proportional to their mass.

One notable aspect of our model is its capability to simultaneously perform DR and clustering. Indeed, the optimal coupling $\mathbf{T} \in [0,1]^{N \times n}$ of problem eq. (DistR) is, by construction, a soft-assignment matrix from the input data to the embeddings. It allows each point $\mathbf{x}_i$ to be linked to one or more prototypes $\mathbf{z}_j$ (clusters). In Section 4.2 we explore conditions where these soft assignments transform into hard ones, such that each point is therefore linked to a unique prototype/cluster.

**A semi-relaxed objective.** For a given embedding $\mathbf{Z}$ and $L = L_2$, it is known that minimizing the DistR objective w.r.t $\mathbf{h}_Z$ is *equivalent* to a problem that is computationally simpler than the usual GW one, namely the semi-relaxed GW divergence srGW$_L$ (Vincent-Cuaz et al., 2022a) given by

$$\min_{\mathbf{T} \in \mathcal{U}_n(\mathbf{h}_X)} E_L(\mathbf{C}_X(\mathbf{X}), \mathbf{C}_Z(\mathbf{Z}), \mathbf{T}) , \qquad \text{(srGW)}$$

where $\mathcal{U}_n(\mathbf{h}_X) := \left\{ \mathbf{T} \in \mathbb{R}_+^{N \times n} : \mathbf{T}\mathbf{1}_n = \mathbf{h}_X \right\}$. To efficiently address eq. (DistR), we first observe that this equivalence holds for any inner divergence $L$. Additionally, we prove that srGW$_L$ remains a divergence as soon as $L$ is itself a divergence. Consequently, srGW$_L$ vanishes iff both compared measures are isomorphic in a weak sense (Chowdhury and Mémoli, 2019). We emphasize that taking a proper divergence $L$ is important (and basic assumptions on $\mathbf{X}$), as it avoids some trivial solutions. These results are detailed in Appendix C.

Interestingly, srGW projections, *i.e.* optimizing only the weights $\mathbf{h}_Z$ over simple fixed supports $\mathbf{Z}$, have already remarkable representational capability. We illustrate this in Figure 2, by considering projections of a real-world dataset over 2D grids of increasing resolutions. Setting $\mathbf{C}_X(\mathbf{X}) = \mathbf{X}\mathbf{X}^\top$ and $\mathbf{C}_Z(\mathbf{X}) = \mathbf{Z}\mathbf{Z}^\top$, we can see that those projections recover faithful coarsened representations of the embeddings learned using PCA. DistR aims to exploit the full potential of this divergence by learning a few optimal prototypes that best represent the dataset.

**Computation.** DistR is a non-convex problem that we propose to tackle using a Block Coordinate Descent algorithm (BCD, Tseng 2001) guaranteed to converge to local optimum (Grippo and Sciandrone, 2000; Lyu and Li, 2023). The BCD alternates between the two following steps. First, we optimize in $\mathbf{Z}$ for a fixed transport plan using gradient descent with adaptive learning rates (Kingma and Ba, 2014). Then we solve for a srGW problem given $\mathbf{Z}$. To this end, we extended both solvers proposed in Vincent-Cuaz et al. (2022a) to support losses $L_2$, $L_{\text{KL}}$ and $L_{BCE}$ (see Appendix E.1). Namely, the Mirror Descent promoting smoothness along iterations depending on a hyparameter $\varepsilon > 0$ and the Conditional Gradient seen as the edge-case $\varepsilon = 0$. Following Proposition 1 in (Peyré et al., 2016), a vanilla implementation leads to $\mathcal{O}(nN^2 + n^2N)$ operations to compute the loss or its gradient. However in many DR methods, $\mathbf{C}_X(\mathbf{X})$ or $\mathbf{C}_Z(\mathbf{Z})$, or their transformations within the loss $L$, admit explicit low-rank factorizations. Including *e.g.* matrices involved in spectral methods and other similarity matrices derived from squared Euclidean distance matrices (Scetbon et al., 2022). In these settings, we exploit these factorizations to reduce the computational complexity of our solvers down to $\mathcal{O}(Nn(p+d) + (N+n)pd + min(n^2, nd^2))$ when $L = L_2$, and $O(Nnd + n^2d)$ when $L = L_{\text{KL}}$. We refer the reader interested in these algorithmic details to Appendix E. As shown in Appendix F.9, DistR already runs in a few seconds on datasets with several thousands of samples without leveraging low-rank properties.

**Related work.** The CO-Optimal Transport (COOT) clustering approach proposed in Redko et al. (2020) is the closest to our work. It simultaneously estimates sample and feature clustering and thus can be

directly applied to perform joint clustering and DR. Despite SOTA performances on co-clustering tasks, COOT-clustering corresponds to a linear DR method which is limiting for joint clustering and DR tasks. In Appendix F.2, we show that COOT leads to a low average homogeneity of the prototypes $\{\mathbf{z}_k\}_{k\in[\![n]\!]}$, indicating that it often assigns points with different labels to the same prototype. In contrast, DistR exploits the more expressive non-linear similarity functions offered by existing DR methods and leads to significantly improved results. Other joint DR-clustering approaches, such as Liu et al. (2022a), involve modeling latent variables by a mixture of distributions. In comparison, DistR is more versatile, as it easily adapts to any $(L, \mathbf{C}_X, \mathbf{C}_Z)$ of existing DR methods.

## 4.2 Clustering Properties

We elaborate now on the links between DistR and clustering methods. In what follows, we call a coupling $\mathbf{T} \in [0,1]^{N \times n}$ with a single non-null element per row a *membership matrix*. When the coupling is a membership matrix each data point is associated with a single prototype thus achieving a hard clustering of the input samples. We will see that a link can be drawn with kernel k-means using the analogy of *GW barycenters*. More precisely the *srGW barycenter* (Vincent-Cuaz et al., 2022a) seeks for a closest target graph $(\overline{\mathbf{C}}, \overline{\mathbf{h}})$ from $(\mathbf{C}_X, \mathbf{h}_X)$ by solving

$$\min_{\overline{\mathbf{C}} \in \mathbb{R}^{n \times n}} \min_{\mathbf{T} \in \mathcal{U}_n(\mathbf{h}_X)} E_L(\mathbf{C}_X(\mathbf{X}), \overline{\mathbf{C}}, \mathbf{T}) \,. \tag{srGWB}$$

We stress that the only (important) difference between eq. (srGWB) and eq. (DistR) is that there is no constraint imposed on $\overline{\mathbf{C}}$ in srGWB. In contrast, eq. (DistR) looks for minimizing over $\overline{\mathbf{C}} \in \{\mathbf{C}_Z(\mathbf{Z}) : \mathbf{Z} \in \mathbb{R}^{N \times d}\}$. For instance, choosing $\mathbf{C}_Z(\mathbf{Z}) = \mathbf{Z}\mathbf{Z}^\top$ in eq. (DistR) is equivalent to enforcing $\text{rank}(\overline{\mathbf{C}}) \leq d$ in eq. (srGWB).

We establish below that srGWB is of particular interest for clustering. The motivation for this arises from the findings of (Chen et al., 2023), which demonstrate that, when $\mathbf{C}_X(\mathbf{X})$ is positive semi-definite and $\mathbf{T}$ is *constrained* to belong to the set of membership matrices (as opposed to couplings in $\mathcal{U}_n(\mathbf{h})$), eq. (srGWB) is equivalent to a kernel k-means whose samples are weighted by $\mathbf{h}_X$ (Dhillon et al., 2004; 2007). Interestingly, these additional constraints are unnecessary. Indeed, we show below that the original srGWB problem admits membership matrices as the optimal coupling for a broader class of $\mathbf{C}_X(\mathbf{X})$ input matrices for $L = L_2$ (see proof in Appendix D).

**Theorem 4.1.** *Let* $\mathbf{h}_X \in \Sigma_N$ *and* $L = L_2$. *Suppose that for any* $\mathbf{X} \in \mathbb{R}^{N \times p}$ *the matrix* $\mathbf{C}_X(\mathbf{X})$ *is CPD or CND. Then the problem eq.* (srGWB) *admits a membership matrix as optimal coupling, i.e. , there is a minimizer of* $\mathbf{T} \in \mathcal{U}_n(\mathbf{h}_X) \to \min_{\overline{\mathbf{C}} \in \mathbb{R}^{n \times n}} E_L(\mathbf{C}_X(\mathbf{X}), \overline{\mathbf{C}}, \mathbf{T})$ *with only one non-zero value per row.*

The implications of this theorem are as follows. First, as shown in Appendix D, given an optimal plan $\mathbf{T}$, the coefficient $(i,j)$ of the barycenter $\overline{\mathbf{C}}$ can be written as $\frac{1}{n_i n_j} \sum_{pq} [\mathbf{C}_X(\mathbf{X})]_{pq} T_{pi} T_{qj}$ where $n_i = \sum_p T_{pi}$. Consequently, when $\mathbf{T}$ is a membership matrix, $\overline{\mathbf{C}}$ represent graph weights, where each element $(i,j)$ is the weighted sum of the original graph weights $\mathbf{C}_X(\mathbf{X})$ for nodes in the clusters of $i$ and $j$, following standard edge contraction methods (Loukas, 2019). Second, Theorem 4.1 and relations proven in Chen et al. (2023) state that eq. (srGWB) is equivalent to the aforementioned kernel k-means when $\mathbf{C}_X(\mathbf{X})$ is positive semi-definite. This equivalence also emphasizes spectrum-preservating properties of the srGWB problem, whose optimal value satisfies $E_{L_2}(\mathbf{C}_X(\mathbf{X}), \overline{\mathbf{C}}, \mathbf{T}) = \sum_{i=1}^N \lambda_i^2 - \sum_{j=1}^n \nu_j^2$, where $\{\lambda_i\}_{i=1}^N$ and $\{\nu_j\}_{j=1}^n$ are the eigenvalues sorted in descending order of $\text{diag}(\mathbf{h}_X)^{-1/2} \mathbf{C}_X(\mathbf{X}) \text{diag}(\mathbf{h}_X)^{-1/2}$ and $\overline{\mathbf{C}}$ respectively, and satisfy the Pointcaré Separation Theorem. Finally, as the (hard) clustering property holds for more generic types of matrices, namely CPD and CND, srGWB stands out as a fully-fledged clustering method. Although these results do not apply directly to DistR, except if *e.g.* the dimension $d$ is set to the unknown rank of the srGW barycenter, we argue that they further legitimize the use of GW projections for clustering. Interestingly, we also observe in practice that the couplings obtained by DistR are always membership matrices, regardless of $\mathbf{C}_Z$.

## 5 Numerical Experiments

In this section, we demonstrate the relevance of our approach for joint clustering and DR, over 8 labeled datasets detailed in Appendix F including: 3 image datasets (COIL-20 Nene et al. 1996, MNIST & fashion-

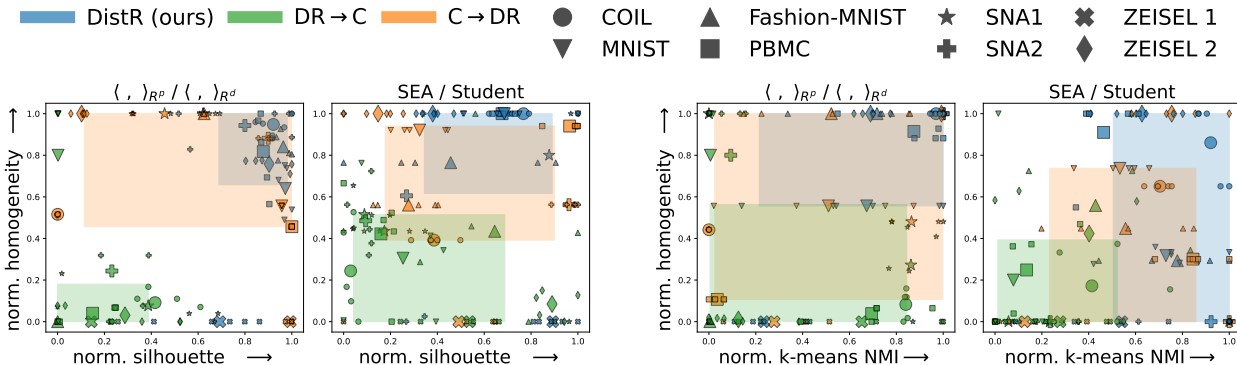

Figure 3: Best trade-off between homogeneity vs silhouette (2 first plots), and homogeneity vs NMI (2 last plots). Scores are normalized in $[0, 1]$ via min-max scaling over a dataset. Small markers are scores per dataset for 5 runs, while big ones are their mean. We illustrate the 20-80% percentiles of scores per method as a colored surface.

MNIST Xiao et al. 2017) and 5 genomic ones (PBMC Wolf et al. 2018, SNA 1 & 2 Chen et al. 2019 and ZEISEL 1 & 2Zeisel et al. 2015). To this end, we examine how effectively DistR prototypes offer discriminative representations of input samples at various granularities, using an evaluation system described below. Code is provided at `https://github.com/huguesva/Distributional-Reduction`.

**Benchmarked methods.** Let us recall that any DR method presented in Section 2.1 is fully characterized by a triplet $(L, \mathbf{C}_X, \mathbf{C}_Z)$ of loss and pairwise similarity functions. Given such triplet, we compare our DistR model against sequential approaches of DR and clustering, namely *DR then clustering* (DR→C) and *Clustering then DR* (C→DR). These 2-step methods are representative of current practice in the field of joint clustering and DR (Baran et al., 2019). In both cases, the clustering step is performed with spectral clustering (a benchmark with Kmeans clustering is provided in Appendix F.7). The three methods produce a sample-to-prototype association matrix (formally described in Appendix F.1) further used to evaluate performance. For the choices of $(L, \mathbf{C}_X, \mathbf{C}_Z)$, we experiment with both spectral and neighbor embedding methods (NE). For the former, we consider the usual PCA setting with $d = 10$. Regarding NE, we rely on the *Symmetric Entropic Affinity* (SEA)[1] from Van Assel et al. (2023) for $\mathbf{C}_X$ and the scalar-normalized student similarity for $\mathbf{C}_Z$ (Van der Maaten and Hinton, 2008), setting the dimension $d = 2$ as it coincides with real-life visualization purposes while outperforming PCA-based approaches. We report validated hyperparemeters (*e.g perplexity, number of prototypes $n$*) and implementation details in Appendix F.1, and provide our code with the submission. Moreover, we report additional experiments using the vanilla tSNE and UMAP kernels in Appendix F, leading to similar behaviors than kernels reported in the main paper.

**Evaluation metrics.** We consider several evaluation metrics. Firstly, we wonder whether the prototypes are *individually* discriminating input samples $\{\mathbf{x}_i\}$ w.r.t their class label. We consider the homogeneity score (Rosenberg and Hirschberg, 2007) that quantifies to which extent each prototype $\mathbf{z}_k$ is associated with points in the same class. Secondly, we aim to evaluate whether the embedding space is also *globally discriminant*, *i.e.* that prototypes organize themselves to form clusters that match the classes, measured by two metrics. *i)* We associate with each prototype $\mathbf{z}_k$ a label using majority voting from the associated samples. Then we make use of the popular silhouette score (Rousseeuw, 1987) to evaluate whether prototypes' positions are correctly aligned with these labels. *ii)* As commonly done to assess DR representations (Huang et al., 2022), we also consider the downstream task of clustering the prototypes using the k-means algorithm. We then assign to each $\{\mathbf{x}_i\}$ the cluster of its associated prototype, acting as a predicted label, and compute the Normalized Mutual Information (NMI) (Kvålseth, 2017) between these labels and $\{y_i\}$. More details about evaluation metrics can be found in Appendix F.

**Results.** First, we illustrate in Figure 3 the best trade-off between the aforementioned metrics achieved by all methods. For each method and dataset, we considered the model maximizing the sum of the two

---

[1]An enhanced version of the original tSNE affinity that preserves entropy normalization during symmetrization.

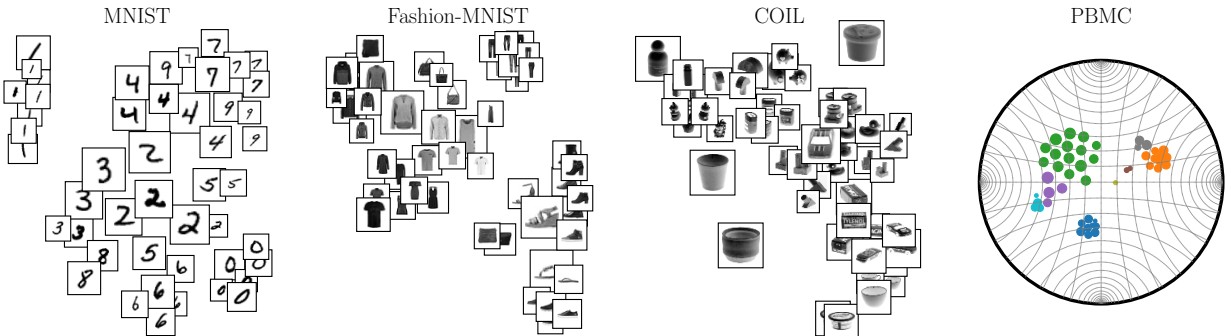

Figure 4: Examples of embeddings produced by DistR, using the SEA similarity for $\mathbf{C}_X$ and the Student's kernel for $\mathbf{C}_Z$, respectively in $\mathbb{R}^2$ for the first three datasets and the Poincaré ball for the last one. Displayed images are medoids for each cluster *i.e.* $\arg\max_i [\mathbf{C}_X(\mathbf{X})\mathbf{T}_{:,k}]_i$ for cluster $k$. The area of image $k$ is proportional to $[\mathbf{h}_Z]_k$.

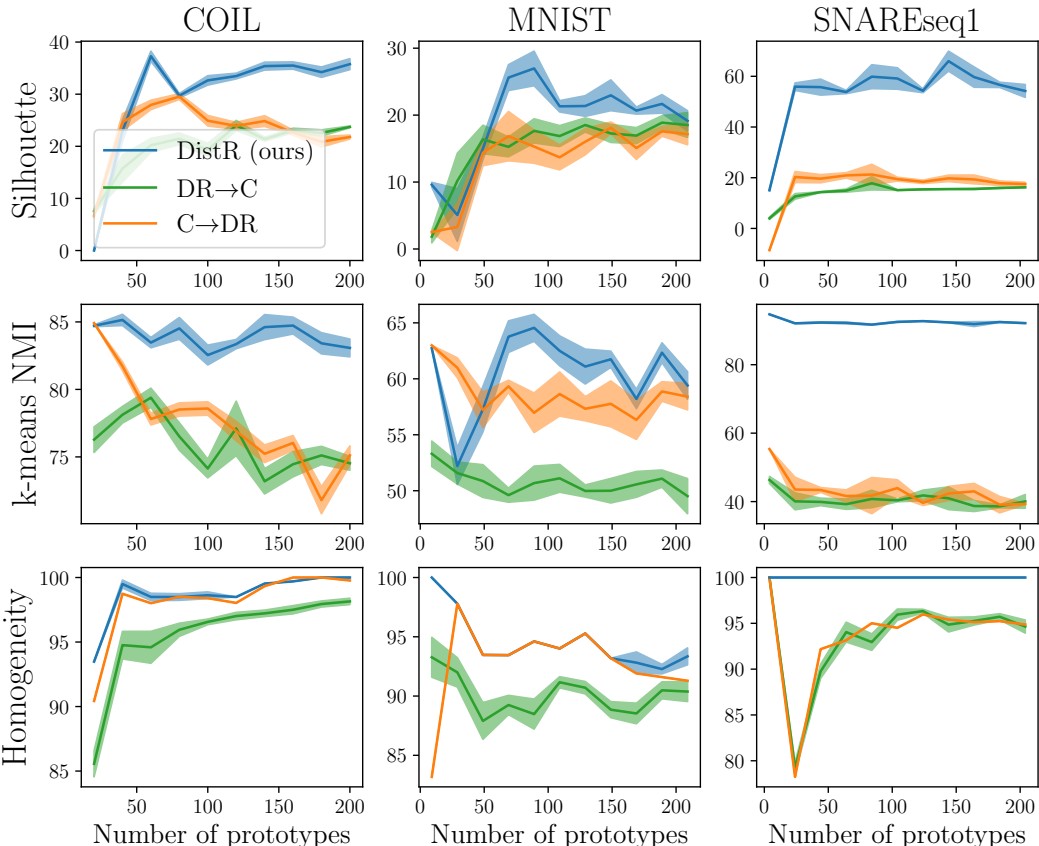

Figure 5: Scores ($\times 100$) w.r.t with respect to the number of prototypes (in $\mathbb{R}^2$) using the t-SNE model: SEA similarity for $\mathbf{C}_X$ (Van Assel et al., 2023), Student's kernel for $\mathbf{C}_Z$ and loss $L_{\mathrm{KL}}$.

normalized metrics to account for their different ranges. DistR, being present on the top-right of all plots, provides on average the most discriminant low-dimensional representations endowed with a simple geometry, seconded by C→DR. The significant variance of these dynamics for all methods emphasize the difficulty of performing jointly clustering and DR. Interestingly, we show in Appendices F.5-F.6 that DistR is the best suited to describing most datasets at various granularities in low-dimension. For a given configuration, DistR leads to the most consistent performance over $n$ on average across datasets. We also check that this consistency holds for different dimensions $d$ in Appendix F.8.

We then display in Figure 5 the scores obtained with the neighbor embedding model for several datasets and

across all considered number of prototypes $n$. Other kernels and datasets are illustrated in Appendix F.4. Interestingly, looking at the homogeneity score (third row), one can notice that DistR is at least as good as C→DR for grouping points of the same label, even though C→DR performs clustering in the high dimensional input space. DistR is also generally better than sequential approaches at representing a meaningful structure of the prototypes in low-dimension, measured by the silhouette and k-means NMI scores. Therefore, our approach DistR seems to effectively achieve the best equilibrium between homogeneity and preservation of the structure of the prototypes. For a qualitative illustration, we also plot some embeddings produced by our method in Figure 4.

## 6 Conclusion

By making a connection between the GW problem and popular clustering and DR algorithms, we proposed a unifying framework denoted as DistR. DistR enables transforming any DR algorithm into a joint DR-clustering method that produces embeddings (or *prototypes*) with chosen granularity. We believe that the versatility of the GW framework will enable new extensions in unsupervised learning. For instance, the formalism associated with (semi-relaxed) GW barycenters naturally enables addressing multi-view settings with multiple unaligned inputs of different sizes. Other promising directions involve, better capturing multiple dependency scales in the input data by hierarchically adapting the resolution of the embedding similarity graph, or enabling batch optimization of embeddings to operate over much larger datasets.

## Acknowledgments

The authors are grateful to Aurélien Garivier for insightful discussions. This project was supported in part by the ANR project OTTOPIA ANR-20-CHIA-0030 and through the MATTER project ANR-23-ERCC-0006-01. We gratefully acknowledge the support of the Centre Blaise Pascal's IT test platform at ENS de Lyon (Lyon, France) for Machine Learning facilities. The platform operates the SIDUS solution (Quemener and Corvellec, 2013). This work benefited from state aid managed by the Agence Nationale de la Recherche under the France 2030 programme, reference ANR-23-IACL-0005. Finally, it received funding from the Fondation de l'École polytechnique and the Personalized Health and Related Technologies PHRT project 2022-644.

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

# A   Appendix

**Outline of the appendix:**

- Appendix B: provide the proofs for the results stated in Section 3, namely the one for Lemma 3.1 in Appendix B.1; for Theorem 3.2 in Appendix B.2 and additional necessary and sufficient conditions under different assumptions as discussed in Remark 3.3 developed in Appendix B.3

- Appendix C: provide the definition of weak isomorphism in the GW framework, proofs regarding the characterization of the generalized srGW discrepancy as a divergence, mentioned in Section 4.1 of the main paper.

- Appendix D: proof of Theorem 4.1 on the clustering properties of srGW barycenters.

- Appendix E: Algorithmic details for DistR under generic and low-rank settings.

- Appendix F: Several additional information and results regarding the experiments detailed in Section 5 of the main paper.

  - Appendix F.1 provides details on methods' implementation, validation of hyperparameters, datasets and metrics.
  - Appendix F.2 compares DistR with COOT clustering.
  - Appendix F.3: Best trade-off between metrics using t-SNE with $l_2$ symmetrization and UMAP.
  - Appendix F.4 reports complete scores on all datasets.
  - Appendix F.5 and F.6 study homogeneity vs silhouette and homogeneity vs kmeans NMI scores for various numbers of prototypes.
  - Appendix F.8 study the sensitivity of benchmarked methods to the embedding dimension $d$ using spectral DR methods.
  - Appendix F.9 compares computation time across methods.
  - Appendix F.10 detail our proofs of concepts with hyperbolic DR kernels.

# B   Proof of results in Section 3

## B.1   Proof of lemma 3.1

We recall the result.

**Lemma 3.1.** *Let* $\mathbf{C}_Z$ *be a permutation equivariant function and* $L$ *any loss. The minimum of eq.* (DR) *is equal to*

$$\min_{\mathbf{Z} \in \mathbb{R}^{N \times d}} \min_{\sigma \in S_N} \sum_{ij} L([\mathbf{C}_X(\mathbf{X})]_{ij}, [\mathbf{C}_Z(\mathbf{Z})]_{\sigma(i)\sigma(j)}) . \tag{2}$$

*Also, any sol.* $\mathbf{Z}$ *of eq.* (DR) *is such that* $(\mathbf{Z}, \mathrm{id})$ *is solution of eq.* (2). *And conversely any* $(\mathbf{Z}, \sigma)$ *sol. of eq.* (2) *is such that* $\mathbf{Z}$ *is a solution of eq.* (DR) *up to* $\sigma$.

*Proof.* By suboptimality of $\sigma = \mathrm{id}$ we clearly have

$$\min_{\mathbf{Z} \in \mathbb{R}^{N \times d}} \min_{\sigma \in S_N} \sum_{ij} L([\mathbf{C}_X(\mathbf{X})]_{ij}, [\mathbf{C}_Z(\mathbf{Z})]_{\sigma(i)\sigma(j)}) \leq \min_{\mathbf{Z} \in \mathbb{R}^{N \times d}} \sum_{ij} L([\mathbf{C}_X(\mathbf{X})]_{ij}, [\mathbf{C}_Z(\mathbf{Z})]_{ij}). \tag{4}$$

For the other direction, take an optimal solution $(\mathbf{Z}, \sigma)$ of eq. (2). Using the permutation equivariance of $\mathbf{C}_Z$, $[\mathbf{C}_Z(\mathbf{Z})]_{\sigma(i)\sigma(j)} = [\mathbf{P}\mathbf{C}_Z(\mathbf{Z})\mathbf{P}^\top]_{ij} = [\mathbf{C}_Z(\mathbf{P}\mathbf{Z})]_{ij}$ for some permutation matrix $\mathbf{P}$. But $\mathbf{P}\mathbf{Z}$ is admissible for problem eq. (DR). Hence

$$\min_{\mathbf{Z} \in \mathbb{R}^{N \times d}} \min_{\sigma \in S_N} \sum_{ij} L([\mathbf{C}_X(\mathbf{X})]_{ij}, [\mathbf{C}_Z(\mathbf{Z})]_{\sigma(i)\sigma(j)}) \geq \min_{\mathbf{Z} \in \mathbb{R}^{N \times d}} \sum_{ij} L([\mathbf{C}_X(\mathbf{X})]_{ij}, [\mathbf{C}_Z(\mathbf{Z})]_{ij}). \tag{5}$$

$\square$

## B.2 Proof of theorem 3.2

In the following DS is the space of $N \times N$ doubly stochastic matrices. We begin by proving the first point of theorem 3.2. We will rely on the simple, but useful, result below.

**Proposition B.1.** *Let* $\Omega \subseteq \mathbb{R}$ *and* $\mathrm{Im}(\mathbf{C}_X) \subseteq \Omega^{N \times N}$. *Suppose that* $L(a, \cdot)$ *is convex for any* $a \in \Omega$ *and*

$$\min_{\mathbf{Z} \in \mathbb{R}^{N \times d}} \sum_{ij} L([\mathbf{C}_X(\mathbf{X})]_{ij}, [\mathbf{C}_Z(\mathbf{Z})]_{ij}) \leq \min_{\mathbf{Z} \in \mathbb{R}^{N \times d}, \mathbf{T} \in \mathrm{DS}} \sum_{ij} L([\mathbf{C}_X(\mathbf{X})]_{ij}, [\mathbf{T}\mathbf{C}_Z(\mathbf{Z})\mathbf{T}^\top]_{ij}) . \tag{6}$$

*Then the minimum eq.* (DR) *is equal to* $\min_{\mathbf{Z}} \mathrm{GW}_L(\mathbf{C}_X(\mathbf{X}), \mathbf{C}_Z(\mathbf{Z}), \frac{1}{N}\mathbf{1}_N, \frac{1}{N}\mathbf{1}_N)$.

*Proof.* Consider any doubly stochastic matrix $\mathbf{T}$ and note that $[\mathbf{T}\mathbf{C}_Z(\mathbf{Z})\mathbf{T}^\top]_{ij} = \sum_{kl}[\mathbf{C}_Z(\mathbf{Z})]_{kl} T_{ik} T_{jl}$. Using the convexity of $L(a, \cdot)$ for any $a \in \Omega$ and Jensen's inequality we have

$$\begin{aligned} \sum_{ij} L([\mathbf{C}_X(\mathbf{X})]_{ij}, [\mathbf{T}\mathbf{C}_Z(\mathbf{Z})\mathbf{T}^\top]_{ij}) &= \sum_{ij} L([\mathbf{C}_X(\mathbf{X})]_{ij}, \sum_{kl}[\mathbf{C}_Z(\mathbf{Z})]_{kl} T_{ik} T_{jl}) \\ &\leq \sum_{ijkl} L([\mathbf{C}_X(\mathbf{X})]_{ij}, [\mathbf{C}_Z(\mathbf{Z})]_{kl}) T_{ik} T_{jl} . \end{aligned} \tag{7}$$

In particular

$$\min_{\mathbf{Z}} \min_{\mathbf{T} \in \mathrm{DS}} \sum_{ij} L([\mathbf{C}_X(\mathbf{X})]_{ij}, [\mathbf{T}\mathbf{C}_Z(\mathbf{Z})\mathbf{T}^\top]_{ij}) \leq \min_{\mathbf{Z}} \min_{\mathbf{T} \in \mathrm{DS}} \sum_{ijkl} L([\mathbf{C}_X(\mathbf{X})]_{ij}, [\mathbf{C}_Z(\mathbf{Z})]_{kl}) T_{ik} T_{jl} . \tag{8}$$

Hence, using eq. (6),

$$\min_{\mathbf{Z}} \sum_{ij} L([\mathbf{C}_X(\mathbf{X})]_{ij}, [\mathbf{C}_Z(\mathbf{Z})]_{ij}) \leq \min_{\mathbf{Z}} \min_{\mathbf{T} \in \mathrm{DS}} \sum_{ijkl} L([\mathbf{C}_X(\mathbf{X})]_{ij}, [\mathbf{C}_Z(\mathbf{Z})]_{kl}) T_{ik} T_{jl} . \tag{9}$$

But the converse inequality is also true by sub-optimality of $\mathbf{T} = \mathbf{I}_N$ for the problem $\min_{\mathbf{Z}} \min_{\mathbf{T} \in \mathrm{DS}} \sum_{ijkl} L([\mathbf{C}_X(\mathbf{X})]_{ij}, [\mathbf{C}_Z(\mathbf{Z})]_{kl}) T_{ik} T_{jl}$. Overall

$$\min_{\mathbf{Z}} \sum_{ij} L([\mathbf{C}_X(\mathbf{X})]_{ij}, [\mathbf{C}_Z(\mathbf{Z})]_{ij}) = \min_{\mathbf{Z}} \min_{\mathbf{T} \in \mathrm{DS}} \sum_{ijkl} L([\mathbf{C}_X(\mathbf{X})]_{ij}, [\mathbf{C}_Z(\mathbf{Z})]_{kl}) T_{ik} T_{jl} . \tag{10}$$

Now we conclude by using that the RHS of this equation is equivalent to the minimization in $\mathbf{Z}$ of $\mathrm{GW}_L(\mathbf{C}_X(\mathbf{X}), \mathbf{C}_Z(\mathbf{Z}), \frac{1}{N}\mathbf{1}_N, \frac{1}{N}\mathbf{1}_N)$ (both problems only differ from a constant scaling factor $N^2$). $\square$

As a consequence we have the following result.

**Proposition B.2.** *Let $\Omega \subseteq \mathbb{R}$ and $\mathrm{Im}(\mathbf{C}_X) \subseteq \Omega^{N \times N}$. The minimum eq. (DR) is equal to $\min_{\mathbf{Z}} \mathrm{GW}_L(\mathbf{C}_X(\mathbf{X}), \mathbf{C}_Z(\mathbf{Z}), \frac{1}{N}\mathbf{1}_N, \frac{1}{N}\mathbf{1}_N)$ when:*

*(i) $L(a, \cdot)$ is convex for any $a \in \Omega$ and the image of $\mathbf{C}_Z$ is stable by doubly-stochastic matrices, i.e. ,*

$$\forall \mathbf{Z} \in \mathbb{R}^{N \times d}, \forall \mathbf{T} \in \mathrm{DS}, \exists \mathbf{Y} \in \mathbb{R}^{N \times d}, \ \mathbf{C}_Z(\mathbf{Y}) = \mathbf{T}\mathbf{C}_Z(\mathbf{Z})\mathbf{T}^\top. \tag{11}$$

*(ii) $L(a, \cdot)$ is convex and non-decreasing for any $a \in \Omega$ and*

$$\forall \mathbf{Z} \in \mathbb{R}^{N \times d}, \forall \mathbf{T} \in \mathrm{DS}, \exists \mathbf{Y} \in \mathbb{R}^{N \times d}, \ \mathbf{C}_Z(\mathbf{Y}) \leq \mathbf{T}\mathbf{C}_Z(\mathbf{Z})\mathbf{T}^\top, \tag{12}$$

*where $\leq$ is understood element-wise, i.e. , $\mathbf{A} \leq \mathbf{B} \iff \forall (i,j), \ A_{ij} \leq B_{ij}$.*

*Proof.* For the first point it suffices to see that the condition eq. (11) implies that $\{\mathbf{T}\mathbf{C}_Z(\mathbf{Z})\mathbf{T}^\top : \mathbf{Z} \in \mathbb{R}^{N \times d}, \mathbf{T} \in \mathrm{DS}\} \subseteq \{\mathbf{C}_Z(\mathbf{Z}) : \mathbf{Z} \in \mathbb{R}^{N \times d}\}$ (in fact we have equality by choosing $\mathbf{T} = \mathbf{I}_N$) and thus eq. (6) holds and we apply proposition B.1.

For the second point we will also show that eq. (6) holds. Consider $\mathbf{Z}^\star, \mathbf{T}^\star$ minimizers of $\min_{\mathbf{Z}, \mathbf{T} \in \mathrm{DS}} \sum_{ij} L([\mathbf{C}_X(\mathbf{X})]_{ij}, [\mathbf{T}\mathbf{C}_Z(\mathbf{Z})\mathbf{T}^\top]_{ij})$. By hypothesis there exists $\mathbf{Y} \in \mathbb{R}^{N \times d}$ such that $\forall (i,j) \in [\![N]\!]^2, [\mathbf{T}^\star \mathbf{C}_Z(\mathbf{Z}^\star)\mathbf{T}^{\star\top}]_{ij} \geq [\mathbf{C}_Z(\mathbf{Y})]_{ij}$. Since $L([\mathbf{C}_X(\mathbf{X})]_{ij}, \cdot)$ is non-decreasing for any $(i,j)$ then $\sum_{ij} L([\mathbf{C}_X(\mathbf{X})]_{ij}, [\mathbf{T}^\star \mathbf{C}_Z(\mathbf{Z}^\star)\mathbf{T}^{\star\top}]_{ij}) \geq \sum_{ij} L([\mathbf{C}_X(\mathbf{X})]_{ij}, [\mathbf{C}_Z(\mathbf{Y})]_{ij})$ and thus $\sum_{ij} L([\mathbf{C}_X(\mathbf{X})]_{ij}, [\mathbf{T}^\star \mathbf{C}_Z(\mathbf{Z}^\star)\mathbf{T}^{\star\top}]_{ij}) \geq \min_{\mathbf{Z}} \sum_{ij} L([\mathbf{C}_X(\mathbf{X})]_{ij}, [\mathbf{C}_Z(\mathbf{Z})]_{ij})$ which gives the condition eq. (6) and we have the conclusion by proposition B.1. $\square$

We recall that a function $R : \mathbb{R}^{N \times d} \to \mathbb{R}$ is called permutation invariant if $R(\mathbf{P}\mathbf{Z}) = R(\mathbf{Z})$ for any $\mathbf{Z} \in \mathbb{R}^{N \times d}$ and $N \times N$ permutation matrix $\mathbf{P}$. From the previous results we have the following corollary, which proves, in particular, the first point of theorem 3.2.

**Corollary B.3.** *We have the following equivalences:*

*(i) Consider the spectral methods which correspond to $\mathbf{C}_Z(\mathbf{Z}) = \mathbf{Z}\mathbf{Z}^\top$ and $L = L_2$. Then for any $\mathbf{C}_X$*

$$\min_{\mathbf{Z} \in \mathbb{R}^{N \times d}} \sum_{ij} L_2([\mathbf{C}_X(\mathbf{X})]_{ij}, \langle \mathbf{z}_i, \mathbf{z}_j \rangle) \tag{13}$$

*and*

$$\min_{\mathbf{Z} \in \mathbb{R}^{N \times d}} \mathrm{GW}_{L_2}(\mathbf{C}_X(\mathbf{X}), \mathbf{Z}\mathbf{Z}^\top, \frac{1}{N}\mathbf{1}_N, \frac{1}{N}\mathbf{1}_N) \tag{14}$$

*are equal.*

*(ii) Consider the cross-entropy loss $L(x,y) = x \log(x/y)$ and $\mathbf{C}_X$ such that $\mathrm{Im}(\mathbf{C}_X) \subseteq \mathbb{R}_+^{N \times N}$. Suppose that the similarity in the output space can be written as*

$$\forall (i,j) \in [\![N]\!]^2, [\mathbf{C}_Z(\mathbf{Z})]_{ij} = f(\mathbf{z}_i - \mathbf{z}_j)/R(\mathbf{Z}), \tag{15}$$

*for some logarithmically concave function $f : \mathbb{R}^d \to \mathbb{R}_+$ and normalizing factor $R : \mathbb{R}^{N \times d} \to \mathbb{R}_+^*$ which is both convex and permutation invariant. Then,*

$$\min_{\mathbf{Z} \in \mathbb{R}^{N \times d}} \sum_{ij} L([\mathbf{C}_X(\mathbf{X})]_{ij}, [\mathbf{C}_Z(\mathbf{Z})]_{ij}) \tag{16}$$

*and*

$$\min_{\mathbf{Z} \in \mathbb{R}^{N \times d}} \mathrm{GW}_L(\mathbf{C}_X(\mathbf{X}), \mathbf{C}_Z(\mathbf{Z}), \frac{1}{N}\mathbf{1}_N, \frac{1}{N}\mathbf{1}_N) \tag{17}$$

*are equal.*

*Proof.* For the first point we show that the condition eq. (11) of proposition B.2 is satisfied. Indeed take any $\mathbf{Z}, \mathbf{T}$ then $\mathbf{T}\mathbf{C}_Z(\mathbf{Z})\mathbf{T}^\top = \mathbf{T}\mathbf{Z}\mathbf{Z}^\top\mathbf{T}^\top = (\mathbf{T}\mathbf{Z})(\mathbf{T}\mathbf{Z})^\top = \mathbf{C}_Z(\mathbf{T}\mathbf{Z})$.

For the second point if we consider $\tilde{L}(a,b) = a \times b$ then we use that the neighbor embedding problem eq. (16) is equivalent to $\min_{\mathbf{Z}\in\mathbb{R}^{N\times d}} \sum_{ij} -[\mathbf{C}_X(\mathbf{X})]_{ij}\log([\mathbf{C}_Z(\mathbf{Z})]_{ij}) = \min_{\mathbf{Z}\in\mathbb{R}^{N\times d}} \sum_{ij} \tilde{L}([\mathbf{C}_X(\mathbf{X})]_{ij}, [\widetilde{\mathbf{C}_Z}(\mathbf{Z})]_{ij})$ where $[\widetilde{\mathbf{C}_Z}(\mathbf{Z})]_{ij} = g(\mathbf{z}_i - \mathbf{z}_j) + \log(R(\mathbf{Z}))$ with $g = -\log \circ f$. Since $f$ is logarithmically concave $g$ is convex. Moreover we have that $\tilde{L}(a, \cdot)$ is convex (it is linear) and non-decreasing since $a \in \mathbb{R}_+$ in this case ($\mathbf{C}_X(\mathbf{X})$ is non-negative). Also for any $\mathbf{Z} \in \mathbb{R}^{N\times d}$ and $\mathbf{T} \in \text{DS}$ we have, using Jensen's inequality since $\mathbf{T}$ is doubly-stochastic,

$$
\begin{aligned}
[\mathbf{T}\widetilde{\mathbf{C}_Z}(\mathbf{Z})\mathbf{T}^\top]_{ij} &= \sum_{kl} g(\mathbf{z}_k - \mathbf{z}_l)T_{ik}T_{jl} + \log(R(\mathbf{Z})) \\
&\geq g(\sum_{kl}(\mathbf{z}_k - \mathbf{z}_l)T_{ik}T_{jl}) + \log(R(\mathbf{Z})) \\
&= g(\sum_k \mathbf{z}_k T_{ik} - \sum_l \mathbf{z}_l T_{jl}) + \log(R(\mathbf{Z})).
\end{aligned}
\tag{18}
$$

Now we will prove that $\log(R(\mathbf{Z})) \geq \log(R(\mathbf{T}\mathbf{Z}))$. Using Birkhoff's theorem (Birkhoff, 1946) the matrix $\mathbf{T}$ can be decomposed as a convex combination of permutation matrices, *i.e.* , $\mathbf{T} = \sum_k \lambda_k \mathbf{P}_k$ where $(\mathbf{P}_k)_k$ are permutation matrices and $\lambda_k \in \mathbb{R}_+$ with $\sum_k \lambda_k = 1$. Hence by convexity and Jensen's inequality $R(\mathbf{T}\mathbf{Z}) = R(\sum_k \lambda_k \mathbf{P}_k\mathbf{Z}) \leq \sum_k \lambda_k R(\mathbf{P}_k\mathbf{Z})$. Now using that $R$ is permutation invariant we get $R(\mathbf{P}_k\mathbf{Z}) = R(\mathbf{Z})$ and thus $R(\mathbf{T}\mathbf{Z}) \leq \sum_k \lambda_k R(\mathbf{Z}) = R(\mathbf{Z})$. Since the logarithm is non-decreasing we have $\log(R(\mathbf{Z})) \geq \log(R(\mathbf{T}\mathbf{Z}))$ and, overall,

$$
[\mathbf{T}\widetilde{\mathbf{C}_Z}(\mathbf{Z})\mathbf{T}^\top]_{ij} \geq g(\sum_k \mathbf{z}_k T_{ik} - \sum_l \mathbf{z}_l T_{jl}) + \log(R(\mathbf{T}\mathbf{Z})) = [\widetilde{\mathbf{C}_Z}(\mathbf{T}\mathbf{Z})]_{ij}.
\tag{19}
$$

Thus if we introduce $\mathbf{Y} = \mathbf{T}\mathbf{Z}$ we have $[\mathbf{T}\widetilde{\mathbf{C}_Z}(\mathbf{Z})\mathbf{T}^\top]_{ij} \geq [\widetilde{\mathbf{C}_Z}(\mathbf{Y})]_{ij}$ and eq. (12) is satisfied. Thus we can apply proposition B.2 and state that $\min_{\mathbf{Z}\in\mathbb{R}^{N\times d}} \sum_{ij} \tilde{L}([\mathbf{C}_X(\mathbf{X})]_{ij}, [\widetilde{\mathbf{C}_Z}(\mathbf{Z})]_{ij})$ and $\min_{\mathbf{Z}\in\mathbb{R}^{N\times d}} \text{GW}_{\tilde{L}}(\mathbf{C}_X(\mathbf{X}), \widetilde{\mathbf{C}_Z}(\mathbf{Z}), \frac{1}{N}\mathbf{1}_N, \frac{1}{N}\mathbf{1}_N)$ are equivalent which concludes as $\min_{\mathbf{Z}\in\mathbb{R}^{N\times d}} \text{GW}_{\tilde{L}}(\mathbf{C}_X(\mathbf{X}), \widetilde{\mathbf{C}_Z}(\mathbf{Z}), \frac{1}{N}\mathbf{1}_N, \frac{1}{N}\mathbf{1}_N) = \min_{\mathbf{Z}\in\mathbb{R}^{N\times d}} \text{GW}_L(\mathbf{C}_X(\mathbf{X}), \mathbf{C}_Z(\mathbf{Z}), \frac{1}{N}\mathbf{1}_N, \frac{1}{N}\mathbf{1}_N)$. $\square$

It remains to prove the second point of theorem 3.2 as stated below.

**Proposition B.4.** *Consider* $\text{Im}(\mathbf{C}_X) \subseteq \mathbb{R}_+^{N\times N}$, $L = L_{\text{KL}}$. *Suppose that for any* $\mathbf{X}$ *the matrix* $\mathbf{C}_X(\mathbf{X})$ *is CPD and for any* $\mathbf{Z}$

$$
\mathbf{C}_Z(\mathbf{Z}) = \text{diag}(\boldsymbol{\alpha}_\mathbf{Z})\mathbf{K}_\mathbf{Z}\,\text{diag}(\boldsymbol{\beta}_\mathbf{Z}),
\tag{20}
$$

*where* $\boldsymbol{\alpha}_\mathbf{Z}, \boldsymbol{\beta}_\mathbf{Z} \in \mathbb{R}_{>0}^N$ *and* $\mathbf{K}_\mathbf{Z} \in \mathbb{R}_{>0}^{N\times N}$ *is such that* $\log(\mathbf{K}_\mathbf{Z})$ *is CPD. Then the minimum eq.* (DR) *is equal to* $\min_\mathbf{Z} \text{GW}_L(\mathbf{C}_X(\mathbf{X}), \mathbf{C}_Z(\mathbf{Z}), \frac{1}{N}\mathbf{1}_N, \frac{1}{N}\mathbf{1}_N)$.

*Proof.* To prove this result we will show that, for any $\mathbf{Z}$, the function

$$
\mathbf{T} \in \mathcal{U}(\frac{1}{N}\mathbf{1}_N, \frac{1}{N}\mathbf{1}_N) \to E_L(\mathbf{C}_X(\mathbf{X}), \mathbf{C}_Z(\mathbf{Z}), \mathbf{T}),
\tag{21}
$$

is actually concave. Indeed, in this case there exists a minimizer which is an extremal point of $\mathcal{U}(\frac{1}{N}\mathbf{1}_N, \frac{1}{N}\mathbf{1}_N)$. By Birkhoff's theorem (Birkhoff, 1946) these extreme points are the matrices $\frac{1}{N}\mathbf{P}$ where $\mathbf{P}$ is a $N \times N$ permutation matrix. Consequently, when the function eq. (21) is concave minimizing $\text{GW}_L(\mathbf{C}_X(\mathbf{X}), \mathbf{C}_Z(\mathbf{Z}), \frac{1}{N}\mathbf{1}_N, \frac{1}{N}\mathbf{1}_N)$ in $\mathbf{Z}$ is equivalent to minimizing in $\mathbf{Z}$

$$
\min_{\mathbf{P}\in\mathbb{R}^{N\times N} \text{ permutation}} \sum_{ijkl} L([\mathbf{C}_X(\mathbf{X})]_{ik}, [\mathbf{C}_Z(\mathbf{Z})]_{jl})P_{ij}P_{kl} = \min_{\sigma\in S_N} \sum_{ij} L([\mathbf{C}_X(\mathbf{X})]_{ij}, [\mathbf{C}_Z(\mathbf{Z})]_{\sigma(i)\sigma(j)}),
\tag{22}
$$

which is exactly the Gromov-Monge problem described in lemma 3.1 and thus the problem is equivalent to eq. (DR) by lemma 3.1.

First note that $L(x,y) = x\log(x) - x - x\log(y) + y$ so for any $\mathbf{T} \in \mathcal{U}(\frac{1}{N}\mathbf{1}_N, \frac{1}{N}\mathbf{1}_N)$ the loss $E_L(\mathbf{C}_X(\mathbf{X}), \mathbf{C}_Z(\mathbf{Z}), \mathbf{T})$ is equal to

$$
\begin{aligned}
\sum_{ijkl} L([\mathbf{C}_X(\mathbf{X})]_{ik}, [\mathbf{C}_Z(\mathbf{Z})]_{jl}) T_{ij} T_{kl} &= a_{\mathbf{X}} + b_{\mathbf{Z}} - \sum_{ijkl} [\mathbf{C}_X(\mathbf{X})]_{ik} \log([\mathbf{C}_Z(\mathbf{Z})]_{jl}) T_{ij} T_{kl} \\
&= a_{\mathbf{X}} + b_{\mathbf{Z}} - \sum_{ijkl} [\mathbf{C}_X(\mathbf{X})]_{ik} \log([\boldsymbol{\alpha}_{\mathbf{Z}}]_j [\boldsymbol{\beta}_{\mathbf{Z}}]_l [\mathbf{W}]_{jl}) T_{ij} T_{kl} \\
&= a_{\mathbf{X}} + b_{\mathbf{Z}} - \frac{1}{N} \sum_{ijk} [\mathbf{C}_X(\mathbf{X})]_{ik} \log([\boldsymbol{\alpha}_{\mathbf{Z}}]_j) T_{ij} \\
&\quad - \frac{1}{N} \sum_{ikl} [\mathbf{C}_X(\mathbf{X})]_{ik} \log([\boldsymbol{\beta}_{\mathbf{Z}}]_l) T_{kl} \\
&\quad - \sum_{ijkl} [\mathbf{C}_X(\mathbf{X})]_{ik} \log([\mathbf{K}_{\mathbf{Z}}]_{jl}) T_{ij} T_{kl},
\end{aligned}
\tag{23}
$$

where $a_{\mathbf{X}}, b_{\mathbf{Z}}$ are terms that do not depend on $\mathbf{T}$. Since the problem is quadratic the concavity only depends on the term $-\sum_{ijkl}[\mathbf{C}_X(\mathbf{X})]_{ik}\log([\mathbf{K}_{\mathbf{Z}}]_{jl})T_{ij}T_{kl} = -\operatorname{Tr}(\mathbf{C}_X(\mathbf{X})\mathbf{T}^\top \log(\mathbf{K}_{\mathbf{Z}})\mathbf{T})$. From (Maron and Lipman, 2018) we know that the function $\mathbf{T} \to -\operatorname{Tr}(\mathbf{C}_X(\mathbf{X})\mathbf{T}^\top \log(\mathbf{K}_{\mathbf{Z}})\mathbf{T})$ is concave on $\mathcal{U}(\frac{1}{N}\mathbf{1}_N, \frac{1}{N}\mathbf{1}_N)$ when $\mathbf{C}_X(\mathbf{X})$ is CPD and $\log(\mathbf{W}_{\mathbf{Z}})$ is CPD. This concludes the proof.

$\square$

### B.3 Necessary and sufficient condition

We give a necessary and sufficient condition under which the DR problem is equivalent to a GW problem

**Proposition B.5.** *Let* $\mathbf{C}_1 \in \mathbb{R}^{N \times N}, L : \mathbb{R} \times \mathbb{R} \to \mathbb{R}$ *and* $\mathcal{C} \subseteq \mathbb{R}^{N \times N}$ *a subspace of* $N \times N$ *matrices. We suppose that* $\mathcal{C}$ *is stable by permutations i.e. ,* $\mathbf{C} \in \mathcal{C}$ *implies that* $\mathbf{PCP}^\top \in \mathcal{C}$ *for any* $N \times N$ *permutation matrix* $\mathbf{P}$*. Then*

$$
\min_{\mathbf{C}_2 \in \mathcal{C}} \sum_{(i,j) \in [\![N]\!]^2} L([\mathbf{C}_1]_{ij}, [\mathbf{C}_2]_{ij}) = \min_{\mathbf{C}_2 \in \mathcal{C}} \min_{\substack{\mathbf{T} \in \mathbb{R}_+^{N \times N} \\ \mathbf{T}\mathbf{1}_N = \mathbf{1}_N \\ \mathbf{T}^\top \mathbf{1}_N = \mathbf{1}_N}} \sum_{ijkl} L([\mathbf{C}_1]_{ij}, [\mathbf{C}_2]_{kl}) \, T_{ik} T_{jl}
\tag{24}
$$

*if and only if the optimal assignment problem*

$$
\min_{\sigma_1, \sigma_2 \in S_N} f(\sigma_1, \sigma_2) := \min_{\mathbf{C}_2 \in \mathcal{C}} \sum_{ij} L([\mathbf{C}_1]_{ij}, [\mathbf{C}_2]_{\sigma_1(i)\sigma_2(j)})
\tag{25}
$$

*admits an optimal solution* $(\sigma_1^\star, \sigma_2^\star)$ *with* $\sigma_1^\star = \sigma_2^\star$*.*

*Proof.* We note that the LHS of eq. (24) is always smaller than the RHS since $\mathbf{T} = \mathbf{I}_N$ is admissible for the RHS problem. So both problems are equal if and only if

$$
\min_{\mathbf{C}_2 \in \mathcal{C}} \sum_{(i,j) \in [\![N]\!]^2} L([\mathbf{C}_1]_{ij}, [\mathbf{C}_2]_{ij}) \le \min_{\mathbf{C}_2 \in \mathcal{C}} \min_{\substack{\mathbf{T} \in \mathbb{R}_+^{N \times N} \\ \mathbf{T}\mathbf{1}_N = \mathbf{1}_N \\ \mathbf{T}^\top \mathbf{1}_N = \mathbf{1}_N}} \sum_{ijkl} L([\mathbf{C}_1]_{ij}, [\mathbf{C}_2]_{kl}) \, T_{ik} T_{jl}.
\tag{26}
$$

Now consider any $\mathbf{C}_2$ fixed and observe that

$$
\min_{\mathbf{T} \in \mathrm{DS}} \sum_{ijkl} L([\mathbf{C}_1]_{ij}, [\mathbf{C}_2]_{kl}) \, T_{ik} T_{jl} \ge \min_{\mathbf{T}^{(1)}, \mathbf{T}^{(2)} \in \mathrm{DS}} \sum_{ijkl} L([\mathbf{C}_1]_{ij}, [\mathbf{C}_2]_{kl}) \, T_{ik}^{(1)} T_{jl}^{(2)}.
\tag{27}
$$

The latter problem is a co-optimal transport problem (Redko et al., 2020), and, since it is a bilinear problem, there is an optimal solution $(\mathbf{T}^{(1)}, \mathbf{T}^{(2)})$ such that both $\mathbf{T}^{(1)}$ and $\mathbf{T}^{(2)}$ are in an extremal point of DS which is the space of $N \times N$ permutation matrices by Birkhoff's theorem (Birkhoff, 1946). This point was already

noted by Konno (1976) but we recall the proof for completeness. We note $L_{ijkl} = L([\mathbf{C}_1]_{ij}, [\mathbf{C}_2]_{kl})$ and consider an optimal solution $(\mathbf{T}_\star^{(1)}, \mathbf{T}_\star^{(2)})$ of $\min_{\mathbf{T}^{(1)}, \mathbf{T}^{(2)} \in \mathrm{DS}} \phi(\mathbf{T}^{(1)}, \mathbf{T}^{(2)}) := \sum_{ijkl} L_{ijkl} T_{ik}^{(1)} T_{jl}^{(2)}$. Consider the linear problem $\min_{\mathbf{T} \in \mathrm{DS}} \phi(\mathbf{T}, \mathbf{T}_\star^{(2)})$. Since it is a linear over the space of doubly stochastic matrices it admits a permutation matrix $\mathbf{P}^{(1)}$ as optimal solution. Also $\phi(\mathbf{P}^{(1)}, \mathbf{T}_\star^{(2)}) \leq \phi(\mathbf{T}_\star^{(1)}, \mathbf{T}_\star^{(2)})$ by optimality. Now consider the linear problem $\min_{\mathbf{T} \in \mathrm{DS}} \phi(\mathbf{P}^{(1)}, \mathbf{T})$, in the same it admits a permutation matrix $\mathbf{P}^{(2)}$ as optimal solution and by optimality $\phi(\mathbf{P}^{(1)}, \mathbf{P}^{(2)}) \leq \phi(\mathbf{P}^{(1)}, \mathbf{T}_\star^{(2)})$ thus $\phi(\mathbf{P}^{(1)}, \mathbf{P}^{(2)}) \leq \phi(\mathbf{T}_\star^{(1)}, \mathbf{T}_\star^{(2)})$ which implies that $(\mathbf{P}^{(1)}, \mathbf{P}^{(2)})$ is an optimal solution. Combining with eq. (27) we get

$$
\begin{aligned}
\min_{\mathbf{C}_2 \in \mathcal{C}} \min_{\mathbf{T} \in \mathrm{DS}} \sum_{ijkl} L([\mathbf{C}_1]_{ij}, [\mathbf{C}_2]_{kl}) \, T_{ik} T_{jl} &\geq \min_{\mathbf{C}_2 \in \mathcal{C}} \min_{\sigma_1, \sigma_2 \in S_N} \sum_{ij} L([\mathbf{C}_1]_{ij}, [\mathbf{C}_2]_{\sigma_1(i)\sigma_2(j)}) \\
&= \min_{\sigma_1, \sigma_2 \in S_N} f(\sigma_1, \sigma_2).
\end{aligned}
\tag{28}
$$

Now suppose that the optimal assignment problem $\min_{\sigma_1, \sigma_2 \in S_N} f(\sigma_1, \sigma_2)$ admits an optimal solution $(\sigma_1^\star, \sigma_2^\star)$ with $\sigma_1^\star = \sigma_2^\star$. Then with eq. (31)

$$
\begin{aligned}
\min_{\mathbf{C}_2 \in \mathcal{C}} \min_{\mathbf{T} \in \mathrm{DS}} \sum_{ijkl} L([\mathbf{C}_1]_{ij}, [\mathbf{C}_2]_{kl}) \, T_{ik} T_{jl} &\geq \min_{\mathbf{C}_2 \in \mathcal{C}} \sum_{ij} L([\mathbf{C}_1]_{ij}, [\mathbf{C}_2]_{\sigma_1^\star(i)\sigma_1^\star(j)}) \\
&\geq \min_{\sigma \in S_N} \min_{\mathbf{C}_2 \in \mathcal{C}} \sum_{ij} L([\mathbf{C}_1]_{ij}, [\mathbf{C}_2]_{\sigma(i)\sigma(j)}).
\end{aligned}
\tag{29}
$$

Now since $\mathcal{C}$ is stable by permutation then $\{([\mathbf{C}]_{\sigma(i)\sigma(j)})_{(i,j) \in [\![N]\!]^2} : \mathbf{C} \in \mathcal{C}, \sigma \in S_N\} = \mathcal{C}$ and consequently $\min_{\sigma \in S_N} \min_{\mathbf{C}_2 \in \mathcal{C}} \sum_{ij} L([\mathbf{C}_1]_{ij}, [\mathbf{C}_2]_{\sigma(i)\sigma(j)}) = \min_{\mathbf{C}_2 \in \mathcal{C}} \sum_{ij} L([\mathbf{C}_1]_{ij}, [\mathbf{C}_2]_{ij})$. Consequently, using eq. (29),

$$
\min_{\mathbf{C}_2 \in \mathcal{C}} \min_{\mathbf{T} \in \mathrm{DS}} \sum_{ijkl} L([\mathbf{C}_1]_{ij}, [\mathbf{C}_2]_{kl}) \, T_{ik} T_{jl} \geq \min_{\mathbf{C}_2 \in \mathcal{C}} \sum_{ij} L([\mathbf{C}_1]_{ij}, [\mathbf{C}_2]_{ij}),
\tag{30}
$$

and thus both are equal.

Conversely suppose that eq. (24) holds. Then, from eq. (31) we have

$$
\begin{aligned}
\min_{\sigma_1, \sigma_2 \in S_N} f(\sigma_1, \sigma_2) &= \min_{\sigma_1, \sigma_2 \in S_N} \min_{\mathbf{C}_2 \in \mathcal{C}} \sum_{ij} L([\mathbf{C}_1]_{ij}, [\mathbf{C}_2]_{\sigma_1(i)\sigma_2(j)}) \\
&\leq \min_{\mathbf{C}_2 \in \mathcal{C}} \min_{\mathbf{T} \in \mathrm{DS}} \sum_{ijkl} L([\mathbf{C}_1]_{ij}, [\mathbf{C}_2]_{kl}) \, T_{ik} T_{jl} \\
&= \min_{\mathbf{C}_2 \in \mathcal{C}} \sum_{(i,j) \in [\![N]\!]^2} L([\mathbf{C}_1]_{ij}, [\mathbf{C}_2]_{ij}) = f(\mathrm{id}, \mathrm{id}),
\end{aligned}
\tag{31}
$$

which concludes the proof. $\qquad\square$

*Remark* B.6. The condition on the set of similarity matrices $\mathcal{C}$ is quite reasonable: it indicates that if $\mathbf{C}$ is an admissible similarity matrix, then permuting the rows and columns of $\mathbf{C}$ results in another admissible similarity matrix. For DR, the corresponding $\mathcal{C}$ is $\mathcal{C} = \{\mathbf{C}_Z(\mathbf{Z}) : \mathbf{Z} \in \mathbb{R}^{N \times d}\}$. In this case, if $\mathbf{C}_Z(\mathbf{Z})$ is of the form

$$
[\mathbf{C}_Z(\mathbf{Z})]_{ij} = h(f(\mathbf{z}_i, \mathbf{z}_j), g(\mathbf{Z})),
\tag{32}
$$

where $f : \mathbb{R}^d \times \mathbb{R}^d \to \mathbb{R}$, $h : \mathbb{R} \times \mathbb{R} \to \mathbb{R}$ and $g : \mathbb{R}^{N \times d} \to \mathbb{R}$ which is permutation invariant (Bronstein et al., 2021), then $\mathcal{C}$ is stable under permutation. Indeed, permuting the rows and columns of $\mathbf{C}_Z(\mathbf{Z})$ by $\sigma$ is equivalent to considering the similarity $\mathbf{C}_Z(\mathbf{Y})$, where $\mathbf{Y} = (\mathbf{z}_{\sigma(1)}, \cdots, \mathbf{z}_{\sigma(n)})^\top$. Moreover, most similarities in the target space considered in DR take the form eq. (32): $\langle \Phi(\mathbf{z}_i), \Phi(\mathbf{z}_j) \rangle_{\mathcal{H}}$ (kernels such as in spectral methods with $\Phi = \mathrm{id}$), $f(\mathbf{z}_i, \mathbf{z}_j)/\sum_{nm} f(\mathbf{z}_n, \mathbf{z}_m)$ (normalized similarities such as in SNE and t-SNE). Also note that the condition on $\mathcal{C} = \{\mathbf{C}_Z(\mathbf{Z}) : \mathbf{Z} \in \mathbb{R}^{N \times d}\}$ of proposition B.5 is met as soon as $\mathbf{C}_Z : \mathbb{R}^{N \times d} \to \mathbb{R}^{N \times N}$ is permutation equivariant.

## C  Generalized Semi-relaxed Gromov-Wasserstein is a divergence

*Remark* C.1 (Weak isomorphism). According to the notion of weak isomorphism in Chowdhury and Mémoli (2019), for a graph $(\mathbf{C}, \mathbf{h})$ with corresponding discrete measure $\mu = \sum_i h_i \delta_{x_i}$, two nodes $x_i$ and $x_j$ are the "same" if they have the same internal perception *i.e* $C_{ii} = C_{jj} = C_{ij} = C_{ji}$ and external perception $\forall k \neq (i,j), C_{ik} = C_{jk}, C_{ki} = C_{kj}$. So two graphs $(\mathbf{C}_1, \mathbf{h}_1)$ and $(\mathbf{C}_2, \mathbf{h}_2)$ are said to be weakly isomorphic, if there exist a canonical representation $(\mathbf{C}_c, \mathbf{h}_c)$ such that $\mathrm{card}(\mathrm{supp}(\mathbf{h}_c)) = p \leq n, m$ and $M_1 \in \{0,1\}^{n \times p}$ (resp. $M_2 \in \{0,1\}^{m \times p}$) such that for $k \in \{1,2\}$

$$\mathbf{C}_c = \mathbf{M}_k^\top \mathbf{C}_c \mathbf{M}_k \quad \text{and} \quad \mathbf{h}_c = \mathbf{M}_k^\top \mathbf{h}_k \tag{33}$$

We first emphasize a simple result extending a proof in Vincent-Cuaz et al. (2022a).

**Proposition C.2.** *Let any divergence $L : \Omega \times \Omega \to \mathbb{R}_+$ for $\Omega \subseteq \mathbb{R}$, then for any $(\mathbf{C}, \mathbf{h})$ and $(\overline{\mathbf{C}}, \overline{\mathbf{h}})$, we have $\mathrm{GW}_L(\mathbf{C}, \overline{\mathbf{C}}, \mathbf{h}, \overline{\mathbf{h}}) = 0$ if and only if $\mathrm{GW}_{L_2}(\mathbf{C}, \overline{\mathbf{C}}, \mathbf{h}, \overline{\mathbf{h}}) = 0$.*

*Proof.* If $\mathrm{GW}_L(\mathbf{C}, \overline{\mathbf{C}}, \mathbf{h}, \overline{\mathbf{h}}) = 0$, then there exists $\mathbf{T} \in \mathcal{U}(\mathbf{h}, \overline{\mathbf{h}})$ such that

$$E_L(\mathbf{C}, \overline{\mathbf{C}}, \mathbf{T}) = \sum_{ijkl} L(C_{ij}, \overline{C}_{kl}) T_{ik} T_{jl} = 0 \tag{34}$$

so whenever $T_{ik} T_{jl} \neq 0$, we must have $L(C_{ij}, \overline{C}_{kl}) = 0$ *i.e* $C_{ij} = \overline{C}_{kl}$ as $L$ is a divergence. Which implies that $E_{L'}(\mathbf{C}, \overline{\mathbf{C}}, \mathbf{T}) = 0$ for any other divergence $L'$ well defined on any domain $\Omega \times \Omega$, necessarily including $L_2$. □

**Lemma C.3.** *Let any divergence $L : \Omega \times \Omega \to \mathbb{R}_+$ for $\Omega \subseteq \mathbb{R}$. Let $(\mathbf{C}, \mathbf{h}) \in \Omega^{n \times n} \times \Sigma_n$ and $(\overline{\mathbf{C}}, \overline{\mathbf{h}}) \in \mathbb{R}^{m \times m} \times \Sigma_m$. Then $\mathrm{srGW}_L(\mathbf{C}, \overline{\mathbf{C}}, \mathbf{h}, \overline{\mathbf{h}}) = 0$ if and only if there exists $\overline{\mathbf{h}} \in \Sigma_m$ such that $(\mathbf{C}, \mathbf{h})$ and $(\overline{\mathbf{C}}, \overline{\mathbf{h}})$ are weakly isomorphic.*

*Proof.* ($\Rightarrow$) As $\mathrm{srGW}_L(\mathbf{C}, \overline{\mathbf{C}}, \mathbf{h}, \overline{\mathbf{h}}) = 0$ there exists $\mathbf{T} \in \mathcal{U}(\mathbf{h}, \overline{\mathbf{h}})$ such that $E_L(\mathbf{C}, \overline{\mathbf{C}}, \mathbf{T}) = 0$ hence $\mathrm{GW}_L(\mathbf{C}, \overline{\mathbf{C}}, \mathbf{h}, \overline{\mathbf{h}}) = 0$. Using proposition C.2, $\mathrm{GW}_{L_2} = 0$ hence using Theorem 18 in Chowdhury and Mémoli (2019), it implies that $(\mathbf{C}, \mathbf{h})$ and $(\overline{\mathbf{C}}, \overline{\mathbf{h}})$ are weakly isomorphic.

($\Leftarrow$) As mentioned, $(\mathbf{C}, \mathbf{h})$ and $(\overline{\mathbf{C}}, \overline{\mathbf{h}})$ being weakly isomorphic implies that $\mathrm{GW}_{L_2} = 0$. So there exists $\mathbf{T} \in \mathcal{U}(\mathbf{h}, \overline{\mathbf{h}})$, such that $E_L(\mathbf{C}, \overline{\mathbf{C}}, \mathbf{T}) = 0$. Moreover $\mathbf{T}$ is admissible for the srGW problem as $\mathbf{T} \in \mathcal{U}(\mathbf{h}, \overline{\mathbf{h}}) \subset \mathcal{U}_n(\mathbf{h})$, thus $\mathrm{srGW}_L(\mathbf{C}, \overline{\mathbf{C}}, \mathbf{h}, \overline{\mathbf{h}}) = 0$. □

### C.1  About trivial solutions of semi-relaxed GW when $L$ is not a proper divergence

We briefly describe here some trivial solutions of $\mathrm{srGW}_L$ when $L$ is not a proper divergence. We recall that

$$\mathrm{srGW}_L(\mathbf{C}, \overline{\mathbf{C}}, \mathbf{h}) = \min_{\mathbf{T} \in \mathbb{R}_+^{N \times n} : \mathbf{T}\mathbf{1}_n = \mathbf{h}} E_L(\mathbf{C}, \overline{\mathbf{C}}, \mathbf{T}) = \sum_{ijkl} L(C_{ij}, \overline{C}_{kl}) T_{ik} T_{jl} . \tag{35}$$

Suppose that $\mathbf{h} \in \Sigma_N^*$ and that $\overline{\mathbf{C}}$ has a minimum value on its diagonal *i.e.* $\min_{(i,j)} \overline{C}_{ij} = \min_{ii} \overline{C}_{ii}$. Suppose also that $\forall a, L(a, \cdot)$ is both convex *and* non-decreasing. First we have $\sum_j \frac{T_{ij}}{h_i} = 1$ for any $i \in [\![N]\!]$. Hence

using the convexity of $L$, Jensen inequality and the fact that $L(a, \cdot)$ is non-decreasing for any $a$

$$
\begin{aligned}
\sum_{ijkl} L(C_{ij}, \overline{C}_{kl}) T_{ik} T_{jl} &= \sum_{ijkl} L(C_{ij}, \overline{C}_{kl}) h_i h_j \frac{T_{ik}}{h_i} \frac{T_{jl}}{h_j} \\
&\geq \sum_{ij} L(C_{ij}, \sum_{kl} \overline{C}_{kl} \frac{T_{ik}}{h_i} \frac{T_{jl}}{h_j}) h_i h_j \\
&\geq \sum_{ij} L(C_{ij}, \sum_{kl} (\min_{nm} \overline{C}_{nm}) \frac{T_{ik}}{h_i} \frac{T_{jl}}{h_j}) h_i h_j \\
&= \sum_{ij} L(C_{ij}, (\min_{nm} \overline{C}_{nm})) h_i h_j \\
&= \sum_{ij} L(C_{ij}, (\min_{nn} \overline{C}_{nn})) h_i h_j .
\end{aligned}
\tag{36}
$$

Now suppose without loss of generality that $\min_{ii} \overline{C}_{ii} = \overline{C}_{11}$ then this gives

$$
\min_{\mathbf{T} \in \mathbb{R}_+^{N \times n}: \mathbf{T}\mathbf{1}_n = \mathbf{h}} \sum_{ijkl} L(C_{ij}, \overline{C}_{kl}) T_{ik} T_{jl} \geq \sum_{ij} L(C_{ij}, \overline{C}_{11}) h_i h_j .
\tag{37}
$$

Now consider the coupling $\mathbf{T}^\star = \begin{pmatrix} h_1 & 0 & 0 & 0 \\ h_2 & 0 & 0 & 0 \\ \vdots & \vdots & \vdots & \vdots \\ h_N & 0 & 0 & 0 \end{pmatrix}$. It is admissible and satisfies

$$
E_L(\mathbf{C}, \overline{\mathbf{C}}, \mathbf{T}^\star) = \sum_{ij} L(C_{ij}, \overline{C}_{11}) h_i h_j \leq \min_{\mathbf{T} \in \mathbb{R}_+^{N \times n}: \mathbf{T}\mathbf{1}_n = \mathbf{h}} E_L(\mathbf{C}, \overline{\mathbf{C}}, \mathbf{T}) .
\tag{38}
$$

Consequently the coupling $\mathbf{T}^\star$ is optimal. However the solution given by this coupling is trivial: it consists in sending all the mass to one unique point. In another words, all the nodes in the input graph are sent to a unique node in the target graph. Note that this phenomena is impossible for standard GW because of the coupling constraints.

We emphasize that this hypothesis on $L$ *cannot be satisfied* as soon as $L$ is a proper divergence. Indeed when $L$ is a divergence the constraint "$L(a, \cdot)$ is non-decreasing for any $a$" is not possible as it would break the divergence constraints $\forall a, b \; L(a, b) \geq 0$ and $L(a, b) = 0 \iff a = b$ (at some point $L$ must be decreasing).

## D    Clustering properties: Proof of theorem 4.1

We recall that a matrix $\mathbf{C} \in \mathbb{R}^{N \times N}$ is conditionally positive definite (CPD), *resp.* negative definite (CND), if $\forall \mathbf{x} \in \mathbb{R}^N, \mathbf{x}^\top \mathbf{1}_N = 0$ s.t. $\mathbf{x}^\top \mathbf{C} \mathbf{x} \geq 0$, *resp.* $\leq 0$. We also consider the Hadamard product of matrices as $\mathbf{A} \odot \mathbf{B} = (A_{ij} \times B_{ij})_{ij}$. The $i$-th column of a matrix $\mathbf{T}$ is the vector denoted by $\mathbf{T}_{:,i}$. For a vector $\mathbf{x} \in \mathbb{R}^n$ we denote by $\text{diag}(\mathbf{x})$ the diagonal $n \times n$ matrix whose elements are the $x_i$.

We state below the theorem that we prove in this section.

**Theorem 4.1.** *Let $\mathbf{h}_X \in \Sigma_N$ and $L = L_2$. Suppose that for any $\mathbf{X} \in \mathbb{R}^{N \times p}$ the matrix $\mathbf{C}_X(\mathbf{X})$ is CPD or CND. Then the problem eq. (srGWB) admits a membership matrix as optimal coupling, i.e. , there is a minimizer of $\mathbf{T} \in \mathcal{U}_n(\mathbf{h}_X) \to \min_{\overline{\mathbf{C}} \in \mathbb{R}^{n \times n}} E_L(\mathbf{C}_X(\mathbf{X}), \overline{\mathbf{C}}, \mathbf{T})$ with only one non-zero value per row.*

In order to prove this result we introduce the space of semi-relaxed couplings whose columns are not zero

$$
\mathcal{U}_n^+(\mathbf{h}_X) = \{ \mathbf{T} \in \mathcal{U}_n(\mathbf{h}_X) : \forall i \in [\![n]\!], \mathbf{T}_{:,i} \neq 0 \} ,
\tag{39}
$$

and we will use the following lemma.

**Lemma D.1.** *Let $\mathbf{h}_X \in \Sigma_N$, $L = L_2$ and $\mathbf{C}_X(\mathbf{X})$ symmetric. For any $\mathbf{T} \in \mathcal{U}_n(\mathbf{h}_X)$, the matrix $\overline{\mathbf{C}}(\mathbf{T}) \in \mathbb{R}^{n \times n}$ defined by*

$$\overline{\mathbf{C}}(\mathbf{T}) = \begin{cases} \mathbf{T}_{:,i}^\top \mathbf{C}_X(\mathbf{X}) \mathbf{T}_{:,j} / (\mathbf{T}_{:,i}^\top \mathbf{1}_N)(\mathbf{T}_{:,j}^\top \mathbf{1}_N) & \text{for } (i,j) \text{ such that } \mathbf{T}_{:,i} \text{ and } \mathbf{T}_{:,j} \neq 0 \\ 0 & \text{otherwise} \end{cases} \tag{40}$$

*is a minimizer of $G : \overline{\mathbf{C}} \in \mathbb{R}^{n \times n} \to E_L(\mathbf{C}_X(\mathbf{X}), \overline{\mathbf{C}}, \mathbf{T})$. For $\mathbf{T} \in \mathcal{U}_n(\mathbf{h}_X)$ the expression of the minimum is*

$$G(\mathbf{T}) = \text{cte} - \text{Tr}\left((\mathbf{T}^\top \mathbf{1}_N)(\mathbf{T}^\top \mathbf{1}_N)^\top (\overline{\mathbf{C}}(\mathbf{T}) \odot \overline{\mathbf{C}}(\mathbf{T}))\right), \tag{41}$$

*which defines a continuous function on $\mathcal{U}_n(\mathbf{h}_X)$. If $\mathbf{T} \in \mathcal{U}_n^+(\mathbf{h}_X)$ it becomes*

$$G(\mathbf{T}) = \text{cte} - \sum_{ij} \frac{(\mathbf{T}_{:,i}^\top \mathbf{C}_X(\mathbf{X}) \mathbf{T}_{:,j})^2}{(\mathbf{T}_{:,i}^\top \mathbf{1}_N)(\mathbf{T}_{:,j}^\top \mathbf{1}_N)} = \text{cte} - \| \text{diag}(\mathbf{T}^\top \mathbf{1}_N)^{-\frac{1}{2}} \mathbf{T}^\top \mathbf{C}_X(\mathbf{X}) \mathbf{T} \, \text{diag}(\mathbf{T}^\top \mathbf{1}_N)^{-\frac{1}{2}} \|_F^2. \tag{42}$$

*Proof.* First see that $\overline{\mathbf{C}}(\mathbf{T})$ is well defined since $\mathbf{T}_{:,i} \neq 0 \iff \mathbf{T}_{:,i}^\top \mathbf{1}_N \neq 0$ because $\mathbf{T}$ is non-negative. Consider, for $\mathbf{T} \in \mathcal{U}_n(\mathbf{h}_X)$, the function

$$F(\mathbf{T}, \overline{\mathbf{C}}) := E_L(\mathbf{C}_X(\mathbf{X}), \overline{\mathbf{C}}, \mathbf{T}) = \sum_{ijkl} ([\mathbf{C}_X(\mathbf{X})]_{ik} - \overline{C}_{jl})^2 T_{ij} T_{kl}, \tag{43}$$

A development yields (using $\mathbf{T}^\top \mathbf{1}_N = \mathbf{h}_X$)

$$F(\mathbf{T}, \overline{\mathbf{C}}) = \sum_{ik} [\mathbf{C}_X(\mathbf{X})]_{ik}^2 [\mathbf{h}_X]_i [\mathbf{h}_X]_k + \sum_{jl} \overline{C}_{jl}^2 (\sum_i T_{ij})(\sum_k T_{kl}) - 2 \sum_{ijkl} \overline{C}_{jl} [\mathbf{C}_X(\mathbf{X})]_{ik} T_{ij} T_{kl}. \tag{44}$$

We can rewrite $\sum_{ijkl} \overline{C}_{jl} [\mathbf{C}_X(\mathbf{X})]_{ik} T_{ij} T_{kl} = \text{Tr}(\mathbf{T}^\top \mathbf{C}_X(\mathbf{X}) \mathbf{T} \overline{\mathbf{C}})$. Also we have $(\sum_i T_{ij})(\sum_k T_{kl}) = [(\mathbf{T}^\top \mathbf{1}_N)(\mathbf{T}^\top \mathbf{1}_N)^\top]_{jl}$ Thus

$$\sum_{jl} \overline{C}_{jl}^2 (\sum_i T_{ij})(\sum_k T_{kl}) = \text{Tr}((\mathbf{T}^\top \mathbf{1}_N)(\mathbf{T}^\top \mathbf{1}_N)^\top (\overline{\mathbf{C}} \odot \overline{\mathbf{C}})). \tag{45}$$

Overall

$$F(\mathbf{T}, \overline{\mathbf{C}}) = \text{cte} + \text{Tr}((\mathbf{T}^\top \mathbf{1}_N)(\mathbf{T}^\top \mathbf{1}_N)^\top (\overline{\mathbf{C}} \odot \overline{\mathbf{C}})) - 2 \text{Tr}(\mathbf{T}^\top \mathbf{C}_X(\mathbf{X}) \mathbf{T} \overline{\mathbf{C}}). \tag{46}$$

Now taking the derivative with respect to $\overline{\mathbf{C}}$, the first order conditions are

$$\partial_2 F(\mathbf{T}, \overline{\mathbf{C}}) = 2(\overline{\mathbf{C}} \odot (\mathbf{T}^\top \mathbf{1}_N)(\mathbf{T}^\top \mathbf{1}_N)^\top - \mathbf{T}^\top \mathbf{C}_X(\mathbf{X}) \mathbf{T}) = 0. \tag{47}$$

For $(i,j)$ such that $\mathbf{T}_{:,i}$ and $\mathbf{T}_{:,j} \neq 0$ we have $[\partial_2 F(\mathbf{T}, \overline{\mathbf{C}}(\mathbf{T}))]_{ij} = 0$. For $(i,j)$ such that $\mathbf{T}_{:,i}$ or $\mathbf{T}_{:,j} = 0$ we have $[\overline{\mathbf{C}} \odot (\mathbf{T}^\top \mathbf{1}_N)(\mathbf{T}^\top \mathbf{1}_N)^\top]_{ij} = 0$ and also $[\mathbf{T}^\top \mathbf{C}_X(\mathbf{X}) \mathbf{T}]_{ij} = \mathbf{T}_{:,i}^\top \mathbf{C}_X(\mathbf{X}) \mathbf{T}_{:,j} = 0$. In particular the matrix $\overline{\mathbf{C}}(\mathbf{T})$ satisfies the first order conditions. When $L = L_2$ the problem $\min_{\overline{\mathbf{C}} \in \mathbb{R}^{n \times n}} E_L(\mathbf{C}_X(\mathbf{X}), \overline{\mathbf{C}}, \mathbf{T}) = \frac{1}{2} \sum_{ijkl} ([\mathbf{C}_X(\mathbf{X})]_{ik} - \overline{C}_{jl})^2 T_{ij} T_{kl}$ is convex in $\overline{\mathbf{C}}$. The first order conditions are sufficient hence $\overline{\mathbf{C}}(\mathbf{T})$ is a minimizer.

Also $\mathbf{T}^\top \mathbf{C}_X(\mathbf{X}) \mathbf{T} = \overline{\mathbf{C}}(\mathbf{T}) \odot (\mathbf{T}^\top \mathbf{1}_N)(\mathbf{T}^\top \mathbf{1}_N)^\top$ by definition of $\overline{\mathbf{C}}(\mathbf{T})$ thus

$$\begin{aligned} \text{Tr}(\mathbf{T}^\top \mathbf{C}_X(\mathbf{X}) \mathbf{T} \overline{\mathbf{C}}(\mathbf{T})) &= \text{Tr}([\overline{\mathbf{C}}(\mathbf{T}) \odot (\mathbf{T}^\top \mathbf{1}_N)(\mathbf{T}^\top \mathbf{1}_N)^\top] \overline{\mathbf{C}}(\mathbf{T})) \\ &= \text{Tr}((\mathbf{T}^\top \mathbf{1}_N)(\mathbf{T}^\top \mathbf{1}_N)^\top [\overline{\mathbf{C}}(\mathbf{T}) \odot \overline{\mathbf{C}}(\mathbf{T})]). \end{aligned} \tag{48}$$

Hence

$$F(\mathbf{T}, \overline{\mathbf{C}}(\mathbf{T})) = \text{cte} - \text{Tr}\left((\mathbf{T}^\top \mathbf{1}_N)(\mathbf{T}^\top \mathbf{1}_N)^\top (\overline{\mathbf{C}}(\mathbf{T}) \odot \overline{\mathbf{C}}(\mathbf{T}))\right). \tag{49}$$

Consequently for $\mathbf{T} \in \mathcal{U}_n(\mathbf{h}_X)$ such that $\forall i \in [\![n]\!], \mathbf{T}_{:,i} \neq 0$ we have

$$\begin{aligned} F(\mathbf{T}, \overline{\mathbf{C}}(\mathbf{T})) &= \text{cte} - \sum_{ij} \frac{(\mathbf{T}_{:,i}^\top \mathbf{C}_X(\mathbf{X}) \mathbf{T}_{:,j})^2}{(\mathbf{T}_{:,i}^\top \mathbf{1}_N)(\mathbf{T}_{:,j}^\top \mathbf{1}_N)} \\ &= \text{cte} - \| \text{diag}(\mathbf{T}^\top \mathbf{1}_N)^{-\frac{1}{2}} \mathbf{T}^\top \mathbf{C}_X(\mathbf{X}) \mathbf{T} \, \text{diag}(\mathbf{T}^\top \mathbf{1}_N)^{-\frac{1}{2}} \|_F^2. \end{aligned} \tag{50}$$

It just remains to demonstrate the continuity of $G$. We consider for $(\mathbf{x}, \mathbf{y}) \in \mathbb{R}_+^N \times \mathbb{R}_+^N$ the function

$$g(\mathbf{x}, \mathbf{y}) = \begin{cases} \frac{(\mathbf{x}^\top \mathbf{C}_X(\mathbf{X})\mathbf{y})^2}{\|\mathbf{x}\|_1 \|\mathbf{y}\|_1} & \text{when } \mathbf{x} \neq 0 \text{ and } \mathbf{y} \neq 0 \\ 0 & \text{otherwise} \end{cases} \tag{51}$$

and we show that $g$ is continuous. For $(\mathbf{x}, \mathbf{y}) \neq (0, 0)$ this is clear. Now using that

$$0 \leq g(\mathbf{x}, \mathbf{y}) = \frac{(\sum_{ij}[\mathbf{C}_X(\mathbf{X})]_{ij} x_i y_j)^2}{(\sum_i x_i)(\sum_j y_j)} \leq \|\mathbf{C}_X(\mathbf{X})\|_\infty^2 \frac{(\sum_i x_i)^2(\sum_j y_j)^2}{(\sum_i x_i)(\sum_j y_j)} = \|\mathbf{C}_X(\mathbf{X})\|_\infty^2 \|\mathbf{x}\|_1 \|\mathbf{y}\|_1, \tag{52}$$

this shows $\lim_{\mathbf{x} \to 0} g(\mathbf{x}, \mathbf{y}) = 0 = g(0, \mathbf{y})$ and $\lim_{\mathbf{y} \to 0} g(\mathbf{x}, \mathbf{y}) = 0 = g(\mathbf{x}, 0)$. Now for $\mathbf{T} \in \mathcal{U}_n(\mathbf{h}_X)$ we have $G(\mathbf{T}) = \text{cte} - \sum_{ij} g(\mathbf{T}_{:,i}, \mathbf{T}_{:,j})$ which defines a continuous function.

$\square$

To prove the theorem we will first prove that the function $G : \mathbf{T} \to \min_{\overline{\mathbf{C}} \in \mathbb{R}^{n \times n}} E_L(\mathbf{C}_X(\mathbf{X}), \overline{\mathbf{C}}, \mathbf{T})$ is *concave* on $\mathcal{U}_n^+(\mathbf{h}_X)$ and by a continuity argument it will be concave on $\mathcal{U}_n(\mathbf{h}_X)$. The concavity will allow us to prove that the minimum of $G$ is achieved in an extreme point of $\mathcal{U}_n(\mathbf{h}_X)$ which is a membership matrix.

**Proposition D.2.** *Let $\mathbf{h_X} \in \Sigma_N, L = L_2$ and suppose that $\mathbf{C}_X(\mathbf{X})$ is CPD or CND. Then the function $G : \mathbf{T} \to \min_{\overline{\mathbf{C}} \in \mathbb{R}^{n \times n}} E_L(\mathbf{C}_X(\mathbf{X}), \overline{\mathbf{C}}, \mathbf{T})$ is concave on $\mathcal{U}_n(\mathbf{h}_X)$. Consequently theorem 4.1 holds.*

*Proof.* We recall that $F(\mathbf{T}, \overline{\mathbf{C}}) := E_L(\mathbf{C}_X(\mathbf{X}), \overline{\mathbf{C}}, \mathbf{T})$ and $G(\mathbf{T}) = F(\mathbf{T}, \overline{\mathbf{C}}(\mathbf{T}))$. From lemma D.1 we know that $\overline{\mathbf{C}}(\mathbf{T})$ is a minimizer of $\overline{\mathbf{C}} \to F(\mathbf{T}, \overline{\mathbf{C}})$ hence it satisfies the first order conditions $\partial_2 F(\mathbf{T}, \overline{\mathbf{C}}(\mathbf{T})) = 0$. Every quantity is differentiable on $\mathcal{U}_n^+(\mathbf{h}_X)$. Hence, taking the derivative of $G$ and using the first order conditions

$$\nabla G(\mathbf{T}) = \partial_1 F(\mathbf{T}, \overline{\mathbf{C}}(\mathbf{T})) + \partial_2 F(\mathbf{T}, \overline{\mathbf{C}}(\mathbf{T}))[\nabla \overline{\mathbf{C}}(\mathbf{T})] = \partial_1 F(\mathbf{T}, \overline{\mathbf{C}}(\mathbf{T})). \tag{53}$$

We will found the expression of this gradient. In the proof of lemma D.1 we have seen that

$$F(\mathbf{T}, \overline{\mathbf{C}}) = \text{cte} + \text{Tr}((\mathbf{T}^\top \mathbf{1}_N)(\mathbf{T}^\top \mathbf{1}_N)^\top(\overline{\mathbf{C}} \odot \overline{\mathbf{C}})) - 2\,\text{Tr}(\mathbf{T}^\top \mathbf{C}_X(\mathbf{X})\mathbf{T}\overline{\mathbf{C}})$$
$$= \text{cte} + \text{Tr}(\mathbf{T}^\top \mathbf{1}_N \mathbf{1}_N^\top \mathbf{T}(\overline{\mathbf{C}} \odot \overline{\mathbf{C}})) - 2\,\text{Tr}(\mathbf{T}^\top \mathbf{C}_X(\mathbf{X})\mathbf{T}\overline{\mathbf{C}}). \tag{54}$$

Using that the derivative of $\mathbf{T} \to \text{Tr}(\mathbf{T}^\top \mathbf{A}\mathbf{T}\mathbf{B})$ is $\mathbf{A}^\top \mathbf{T}\mathbf{B}^\top + \mathbf{A}\mathbf{T}\mathbf{B}$ and that $\mathbf{C}_X(\mathbf{X})$ is symmetric we get

$$\partial_1 F(\mathbf{T}, \overline{\mathbf{C}}) = \mathbf{1}_N \mathbf{1}_N^\top \mathbf{T}(\overline{\mathbf{C}} \odot \overline{\mathbf{C}})^\top + \mathbf{1}_N \mathbf{1}_N^\top \mathbf{T}(\overline{\mathbf{C}} \odot \overline{\mathbf{C}}) - 2\mathbf{C}_X(\mathbf{X})\mathbf{T}\overline{\mathbf{C}}^\top - 2\mathbf{C}_X(\mathbf{X})\mathbf{T}\overline{\mathbf{C}}. \tag{55}$$

Finally, applying to the symmetric matrix $\overline{\mathbf{C}} = \overline{\mathbf{C}}(\mathbf{T})$

$$\nabla G(\mathbf{T}) = \partial_1 F(\mathbf{T}, \overline{\mathbf{C}}(\mathbf{T})) = 2\left(\mathbf{1}_N \mathbf{1}_N^\top \mathbf{T}(\overline{\mathbf{C}}(\mathbf{T}) \odot \overline{\mathbf{C}}(\mathbf{T})) - 2\mathbf{C}_X(\mathbf{X})\mathbf{T}\overline{\mathbf{C}}(\mathbf{T})\right). \tag{56}$$

In what follows we define

$$D(\mathbf{T}) := \text{diag}(\mathbf{T}^\top \mathbf{1}_N)^{-1} \in \mathbb{R}^{n \times n}, \tag{57}$$

when applicable. Using the expression of the gradient we will show that $G$ is concave on $\mathcal{U}_n^+(\mathbf{h}_X)$ and we will conclude by a continuity argument on $\mathcal{U}_n(\mathbf{h}_X)$. Take $(\mathbf{P}, \mathbf{Q}) \in \mathcal{U}_n^+(\mathbf{h}_X) \times \mathcal{U}_n^+(\mathbf{h}_X)$ we will prove

$$G(\mathbf{P}) - G(\mathbf{Q}) - \langle \nabla G(\mathbf{Q}), \mathbf{P} - \mathbf{Q} \rangle \leq 0. \tag{58}$$

From lemma D.1 we have the expression (since $\mathbf{P} \in \mathcal{U}_n^+(\mathbf{h}_X)$)

$$G(\mathbf{P}) = \text{cte} - \|D(\mathbf{P})^{\frac{1}{2}} \mathbf{P}^\top \mathbf{C}_X(\mathbf{X})\mathbf{P}D(\mathbf{P})^{\frac{1}{2}}\|_F^2, \tag{59}$$

(same for $G(\mathbf{Q})$) and

$$\overline{\mathbf{C}}(\mathbf{Q}) = D(\mathbf{Q})\mathbf{Q}^\top \mathbf{C}_X(\mathbf{X})\mathbf{Q}D(\mathbf{Q}). \tag{60}$$

We will now calculate $\langle \nabla G(\mathbf{Q}), \mathbf{P} \rangle$ which involves $\langle \mathbf{1}_N \mathbf{1}_N^\top \mathbf{Q}(\overline{\mathbf{C}}(\mathbf{Q}) \odot \overline{\mathbf{C}}(\mathbf{Q}), \mathbf{P} \rangle$ and $\langle \mathbf{C}_X(\mathbf{X}) \mathbf{Q} \overline{\mathbf{C}}(\mathbf{Q}), \mathbf{P} \rangle$. For the first term we have

$$
\begin{aligned}
&\langle \mathbf{1}_N \mathbf{1}_N^\top \mathbf{Q}(\overline{\mathbf{C}}(\mathbf{Q}) \odot \overline{\mathbf{C}}(\mathbf{Q})), \mathbf{P} \rangle \\
&= \mathrm{Tr}(\mathbf{P}^\top \mathbf{1}_N \mathbf{1}_N^\top \mathbf{Q} \overline{\mathbf{C}}(\mathbf{Q})^{\odot 2}) = \mathrm{Tr}(\mathbf{1}_N^\top \mathbf{Q} \overline{\mathbf{C}}(\mathbf{Q})^{\odot 2} \mathbf{P}^\top \mathbf{1}_N) \\
&= (\mathbf{Q}^\top \mathbf{1}_N)^\top (\overline{\mathbf{C}}(\mathbf{Q}) \odot \overline{\mathbf{C}}(\mathbf{Q})) \mathbf{P}^\top \mathbf{1}_N \\
&= \mathrm{Tr}(\overline{\mathbf{C}}(\mathbf{Q}) \mathrm{diag}(\mathbf{Q}^\top \mathbf{1}_N) \overline{\mathbf{C}}(\mathbf{Q}) \mathrm{diag}(\mathbf{P}^\top \mathbf{1}_N)) \\
&= \mathrm{Tr}\left([D(\mathbf{Q})\mathbf{Q}^\top \mathbf{C}_X(\mathbf{X})\mathbf{Q}D(\mathbf{Q})]D(\mathbf{Q})^{-1}[D(\mathbf{Q})\mathbf{Q}^\top \mathbf{C}_X(\mathbf{X})\mathbf{Q}D(\mathbf{Q})]D(\mathbf{P})^{-1}\right) \\
&= \mathrm{Tr}(D(\mathbf{Q})\mathbf{Q}^\top \mathbf{C}_X(\mathbf{X})\mathbf{Q}D(\mathbf{Q})\mathbf{Q}^\top \mathbf{C}_X(\mathbf{X})\mathbf{Q}D(\mathbf{Q})D(\mathbf{P})^{-1}) \\
&= \mathrm{Tr}(D(\mathbf{P})^{-\frac{1}{2}}[D(\mathbf{Q})\mathbf{Q}^\top \mathbf{C}_X(\mathbf{X})\mathbf{Q}D(\mathbf{Q})\mathbf{Q}^\top \mathbf{C}_X(\mathbf{X})\mathbf{Q}D(\mathbf{Q})]D(\mathbf{P})^{-\frac{1}{2}}) \\
&= \mathrm{Tr}(D(\mathbf{P})^{-\frac{1}{2}}D(\mathbf{Q})\mathbf{Q}^\top \mathbf{C}_X(\mathbf{X})\mathbf{Q}D(\mathbf{Q})^{\frac{1}{2}}D(\mathbf{Q})^{\frac{1}{2}}\mathbf{Q}^\top \mathbf{C}_X(\mathbf{X})\mathbf{Q}D(\mathbf{Q})D(\mathbf{P})^{-\frac{1}{2}}) \\
&= \langle D(\mathbf{P})^{-\frac{1}{2}}D(\mathbf{Q})\mathbf{Q}^\top \mathbf{C}_X(\mathbf{X})\mathbf{Q}D(\mathbf{Q})^{\frac{1}{2}}, D(\mathbf{P})^{-\frac{1}{2}}D(\mathbf{Q})\mathbf{Q}^\top \mathbf{C}_X(\mathbf{X})\mathbf{Q}D(\mathbf{Q})^{\frac{1}{2}} \rangle \\
&= \| D(\mathbf{P})^{-\frac{1}{2}}D(\mathbf{Q})\mathbf{Q}^\top \mathbf{C}_X(\mathbf{X})\mathbf{Q}D(\mathbf{Q})^{\frac{1}{2}} \|_F^2 \, .
\end{aligned}
\tag{61}
$$

For the second term

$$
\begin{aligned}
&\langle \mathbf{C}_X(\mathbf{X}) \mathbf{Q} \overline{\mathbf{C}}(\mathbf{Q}), \mathbf{P} \rangle \\
&= \mathrm{Tr}(\mathbf{P}^\top \mathbf{C}_X(\mathbf{X})\mathbf{Q}D(\mathbf{Q})\mathbf{Q}^\top \mathbf{C}_X(\mathbf{X})\mathbf{Q}D(\mathbf{Q})) \\
&= \mathrm{Tr}(D(\mathbf{Q})^{\frac{1}{2}}\mathbf{P}^\top \mathbf{C}_X(\mathbf{X})\mathbf{Q}D(\mathbf{Q})^{\frac{1}{2}}D(\mathbf{Q})^{\frac{1}{2}}\mathbf{Q}^\top \mathbf{C}_X(\mathbf{X})\mathbf{Q}D(\mathbf{Q})^{\frac{1}{2}}) \\
&= \langle D(\mathbf{Q})^{\frac{1}{2}}\mathbf{P}^\top \mathbf{C}_X(\mathbf{X})\mathbf{Q}D(\mathbf{Q})^{\frac{1}{2}}, D(\mathbf{Q})^{\frac{1}{2}}\mathbf{Q}^\top \mathbf{C}_X(\mathbf{X})\mathbf{Q}D(\mathbf{Q})^{\frac{1}{2}} \rangle \\
&= \langle D(\mathbf{Q})^{\frac{1}{2}}\mathbf{P}^\top \mathbf{C}_X(\mathbf{X})\mathbf{Q}D(\mathbf{Q})^{\frac{1}{2}}, D(\mathbf{P})^{\frac{1}{2}}D(\mathbf{Q})^{-\frac{1}{2}}D(\mathbf{P})^{-\frac{1}{2}}D(\mathbf{Q})\mathbf{Q}^\top \mathbf{C}_X(\mathbf{X})\mathbf{Q}D(\mathbf{Q})^{\frac{1}{2}} \rangle \\
&= \langle D(\mathbf{P})^{\frac{1}{2}}\mathbf{P}^\top \mathbf{C}_X(\mathbf{X})\mathbf{Q}D(\mathbf{Q})^{\frac{1}{2}}, D(\mathbf{P})^{-\frac{1}{2}}D(\mathbf{Q})\mathbf{Q}^\top \mathbf{C}_X(\mathbf{X})\mathbf{Q}D(\mathbf{Q})^{\frac{1}{2}} \rangle \, .
\end{aligned}
\tag{62}
$$

This gives

$$
\begin{aligned}
\langle \nabla G(\mathbf{Q}), \mathbf{P} \rangle &= 2\| D(\mathbf{P})^{-\frac{1}{2}}D(\mathbf{Q})\mathbf{Q}^\top \mathbf{C}_X(\mathbf{X})\mathbf{Q}D(\mathbf{Q})^{\frac{1}{2}} \|_F^2 \\
&\quad - 4\langle D(\mathbf{P})^{\frac{1}{2}}\mathbf{P}^\top \mathbf{C}_X(\mathbf{X})\mathbf{Q}D(\mathbf{Q})^{\frac{1}{2}}, D(\mathbf{P})^{-\frac{1}{2}}D(\mathbf{Q})\mathbf{Q}^\top \mathbf{C}_X(\mathbf{X})\mathbf{Q}D(\mathbf{Q})^{\frac{1}{2}} \rangle \\
&= 2\| D(\mathbf{P})^{-\frac{1}{2}}D(\mathbf{Q})\mathbf{Q}^\top \mathbf{C}_X(\mathbf{X})\mathbf{Q}D(\mathbf{Q})^{\frac{1}{2}} - D(\mathbf{P})^{\frac{1}{2}}\mathbf{P}^\top \mathbf{C}_X(\mathbf{X})\mathbf{Q}D(\mathbf{Q})^{\frac{1}{2}} \|_F^2 \\
&\quad - 2\| D(\mathbf{P})^{\frac{1}{2}}\mathbf{P}^\top \mathbf{C}_X(\mathbf{X})\mathbf{Q}D(\mathbf{Q})^{\frac{1}{2}} \|_F^2 \, .
\end{aligned}
\tag{63}
$$

From this equation we get directly that

$$
\begin{aligned}
\langle \nabla G(\mathbf{Q}), \mathbf{Q} \rangle &= -2\| D(\mathbf{Q})^{\frac{1}{2}}\mathbf{Q}^\top \mathbf{C}_X(\mathbf{X})\mathbf{Q}D(\mathbf{Q})^{\frac{1}{2}} \|_F^2 \\
\text{and } \langle \nabla G(\mathbf{Q}), \mathbf{P} \rangle &\geq -2\| D(\mathbf{P})^{\frac{1}{2}}\mathbf{P}^\top \mathbf{C}_X(\mathbf{X})\mathbf{Q}D(\mathbf{Q})^{\frac{1}{2}} \|_F^2 \, .
\end{aligned}
\tag{64}
$$

Hence

$$
\begin{aligned}
&G(\mathbf{P}) - G(\mathbf{Q}) - \langle \nabla G(\mathbf{Q}), \mathbf{P} - \mathbf{Q} \rangle \\
&= -\| D(\mathbf{P})^{\frac{1}{2}}\mathbf{P}^\top \mathbf{C}_X(\mathbf{X})\mathbf{P}D(\mathbf{P})^{\frac{1}{2}} \|_F^2 + \| D(\mathbf{Q})^{\frac{1}{2}}\mathbf{Q}^\top \mathbf{C}_X(\mathbf{X})\mathbf{Q}D(\mathbf{Q})^{\frac{1}{2}} \|_F^2 \\
&\quad - \langle \nabla G(\mathbf{Q}), \mathbf{P} \rangle + \langle \nabla G(\mathbf{Q}), \mathbf{Q} \rangle \\
&\overset{eq. (64)}{=} -\| D(\mathbf{P})^{\frac{1}{2}}\mathbf{P}^\top \mathbf{C}_X(\mathbf{X})\mathbf{P}D(\mathbf{P})^{\frac{1}{2}} \|_F^2 - \| D(\mathbf{Q})^{\frac{1}{2}}\mathbf{Q}^\top \mathbf{C}_X(\mathbf{X})\mathbf{Q}D(\mathbf{Q})^{\frac{1}{2}} \|_F^2 - \langle \nabla G(\mathbf{Q}), \mathbf{P} \rangle \\
&\overset{eq. (64)}{\leq} -\| D(\mathbf{P})^{\frac{1}{2}}\mathbf{P}^\top \mathbf{C}_X(\mathbf{X})\mathbf{P}D(\mathbf{P})^{\frac{1}{2}} \|_F^2 - \| D(\mathbf{Q})^{\frac{1}{2}}\mathbf{Q}^\top \mathbf{C}_X(\mathbf{X})\mathbf{Q}D(\mathbf{Q})^{\frac{1}{2}} \|_F^2 \\
&\quad + 2\| D(\mathbf{P})^{\frac{1}{2}}\mathbf{P}^\top \mathbf{C}_X(\mathbf{X})\mathbf{Q}D(\mathbf{Q})^{\frac{1}{2}} \|_F^2 \, .
\end{aligned}
\tag{65}
$$

We note $\mathbf{U} = \mathbf{P}D(\mathbf{P})\mathbf{P}^\top \in \mathbb{R}^{N \times N}, \mathbf{V} = \mathbf{Q}D(\mathbf{Q})\mathbf{Q}^\top \in \mathbb{R}^{N \times N}$, the previous calculus shows that

$$
\begin{aligned}
G(\mathbf{P}) - G(\mathbf{Q}) - \langle \nabla G(\mathbf{Q}), \mathbf{P} - \mathbf{Q} \rangle &\leq -\mathrm{Tr}(\mathbf{U}\mathbf{C}_X(\mathbf{X})\mathbf{U}\mathbf{C}_X(\mathbf{X})) - \mathrm{Tr}(\mathbf{V}\mathbf{C}_X(\mathbf{X})\mathbf{V}\mathbf{C}_X(\mathbf{X})) \\
&\quad + 2\mathrm{Tr}(\mathbf{V}\mathbf{C}_X(\mathbf{X})\mathbf{U}\mathbf{C}_X(\mathbf{X})) \, ,
\end{aligned}
\tag{66}
$$

Now note that
$$\mathbf{U}^\top \mathbf{1}_N = \mathbf{P}D(\mathbf{P})\mathbf{P}^\top \mathbf{1}_N = \mathbf{P}\operatorname{diag}(\mathbf{P}^\top \mathbf{1}_N)^{-1}\mathbf{P}^\top \mathbf{1}_N = \mathbf{P}\mathbf{1}_N = \mathbf{h}_X \,. \tag{67}$$

Since $\mathbf{U}$ is symmetric we also have $\mathbf{U}\mathbf{1}_N = \mathbf{h}_X$ and similarly we have the same result for $\mathbf{V}$. Overall $\mathbf{V}^\top \mathbf{1}_N = \mathbf{U}^\top \mathbf{1}_N$ and $\mathbf{V}\mathbf{1}_N = \mathbf{U}\mathbf{1}_N$. Since $\mathbf{C}_X(\mathbf{X})$ is CPD or CND we can apply lemma D.3 below which proves that $-\operatorname{Tr}(\mathbf{U}\mathbf{C}_X(\mathbf{X})\mathbf{U}\mathbf{C}_X(\mathbf{X})) - \operatorname{Tr}(\mathbf{V}\mathbf{C}_X(\mathbf{X})\mathbf{V}\mathbf{C}_X(\mathbf{X})) + 2\operatorname{Tr}(\mathbf{V}\mathbf{C}_X(\mathbf{X})\mathbf{U}\mathbf{C}_X(\mathbf{X})) \leq 0$ and consequently that $G$ is concave on $\mathcal{U}_n^+(\mathbf{h}_X)$. We now use the continuity of $G$ to prove that it is concave on $\mathcal{U}_n(\mathbf{h}_X)$.

Take $\mathbf{P} \in \mathcal{U}_n^+(\mathbf{h}_X)$ and $\mathbf{Q} \in \mathcal{U}_n(\mathbf{h}_X) \setminus \mathcal{U}_n^+(\mathbf{h}_X)$ *i.e.* there exists $k \in [\![n]\!]$ such that $\mathbf{Q}_{:,k} = 0$. Without loss of generality we suppose $k = 1$. Consider for $m \in \mathbb{N}^*$ the matrix $\mathbf{Q}^{(m)} = (\frac{1}{m}\mathbf{1}_N, \mathbf{Q}_{:,2}, \cdots, \mathbf{Q}_{:,n})$. Then $\mathbf{Q}^{(m)} \to \mathbf{Q}$ as $m \to +\infty$. Also since $\mathbf{Q}^{(m)} \in \mathcal{U}_n^+(\mathbf{h}_X)$ we have by concavity of $G$

$$G((1-\lambda)\mathbf{P} + \lambda\mathbf{Q}^{(m)}) \geq (1-\lambda)G(\mathbf{P}) + \lambda G(\mathbf{Q}^{(m)}) \,, \tag{68}$$

for any $\lambda \in [0,1]$. Taking the limit as $m \to \infty$ gives, by continuity of $G$,

$$G((1-\lambda)\mathbf{P} + \lambda\mathbf{Q}) \geq (1-\lambda)G(\mathbf{P}) + \lambda G(\mathbf{Q}) \,, \tag{69}$$

and hence $G$ is concave on $\mathcal{U}_n(\mathbf{h}_X)$. This proves theorem 4.1. Indeed the minimization of $\mathbf{T} \in \mathcal{U}_n(\mathbf{h}_X) \to \min_{\overline{\mathbf{C}}} E_L(\mathbf{C}_X(\mathbf{X}), \overline{\mathbf{C}}, \mathbf{T})$ is a minimization of a concave function over a polytope, hence admits an extremity of $\mathcal{U}_n(\mathbf{h}_X)$ as minimizer. But these extremities are membership matrices as they can be described as $\{\operatorname{diag}(\mathbf{h}_X)\mathbf{P} : \mathbf{P} \in \{0,1\}^{N \times n}, \mathbf{P}^\top \mathbf{1}_n = \mathbf{1}_N\}$ (Cao et al., 2022). $\qquad\square$

**Lemma D.3.** *Let $\mathbf{C} \in \mathbb{R}^{N \times N}$ be a CPD or CND matrix. Then for any $(\mathbf{P}, \mathbf{Q}) \in \mathbb{R}^{N \times N} \times \mathbb{R}^{N \times N}$ such that $\mathbf{P}^\top \mathbf{1}_N = \mathbf{Q}^\top \mathbf{1}_N$ and $\mathbf{P}\mathbf{1}_N = \mathbf{Q}\mathbf{1}_N$ we have*

$$\operatorname{Tr}(\mathbf{P}^\top \mathbf{C}\mathbf{Q}\mathbf{C}) \leq \frac{1}{2}(\operatorname{Tr}(\mathbf{P}^\top \mathbf{C}\mathbf{P}\mathbf{C}) + \operatorname{Tr}(\mathbf{Q}^\top \mathbf{C}\mathbf{Q}\mathbf{C})) \,. \tag{70}$$

*Proof.* First, since $\mathbf{C}$ is symmetric,

$$\begin{aligned}\operatorname{Tr}\left((\mathbf{P} - \mathbf{Q})^\top \mathbf{C}(\mathbf{P} - \mathbf{Q})\mathbf{C}\right) &= \operatorname{Tr}(\mathbf{P}^\top \mathbf{C}\mathbf{P}\mathbf{C} - \mathbf{P}^\top \mathbf{C}\mathbf{Q}\mathbf{C} - \mathbf{Q}^\top \mathbf{C}\mathbf{P}\mathbf{C} + \mathbf{Q}^\top \mathbf{C}\mathbf{Q}\mathbf{C}) \\ &= \operatorname{Tr}(\mathbf{P}^\top \mathbf{C}\mathbf{P}\mathbf{C}) + \operatorname{Tr}(\mathbf{Q}^\top \mathbf{C}\mathbf{Q}\mathbf{C}) - 2\operatorname{Tr}(\mathbf{P}^\top \mathbf{C}\mathbf{Q}\mathbf{C}) \,. \end{aligned} \tag{71}$$

We note $\mathbf{U} = \mathbf{P} - \mathbf{Q}$. Since $\mathbf{P}^\top \mathbf{1}_N = \mathbf{Q}^\top \mathbf{1}_N$ we have $\mathbf{U}^\top \mathbf{1}_N = 0$. In the same way $\mathbf{U}\mathbf{1}_N = 0$. We introduce $\mathbf{H} = \mathbf{I}_N - \frac{1}{N}\mathbf{1}_N\mathbf{1}_N^\top$ the centering matrix. Note that

$$\mathbf{H}\mathbf{U}\mathbf{H} = (\mathbf{U} - \frac{1}{N}\mathbf{1}_N(\mathbf{1}_N^\top \mathbf{U}))\mathbf{H} = \mathbf{U}\mathbf{H} = \mathbf{U} - \frac{1}{N}(\mathbf{U}\mathbf{1}_N)\mathbf{1}_N^\top = \mathbf{U} \,. \tag{72}$$

Also $\mathbf{C}$ is CPD if and only if $\mathbf{H}\mathbf{C}\mathbf{H}$ is positive semi-definite (PSD). Indeed if $\mathbf{H}\mathbf{C}\mathbf{H}$ is PSD then take $\mathbf{x}$ such that $\mathbf{x}^\top \mathbf{1}_N = 0$. We then have $\mathbf{H}\mathbf{x} = \mathbf{x}$ and thus $\mathbf{x}^\top \mathbf{C}\mathbf{x} = \mathbf{x}^\top(\mathbf{H}\mathbf{C}\mathbf{H})\mathbf{x} \geq 0$. On the other hand when $\mathbf{C}$ is CPD then take any $\mathbf{x}$ and see that $\mathbf{x}^\top \mathbf{H}\mathbf{C}\mathbf{H}\mathbf{x} = (\mathbf{H}\mathbf{x})^\top \mathbf{C}(\mathbf{H}\mathbf{x})$. But $(\mathbf{H}\mathbf{x})^\top \mathbf{1}_N = \mathbf{x}^\top(\mathbf{H}^\top \mathbf{1}_N) = 0$. So $(\mathbf{H}\mathbf{x})^\top \mathbf{C}(\mathbf{H}\mathbf{x}) \geq 0$.

By hypothesis $\mathbf{C}$ is CPD so $\mathbf{H}\mathbf{C}\mathbf{H}$ is PSD and symmetric, so it has a square root. But using eq. (72) we get

$$\begin{aligned}\operatorname{Tr}\left((\mathbf{P} - \mathbf{Q})^\top \mathbf{C}(\mathbf{P} - \mathbf{Q})\mathbf{C}\right) &= \operatorname{Tr}(\mathbf{U}^\top \mathbf{C}\mathbf{U}\mathbf{C}) = \operatorname{Tr}(\mathbf{H}\mathbf{U}^\top \mathbf{H}\mathbf{C}\mathbf{H}\mathbf{U}\mathbf{H}\mathbf{C}) \\ &= \operatorname{Tr}(\mathbf{U}^\top(\mathbf{H}\mathbf{C}\mathbf{H})\mathbf{U}(\mathbf{H}\mathbf{C}\mathbf{H})) \\ &= \|(\mathbf{H}\mathbf{C}\mathbf{H})^{\frac{1}{2}}\mathbf{U}(\mathbf{H}\mathbf{C}\mathbf{H})^{\frac{1}{2}}\|_F^2 \geq 0 \,, \end{aligned} \tag{73}$$

For the CND case is suffices to use that $\mathbf{C}$ is CND if and only if $-\mathbf{C}$ is CPD and that $\operatorname{Tr}\left((\mathbf{P} - \mathbf{Q})^\top \mathbf{C}(\mathbf{P} - \mathbf{Q})\mathbf{C}\right) = \operatorname{Tr}\left((\mathbf{P} - \mathbf{Q})^\top(-\mathbf{C})(\mathbf{P} - \mathbf{Q})(-\mathbf{C})\right)$ which concludes the proof. $\qquad\square$

# E   Algorithmic details

We detail in the following the algorithms mentioned in Section 4 to address the semi-relaxed GW divergence computation in our Block Coordinate Descent algorithm for the DistR problem. We begin with details on the computation of an equivalent objective function and its gradient, potentially under low-rank assumptions over structures $\mathbf{C}$ and $\overline{\mathbf{C}}$.

### E.1 Objective function and gradient computation.

**Problem statement.** Let consider any matrices $\mathbf{C} \in \mathbb{R}^{n \times n}$, $\overline{\mathbf{C}} \in \mathbb{R}^{m \times m}$, and a probability vector $\mathbf{h} \in \Sigma_n$. In all our use cases, we considered inner losses $L : \mathbb{R} \times \mathbb{R} \to \mathbb{R}_+$ which can be decomposed following Proposition 1 in Peyré et al. (2016). Namely we assume the existence of functions $f_1, f_2, h_1, h_2$ such that

$$\forall a, b \in \Omega^2, \quad L(a, b) = f_1(a) + f_2(b) - h_1(a)h_2(b) \tag{74}$$

More specifically we considered

$$L_2(a, b) = (a - b)^2 \implies f_1(a) = a^2, \quad f_2(b) = b^2, \quad h_1(a) = a, \quad h_2(b) = 2b,$$

$$L_{KL}(a, b) = a \log \frac{a}{b} - a + b \implies f_1(a) = a \log a - a, \quad f_2(b) = b, \quad h_1(a) = a, \quad h_2(b) = \log b$$

$$L_{BCE}(a, b) = a \log \frac{a}{b} + (1 - a) \log \frac{1 - a}{1 - b} \implies f_1(a) = a \log a + (1 - a) \log(1 - a), \quad f_2(b) = -\log(1 - b),$$

$$h_1(a) = a, \quad h_2(b) = \log \frac{b}{1 - b} \tag{L2}$$

In this setting, we proposed to solve for the equivalent problem to $\mathrm{srGW}_L$ :

$$\min_{\mathbf{T} \in \mathcal{U}_n(\mathbf{h})} F(\mathbf{T}) \tag{srGW-2}$$

where the objective function reads as

$$
\begin{aligned}
F(\mathbf{T}) &:= \langle F_1(\overline{\mathbf{C}}, \mathbf{T}) - F_2(\mathbf{C}, \overline{\mathbf{C}}, \mathbf{T}), \mathbf{T} \rangle \\
&= \langle \mathbf{1}_N \mathbf{1}_N^\top \mathbf{T} f_2(\overline{\mathbf{C}}), \mathbf{T} \rangle - \langle h_1(\mathbf{C}) \mathbf{T} h_2(\overline{\mathbf{C}})^\top, \mathbf{T} \rangle
\end{aligned}
\tag{75}
$$

Problem srGW-2 is usually a non-convex QP with Hessian $\mathcal{H} = f_2(\overline{\mathbf{C}}) \otimes \mathbf{1}\mathbf{1}^\top - h_2(\overline{\mathbf{C}}) \otimes_K h_1(\mathbf{C})$. In all cases this equivalent form is interesting as it avoids computing the constant term $\langle f_1(\mathbf{C}), \mathbf{h}\mathbf{h}^\top \rangle$ that requires $O(N^2)$ operations in all cases.

The gradient of $F$ w.r.t $\mathbf{T}$ then reads as

$$\nabla_{\mathbf{T}} F(\mathbf{C}, \overline{\mathbf{C}}, \mathbf{T}) = F_1(\overline{\mathbf{C}}, \mathbf{T}) + F_1(\overline{\mathbf{C}}^\top, \mathbf{T}) - F_2(\mathbf{C}, \overline{\mathbf{C}}, \mathbf{T}) - F_2(\mathbf{C}^\top, \overline{\mathbf{C}}^\top, \mathbf{T}) \tag{76}$$

When $C_X(\mathbf{X})$ and $C_Z(\mathbf{Z})$ are symmetric, which is the case in all our experiments, this gradient reduces to $\nabla_{\mathbf{T}} F = 2(F_1 - F_2)$.

**Low-rank factorization.** Inspired from the work of Scetbon et al. (2022), we propose implementations of srGW that can leverage the low-rank nature of $\mathbf{C}_X(\mathbf{X})$ and $\mathbf{C}_Z(\mathbf{Z})$. Let us assume that both structures can be exactly decomposed as follows, $\mathbf{C}_X(\mathbf{X}) = \mathbf{A}_1 \mathbf{A}_2^\top$ where $\mathbf{A}_1, \mathbf{A}_2 \in \mathbb{R}^{N \times r}$ and $\mathbf{C}_Z(\mathbf{Z}) = \mathbf{B}_1 \mathbf{B}_2^\top$ with $\mathbf{B}_1, \mathbf{B}_2 \in \mathbb{R}^{n \times s}$, such that $r << N$ and $s << n$, can differ respectively from respective dimensions $p$ and $d$ (e.g. for used squared Euclidean distance matrices $r = p + 2$ and $s = d + 2$ ). For both inner losses $L$ we make use of the following factorization:

$\underline{L = L_2}$: Computing the first term $F_1$ coming for the optimized second marginal can benefit from being factored if $d^2 << n$. Indeed, as $f_2(\mathbf{C}_Z(\mathbf{Z})) = \mathbf{C}_Z(\mathbf{Z})^2 = (\mathbf{B}_1 \mathbf{B}_2^\top) \odot (\mathbf{B}_1 \mathbf{B}_2^\top)$, one can use the flattened out product operator described in Scetbon et al. (2022, Section 5), to compute $\mathbf{C}_Z(\mathbf{Z})^2 \mathbf{T}^\top \mathbf{1}_N = \mathbf{x}$ in $O(min(n^2, ns^2))$. This way $F_1(\mathbf{T})$ results from stacking $N$ times $\mathbf{x}$ in $O(1)$ operations for a total number of computions of $N + O(min(n^2, ns^2))$. And its scalar product with $\mathbf{T}$ to compute the loss comes down to $O(Nn)$ additional operations.

Then computing $F_2(\mathbf{T})$ and its scalar product with $\mathbf{T}$ can be done following the development of Scetbon et al. (2022, Section 3) for the corresponding GW problem, in $O(Nn(r + s) + rs(N + n))$ operations. So the overall complexity at is $O(Nn(r + s) + rs(N + n) + min(n^2, ns^2))$.

$L = L_{KL}$: In this setting $f_2(\mathbf{C}_Z(\mathbf{Z})) = \mathbf{C}_Z(\mathbf{Z})$ and $h_1(\mathbf{C}_X(\mathbf{X})) = \mathbf{C}_X(\mathbf{X})$ naturally preserves the low-rank nature of input matrices, but $h_2(\mathbf{C}_Z(\mathbf{Z})) = \log(\mathbf{C}_Z(\mathbf{Z}))$ does not. So computing the first term $F_1$, can be performed following this paranthesis order $\mathbf{1}_N((\mathbf{1}_N^\top \mathbf{T})\mathbf{A}_1))\mathbf{A}_2^\top)$ in $O(N(n+s))$ operations. While the second term $F_2$ should be computed following this order $\mathbf{A}_1((\mathbf{A}_2^\top \mathbf{T})\log(\mathbf{C}_Z(\mathbf{Z})))$ in $O(Nnr + rn^2)$ operations. While their respective scalar product can be computed in $O(Nn)$. So the overall complexity is $O(Nnr + n^2 r)$. Similar considerations can be applied to $L_{BCE}$.

Notice that in the gaussian kernel case for neighbor embedding methods, where $[\mathbf{C}_Z(\mathbf{Z})]_{ij} = \exp(-\|\mathbf{z}_i - \mathbf{z}_j\|_2^2)$ up to some normalization. We have $[h_2(\mathbf{C}_Z(\mathbf{Z}))]_{ij} = -\|\mathbf{z}_i - \mathbf{z}_j\|_2^2$ which admits a low-rank factorization such that we can recover the complexity illustrated above for $L = L_2$.

### E.2 Solvers.

We develop next our extension of both the Mirror Descent and Conditional Gradient solvers first introduced in Vincent-Cuaz et al. (2022a), for any inner loss $L$ that decomposes as in equation 74 .

**Mirror Descent algorithm.** This solver comes down to solve for the *exact* srGW problem using mirror-descent scheme w.r.t the KL geometry. At each iteration $(i)$, the solver comes down to, first computing the gradient $\nabla_{\mathbf{T}} F(\mathbf{T}^{(i)})$ given in equation 76 evaluated in $\mathbf{T}^{(i)}$, then updating the transport plan using the following closed-form solution to a KL projection:

$$\mathbf{T}^{(i+1)} \leftarrow \operatorname{diag}\left(\frac{\mathbf{h}}{\mathbf{K}^{(i)}\mathbf{1}_n}\right)\mathbf{K}^{(i)} \tag{77}$$

where $\mathbf{K}^{(i)} = \exp\left(\nabla_{\mathbf{T}} F(\mathbf{T}^{(i)}) - \varepsilon \log(\mathbf{T}^{(i)})\right)$ and $\varepsilon > 0$ is an hyperparameter to tune. Proposition 3 and Lemma 7 in Vincent-Cuaz (2023, Chapter 6) provides that the Mirror-Descent algorithm converges to a stationary point non-asymptotically when $L = L_2$. A quick inspection of the proof suffices to see that this convergence holds for any losses $L$ satisfying equation 74, up to adaptation of constants involved in the Lemma.

**Conditional Gradient algorithm.** This algorithm, known to converge to local optimum (Lacoste-Julien, 2016), iterates over the 3 steps summarized in Algorithm 1: .

---

**Algorithm 1** CG solver for srGW$_L$

---

1: **repeat**
2:     $\mathbf{F}^{(i)} \leftarrow$ Compute gradient w.r.t $\mathbf{T}$ of equation 76.
3:     $\mathbf{X}^{(i)} \leftarrow \min_{\substack{\mathbf{X}\mathbf{1}_m = \mathbf{h} \\ \mathbf{X} \geq 0}} \langle \mathbf{X}, \mathbf{F}^{(i)} \rangle$
4:     $\mathbf{T}^{(i+1)} \leftarrow (1 - \gamma^\star)\mathbf{T}^{(i)} + \gamma^\star \mathbf{X}^{(i)}$ with $\gamma^\star \in [0,1]$ from exact-line search.
5: **until** convergence.

---

This algorithm consists in solving at each iteration $(i)$ a linearization $\langle \mathbf{X}, \mathbf{F}^{(i)} \rangle$ of the problem equation srGW-2 where $\mathbf{F}(\mathbf{T}^{(i)})$ is the gradient of the objective in equation 76. The solution of the linearized problem provides a *descent direction* $\mathbf{X}^{(i)} - \mathbf{T}^{(i)}$, and a line-search whose optimal step can be found in closed form to update the current solution $\mathbf{T}^{(i)}$. We detail in the following this line-search step for any loss that can be decomposed as in equation 74. It comes down for any $\mathbf{T} \in \mathcal{U}_n(\mathbf{h})$, to solve the following problem:

$$\gamma = \operatorname*{arg\,min}_{\gamma \in [0,1]} g(\gamma) := F(\mathbf{T} + \gamma(\mathbf{X} - \mathbf{T})) \tag{78}$$

Observe that this objective function can be developed as a second order polynom $g(\gamma) = a\gamma^2 + b\gamma + c$. To find an optimal $\gamma$ it suffices to express coefficients $a$ and $b$ to conclude using Algorithm 2 in Vayer et al. (2018).

Denoting $\mathbf{X}^\top \mathbf{1}_n = \mathbf{q}_X$ and $\mathbf{T}^\top \mathbf{1}_n = \mathbf{q}_T$ and following equation 75, we have

$$\begin{aligned} a &= \langle \mathbf{1}_n(\mathbf{q}_X - \mathbf{q}_T)^\top f_2(\overline{\mathbf{C}})^\top - h_1(\mathbf{C})(\mathbf{X} - \mathbf{T})h_2(\overline{\mathbf{C}})^\top, \mathbf{X} - \mathbf{T} \rangle \\ &= \langle F_1(\mathbf{X}) - F_1(\mathbf{T}) - F_2(\mathbf{X}) + \mathbf{F}(\mathbf{T}), \mathbf{X} - \mathbf{T} \rangle \end{aligned} \tag{79}$$

Finally the coefficient $b$ of the linear term is

$$b = \langle F_1(\mathbf{T}) - F_2(\mathbf{T}), \mathbf{X} - \mathbf{T} \rangle + \langle F_1(\mathbf{X} - \mathbf{T}) - F_2(\mathbf{X} - \mathbf{T}), \mathbf{T} \rangle \tag{80}$$

# F   Appendix of experimental section

We report in the following subsections of this section:

- F.1: implementation details, validation of hyperparameters, datasets and metrics.

- F.2: comparison with COOT clustering.

- F.3: Best trade-off between metrics using t-SNE with $l_2$ symmetrization and UMAP.

- F.4: complete scores on all datasets for all kernels relating to PCA, t-SNE and UMAP.

- F.5: study homogeneity vs silhouette score for various numbers of prototypes.

- F.6: study homogeneity vs k-means score for various numbers of prototypes.

- F.7: Benchmark between Spectral and Kmeans clustering

- F.8: study sensitivity w.r.t the embedding dimension $d$ on spectral methods.

- F.9: computation time study.

- F.10: Proofs of concepts with hyperbolic DR kernels.

## F.1   Experimental setting

**Sequential methods.** We detail in the following the sequential methods DR→C and C→DR considered in our benchmark. DR→C representations are constructed by first running the DR method (Section 2.1) associated with $(L, \mathbf{C}_X, \mathbf{C}_Z)$ thus obtaining an intermediate representation $\widetilde{\mathbf{Z}} \in \mathbb{R}^{N \times d}$. Then, spectral clustering (Von Luxburg, 2007) on the similarity matrix $\mathbf{C}_Z(\widetilde{\mathbf{Z}})$ is performed to compute a cluster assignment matrix $\widetilde{\mathbf{T}} \in \mathbb{R}^{N \times n}$. The final reduced representation in $\mathbb{R}^{n \times d}$ is the average of each point per cluster, *i.e.* the collection of the centroids, which is formally $\mathrm{diag}(\tilde{\mathbf{h}})^{-1} \widetilde{\mathbf{T}}^\top \widetilde{\mathbf{Z}} \in \mathbb{R}^{n \times d}$ where $\tilde{\mathbf{h}} = \widetilde{\mathbf{T}}^\top \mathbf{1}_N$. For C→DR, a cluster assignment matrix $\widehat{\mathbf{T}} \in \mathbb{R}^{N \times n}$ is first computed using spectral clustering on $\mathbf{C}_X(\mathbf{X})$. Then, the cluster centroid $\mathrm{diag}(\hat{\mathbf{h}})^{-1} \widehat{\mathbf{T}}^\top \mathbf{X}$, where $\hat{\mathbf{h}} = \widehat{\mathbf{T}}^\top \mathbf{1}_N$, is passed as input to the DR method associated with $(L, \mathbf{C}_X, \mathbf{C}_Z)$.

**Implementation.** Throughout, the spectral clustering implementation of `scikit-learn` (Pedregosa et al., 2011) is used to perform either the clustering steps or the initialization of transport plans. For all methods, $\mathbf{Z}$ is initialized from *i.i.d.* sampling of the standard Gaussian distribution $\mathcal{N}(0,1)$ and further optimized using `PyTorch`'s automatic differentiation (Paszke et al., 2017) with Adam optimizer (Kingma and Ba, 2014). OT-based solvers are built upon the `POT` (Flamary et al., 2021) library. k-means is performed using the `scikit-learn` (Pedregosa et al., 2011) implementation.

**Validated hyperparameters.** For the SEA and UMAP based similarities, we validated `perplexity` across the set $\{20, 50, 100, 150, 200, 250\}$. For UMAP, we further optimized before learning prototypes, the coefficients $a$ and $b$ involved in the parameterized Student kernel for $\mathbf{C}_Z$ using the algorithm provided by McInnes et al. (2018). For all kernels, the number of output samples $n$ spans a set of 10 values, starting at the number of classes in the data and incrementing in steps of 20. For the computation of $\mathbf{T}$ in DistR (see Section 4.1), we benchmark our Conditional Gradient solver, and the Mirror Descent algorithm whose hyperparameter $\varepsilon$ is validated in the two first values within the set $\{10^i\}_{i=-3}^3$ leading to stable optimization.

**Datasets.** We provide details about the datasets used in our study. For image datasets, we use COIL-20[2] (Nene et al., 1996), MNIST and fashion-MNIST[3] (Xiao et al., 2017). Regarding single-cell genomics datasets,

---

[2]`https://www1.cs.columbia.edu/CAVE/software/softlib/coil-100.php`
[3]taken from Torchvision (Marcel and Rodriguez, 2010).

we rely on PBMC 3k[4] (Wolf et al., 2018), SNAREseq[5] chromatin and gene expression (Chen et al., 2019) and the scRNA-seq dataset[6] from (Zeisel et al., 2015) with two hierarchical levels of label. Dimensions are provided in Table 1 When the dimensionality of a dataset exceeds 50, we pre-process it by applying a PCA in dimension 50, as done in practice (Van der Maaten and Hinton, 2008).

Table 1: Dataset Sizes.

|  | Number of samples | Dimensionality | Number of classes |
|---|---|---|---|
| MNIST | 10000 | 784 | 10 |
| F-MNIST | 10000 | 784 | 10 |
| COIL | 1440 | 16384 | 20 |
| SNAREseq (chromatin) | 1047 | 19 | 5 |
| SNAREseq (gene expression) | 1047 | 10 | 5 |
| Zeisel | 3005 | 5000 | (8, 49) |
| PBMC | 2638 | 1838 | 8 |

**Scores.** The homogeneity and NMI scores are taken from `Torchmetrics` (Detlefsen et al., 2022). The other scores used in the experiments are computed as follows.

*i) Silhouette*: The first step is to do a weighted majority vote and associate a label $\tilde{y}_k$ to each prototype $\mathbf{z}_k$. This label is defined by the $y$ maximizing $\sum_{i \in [\![N]\!]} T_{ik} \mathbf{1}_{y_i = y}$ where $y_i$ is the true class label of the input data point $\mathbf{x}_i$. Then, to incorporate the relative importance $w_j = [\mathbf{h}_Z]_j$ of each prototype $\mathbf{z}_j$, we define the *weighted* mean intra-cluster and nearest-cluster distances that read

$$\forall k \in [\![n]\!], \ a_k(\mathbf{Z}, \mathbf{Y}, \mathbf{w}) = \frac{\sum_{j \in [\![n]\!]} \mathbf{1}_{\tilde{y}_j = \tilde{y}_k} w_j d(\mathbf{z}_j, \mathbf{z}_k)}{\sum_{j \in [\![n]\!]} \mathbf{1}_{\tilde{y}_j = \tilde{y}_k} w_j} \quad \text{and} \quad b_k(\mathbf{Z}, \mathbf{Y}, \mathbf{w}) = \min_{k' \neq k} \frac{\sum_{j \in [\![n]\!]} \mathbf{1}_{\tilde{y}_j = k'} w_j d(\mathbf{z}_j, \mathbf{z}_k)}{\sum_{j \in [\![n]\!]} \mathbf{1}_{\tilde{y}_j = k'} w_j} \tag{81}$$

such that the silhouette coefficient is $s_k(\mathbf{Z}, \mathbf{Y}, \mathbf{w}) = \frac{b_k - a_k}{\max(b_k, a_k)}$ and the final silhouette score reads as

$$S(\mathbf{Z}, \mathbf{Y}, \mathbf{w}) = \sum_{k \in [\![n]\!]} w_k s_k(\mathbf{Z}, \mathbf{Y}, \mathbf{w}). \tag{82}$$

*ii) k-means*: We first run the k-means algorithm on the prototypes $\{\mathbf{z}_k\}_{k \in [\![n]\!]}$ giving us the predicted label $\tilde{y}_k$ for each $k \in [\![n]\!]$. The predicted label for each input data point $i \in [\![N]\!]$ is then given by its prototype's predicted label *i.e.* $\hat{y}_i = \tilde{y}_{\arg\max_k T_{ik}}$. Then we compute the NMI score (Kvålseth, 2017) between $(\hat{y}_i)_{i \in [\![N]\!]}$ and the ground-truth class labels $(y_i)_{i \in [\![N]\!]}$.

### F.2  Comparison with COOT clustering

The CO-Optimal-Transport (COOT) problem, proposed in (Redko et al., 2020), is as follows,

$$\min_{\mathbf{T}_r \in \mathcal{U}(\mathbf{h}_r, \overline{\mathbf{h}}_r)} \min_{\mathbf{T}_c \in \mathcal{U}(\mathbf{h}_c, \overline{\mathbf{h}}_c)} \sum_{ijkl} (X_{ik} - Z_{jl})^2 [\mathbf{T}_r]_{ij} [\mathbf{T}_c]_{kl}, \tag{COOT}$$

where $\mathbf{h}_r \in \Sigma_N$, $\overline{\mathbf{h}}_r \in \Sigma_n$, $\mathbf{h}_c \in \Sigma_p$ and $\overline{\mathbf{h}}_c \in \Sigma_d$. One can seek to optimize the above objective with respect to $\mathbf{Z}$ to obtain a competitor method to DistR. This problem is called COOT clustering in Redko et al. (2020). In the latter, $\mathbf{T}_r$ then plays the role of a soft clustering matrix of the rows of $\mathbf{X}$ while $\mathbf{T}_c$ can be seen as a soft clustering matrix of its columns. The above is thus a linear DR model.

In Table 2, we display the homogeneity values obtained with COOT along with the methods described Section 5. Precisely, it measures to what extent the clustering given by $\mathbf{T}_r$ groups points with the same ground truth label. One can notice that COOT falls short compared to its competitors that leverage affinity matrices as in state-of-the-art (non-linear) DR methods.

### F.3  Best trade-off between metrics using t-SNE and UMAP

---

[4]downloaded from Scanpy (Wolf et al., 2018).
[5]https://github.com/rsinghlab/SCOT
[6]https://github.com/solevillar/scGeneFit-python

| $\mathcal{Z}$ | methods | $\mathbf{C}_X$ / $\mathbf{C}_Z$ | COIL | MNIST | FMNIST | PBMC | SNA1 | SNA2 |
|---|---|---|---|---|---|---|---|---|
| $\mathbb{R}^2$ | DistR (ours) | SEA / St. | 99.50 (0.60) | 97.80 (0.00) | 93.40 (0.00) | 97.10 (0.00) | 100.00 (0.00) | 100.00 (0.00) |
| | DR→C | - | 98.10 (0.50) | 93.30 (3.40) | 94.30 (2.80) | 96.30 (0.90) | 100.00 (0.00) | 100.00 (0.00) |
| | C→DR | - | 100.00 (0.00) | 97.80 (0.00) | 93.40 (0.00) | 97.10 (0.00) | 100.00 (0.00) | 100.00 (0.00) |
| | COOT | NA | 43.90 (0.70) | 9.80 (3.00) | 8.50 (0.90) | 15.90 (1.90) | 44.00 (4.60) | 49.70 (8.60) |
| $\mathbb{R}^{10}$ | DistR (ours) | $\langle,\rangle_{\mathbb{R}^p}$ / $\langle,\rangle_{\mathbb{R}^d}$ | 96.80 (0.70) | 97.00 (0.90) | 93.40 (0.00) | 97.10 (0.40) | 80.50 (0.00) | 100.00 (0.00) |
| | DR→C | - | 73.30 (1.80) | 98.30 (3.40) | 93.20 (1.90) | 90.20 (1.40) | 72.70 (6.00) | 90.00 (20.00) |
| | C→DR | - | 83.70 (0.00) | 100.00 (0.00) | 93.40 (0.00) | 93.30 (0.00) | 80.50 (0.00) | 100.00 (0.00) |
| | COOT | NA | 45.50 (1.60) | 13.70 (2.10) | 9.30 (2.80) | 16.10 (2.40) | 45.60 (5.90) | 76.50 (16.60) |

Table 2: Best homogeneity scores for $n$ validated in a span up to 200 with increments of 20.

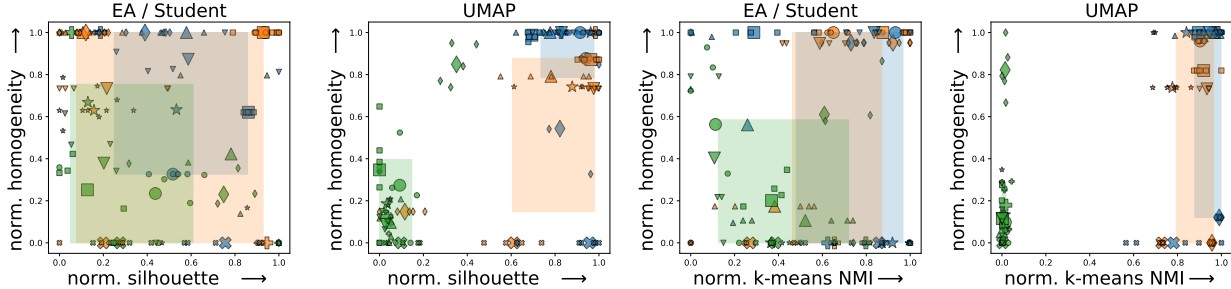

Figure 6: Complements to Figure 3 of the main paper for t-SNE with $l_2$ symmetrization and UMAP : Best trade-off between homogeneity vs silhouette (2 first plots), and homogeneity vs NMI (2 last plots). Scores are normalized in $[0, 1]$ via min-max scaling over a dataset. Small markers are scores per dataset for 5 runs, while big ones are their mean. We illustrate the 20-80% percentiles of scores per method as a colored surface. .

### F.4 Complete scores

We complete the results shown in Section 5 by providing the scores obtained on all datasets and models. Scores are plotted in Figure 7 for the PCA model, in Figure 8 for the t-SNE model with entropic symmetrization, then in Figure 9 with $l_2$ symmetrization, and Figure 10 for the UMAP model.

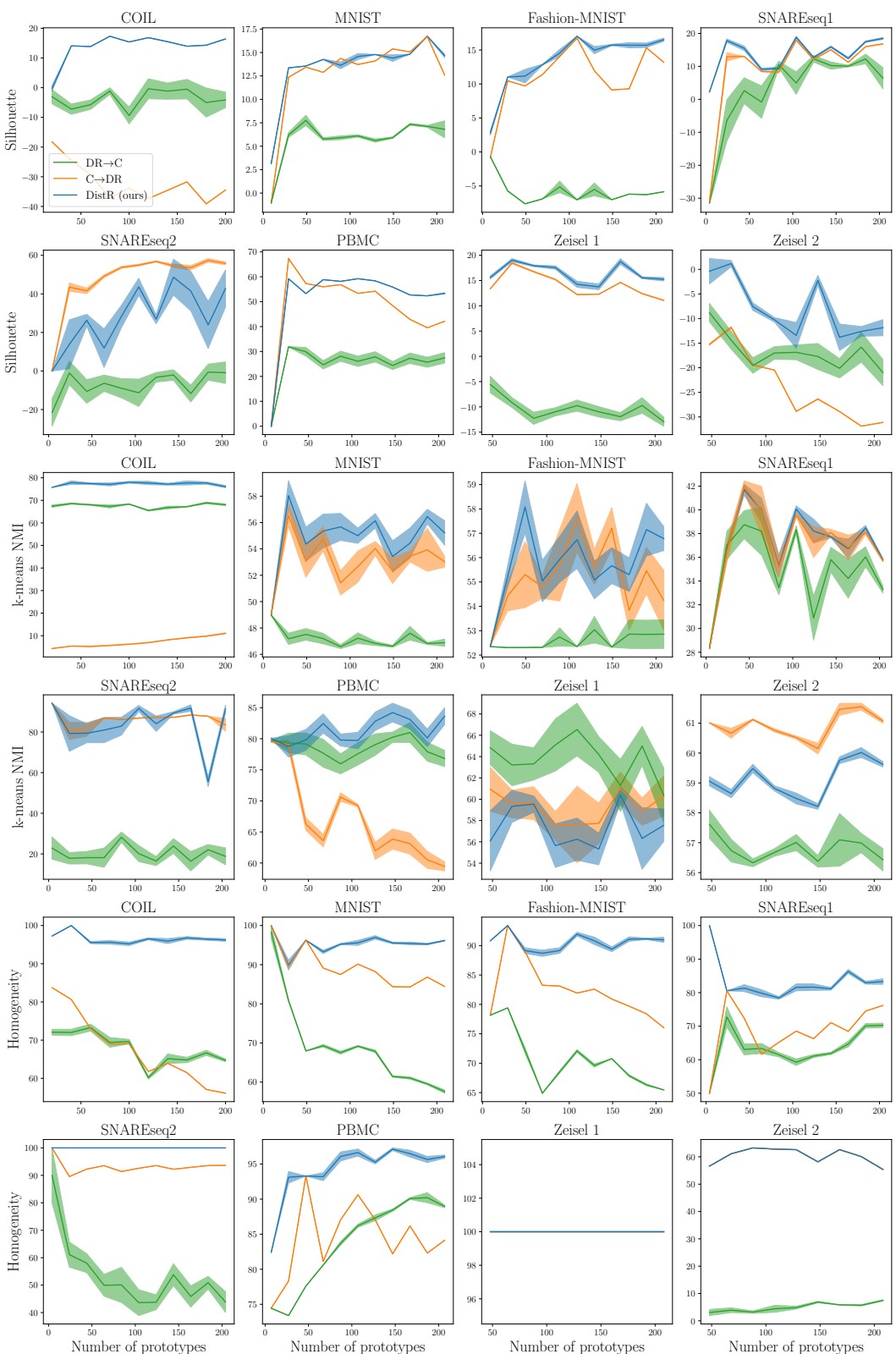

Figure 7: Scores ($\times 100$) with respect to the number of prototypes (in $\mathbb{R}^{10}$) produced by DistR using the PCA model: $\langle , \rangle_{\mathbb{R}^p}$ similarity for $\mathbf{C}_X$ (Van Assel et al., 2023), $\langle , \rangle_{\mathbb{R}^d}$ for $\mathbf{C}_Z$ and loss $L_2$.

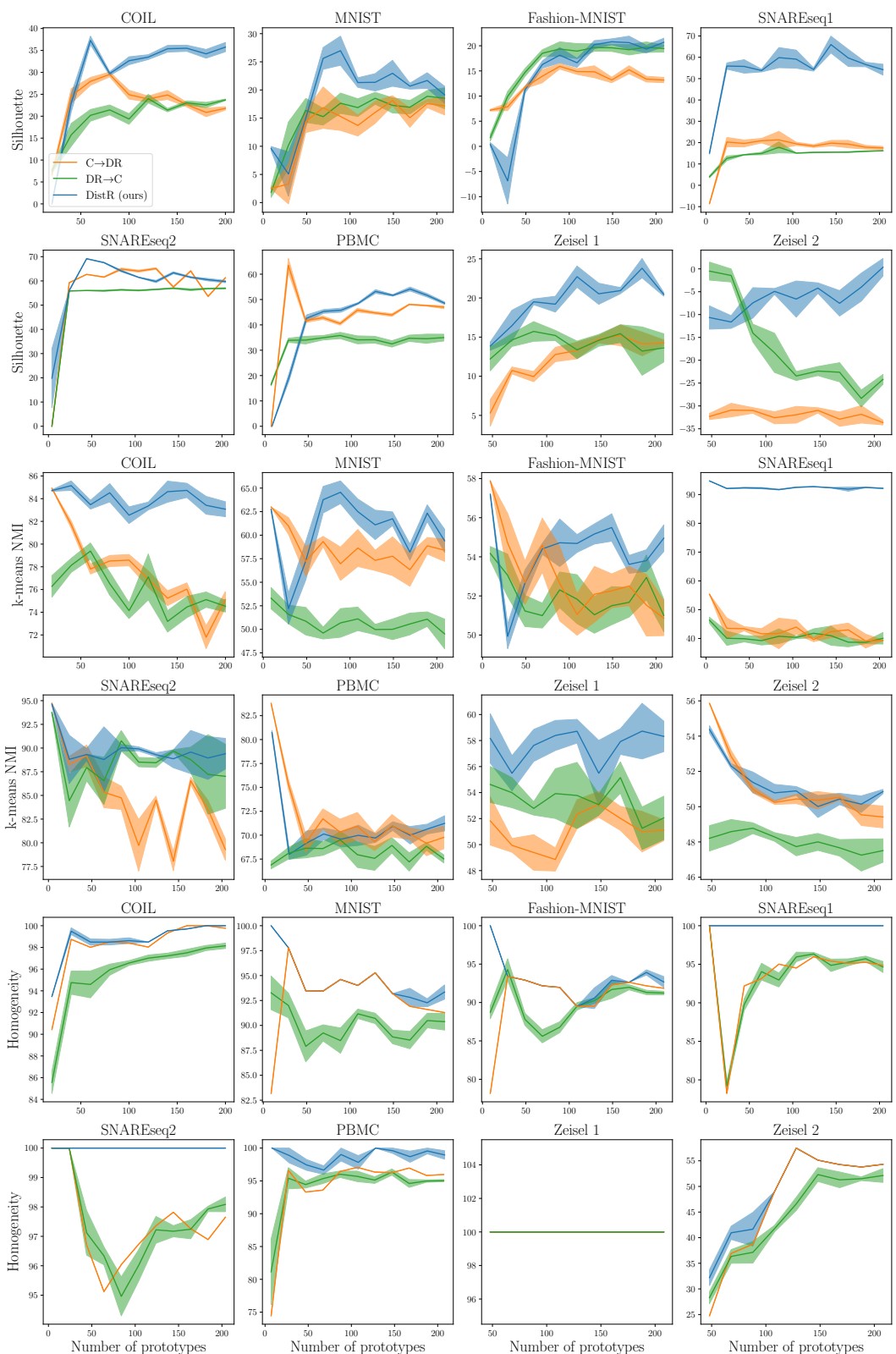

Figure 8: Scores ($\times 100$) with respect to the number of prototypes (in $\mathbb{R}^2$) produced by DistR using the t-SNE model: SEA similarity for $\mathbf{C}_X$ (Van Assel et al., 2023), Student's kernel for $\mathbf{C}_Z$ and loss $L_{\mathrm{KL}}$.

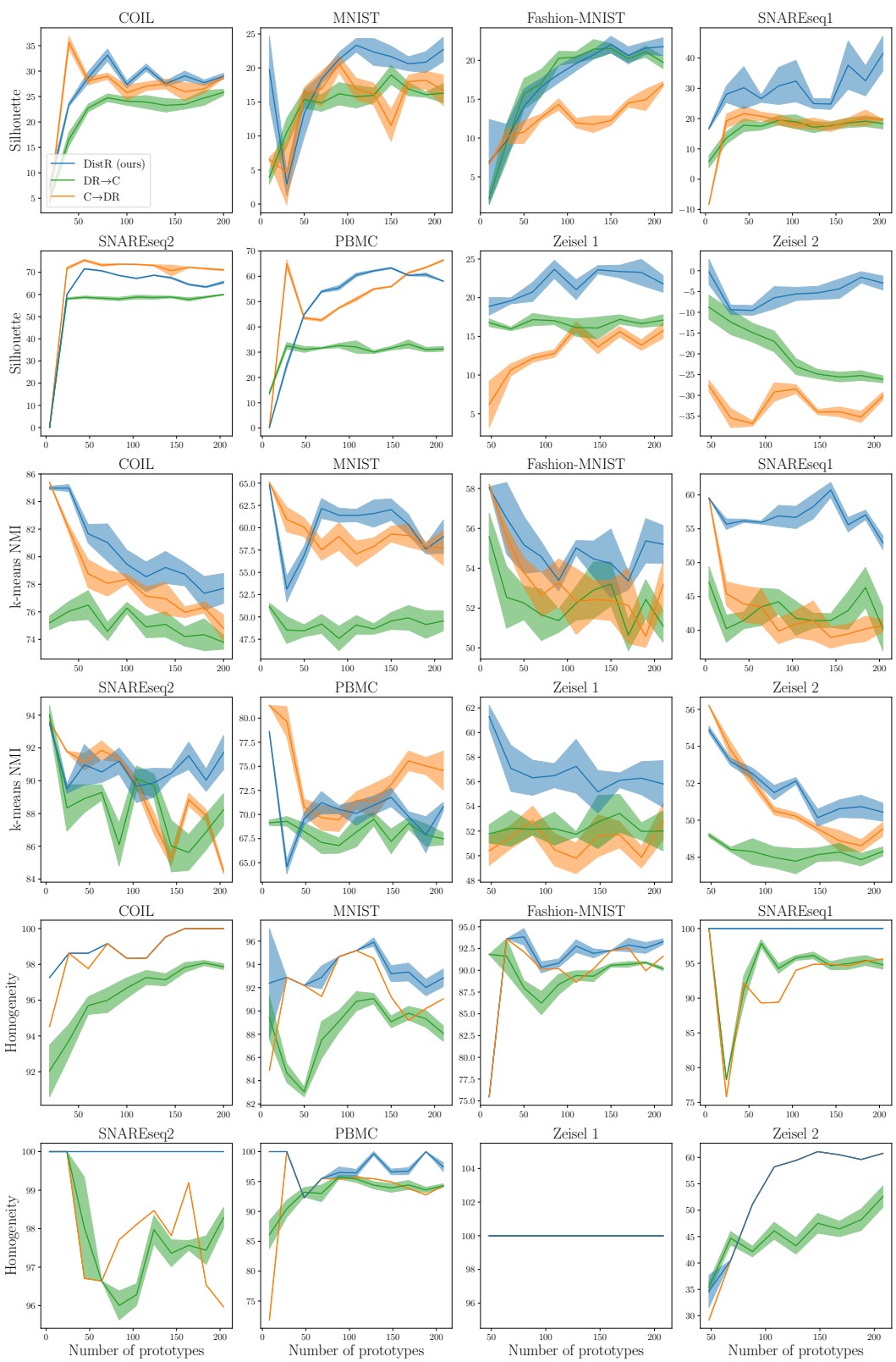

Figure 9: Scores ($\times 100$) with respect to the number of prototypes (in $\mathbb{R}^2$) produced by DistR using the t-SNE model with $l_2$ symmetrization: EA similarity for $\mathbf{C}_X$ (Van der Maaten and Hinton, 2008), Student's kernel for $\mathbf{C}_Z$ and loss $L_{\mathrm{KL}}$.

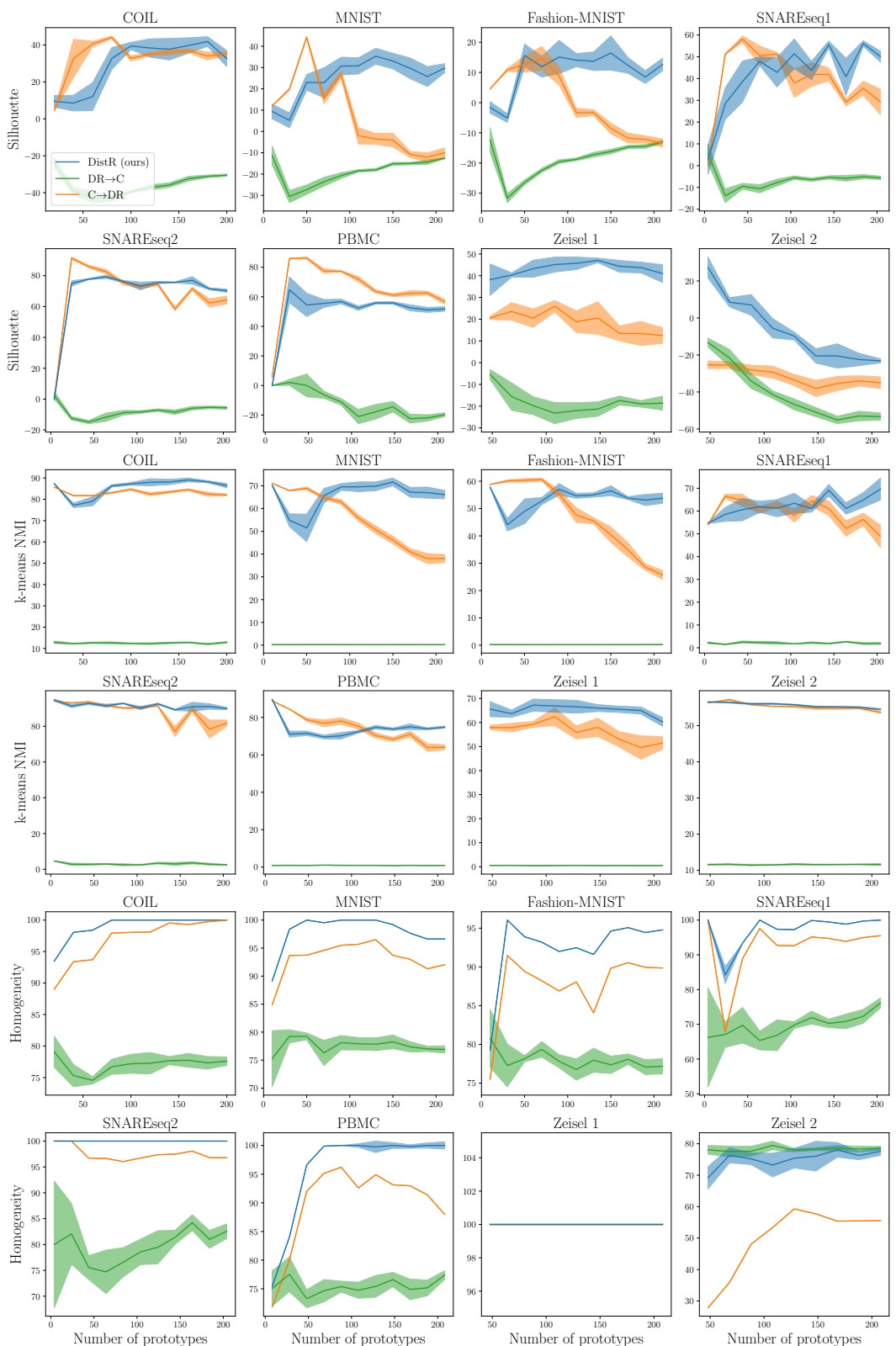

Figure 10: Scores ($\times 100$) with respect to the number of prototypes (in $\mathbb{R}^2$) produced by DistR using the UMAP model: t-conorm of the smoothed nearest neighbors $v_{j|i} = e^{-\|\mathbf{x}_i - \mathbf{x}_j\|_2 - \rho_i / \sigma_i}$ for $\mathbf{C}_X$ (McInnes et al., 2018, Eq.16), parameterized Student's kernel for $\mathbf{C}_Z$ (McInnes et al., 2018, Eq.17) and loss $L_{BCE}$.

**F.5 Dynamics between homogeneity and silhouette scores across numbers of prototypes**

For each method depending on some hyperparameters, models performing the best on average across all numbers of prototypes are illustrated in Figure 11 for spectral methods and in Figure 12 for neighbor embedding ones.

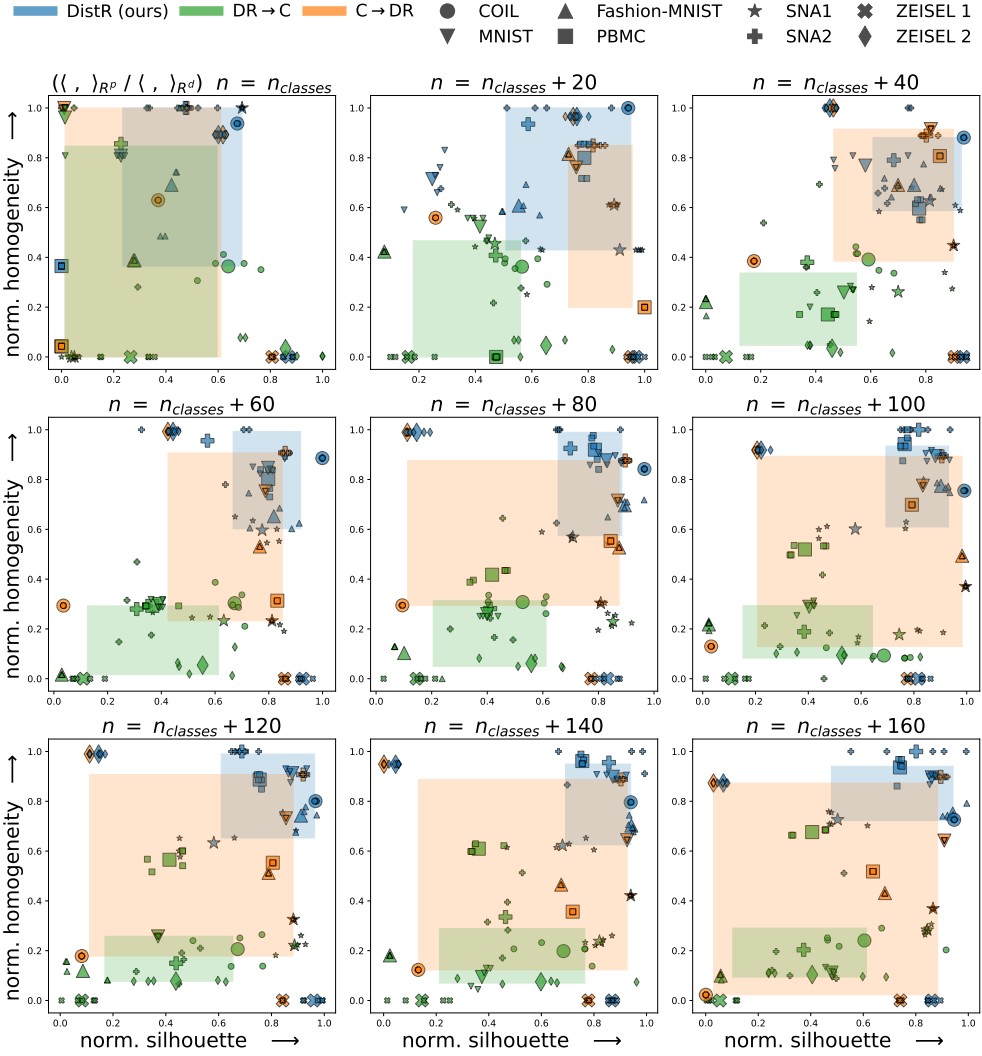

Figure 11: Trade-off between homogeneity vs silhouette scores using PCA model across various numbers of prototypes $n$. The illustration follows the same principal than fig. 3

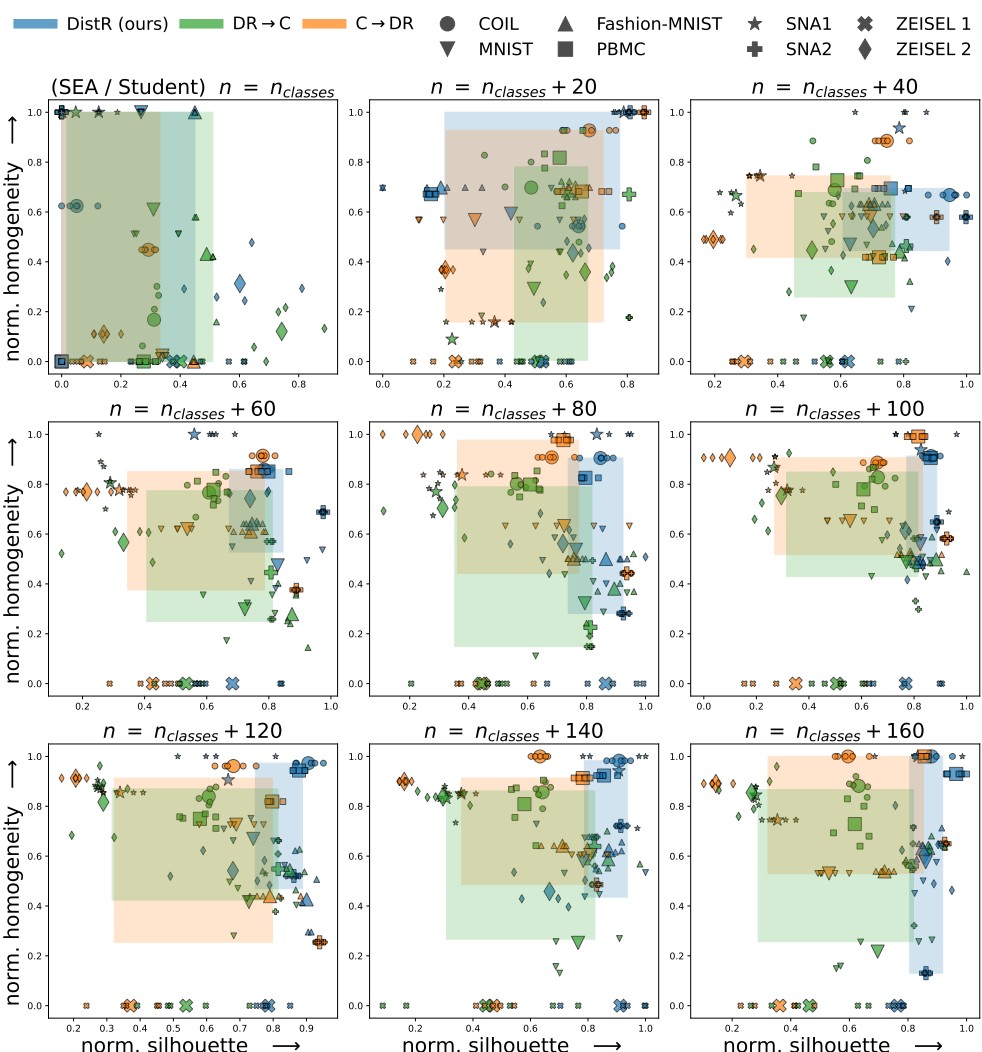

Figure 12: Trade-off between homogeneity vs silhouette scores using t-SNE model across various numbers of prototypes $n$. The illustration follows the same principal than fig. 3

## F.6   Dynamics between homogeneity and NMI scores across numbers of prototypes

For each method depending on some hyperparameters, models performing the best on average across all numbers of prototypes are illustrated in Figure 13 for spectral methods and in Figure 14 for neighbor embedding ones.

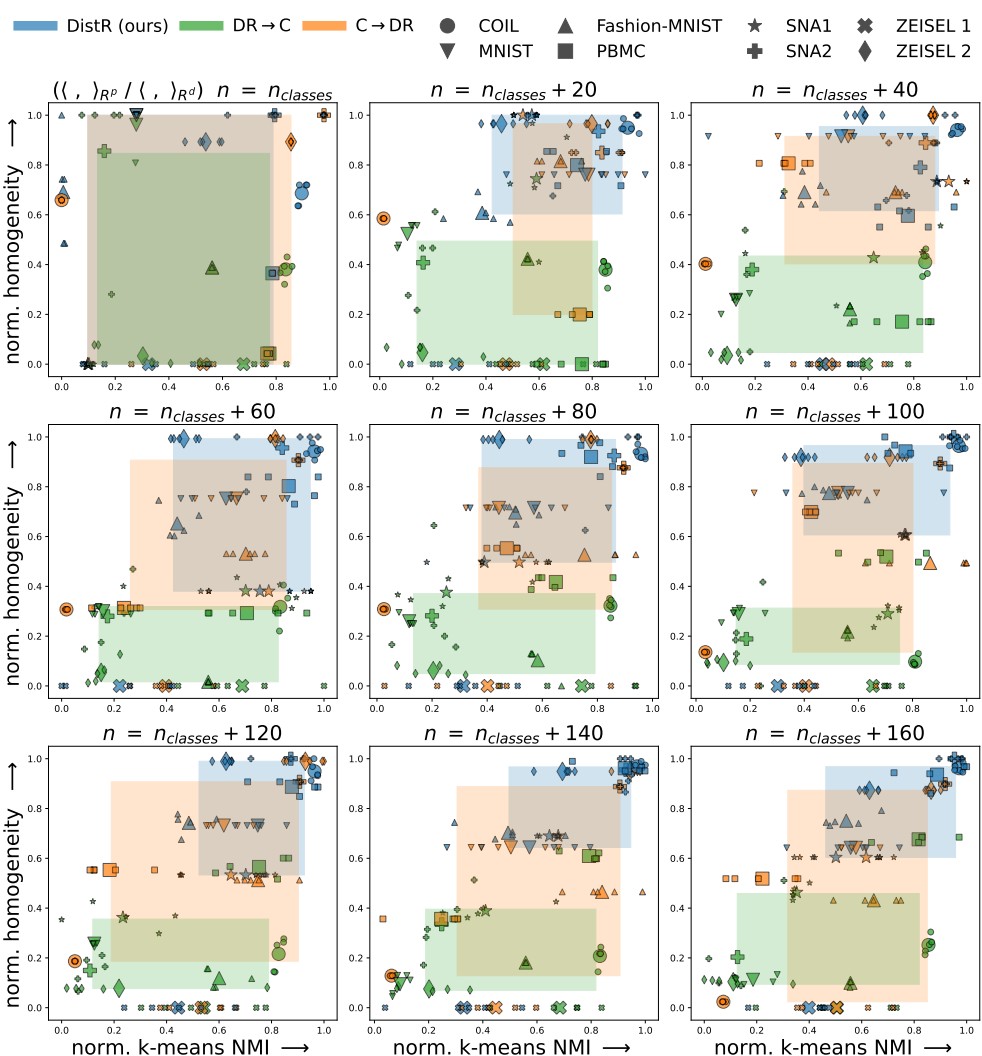

Figure 13: Trade-off between homogeneity vs silhouette scores using PCA model across various numbers of prototypes $n$. The illustration follows the same principal than fig. 3

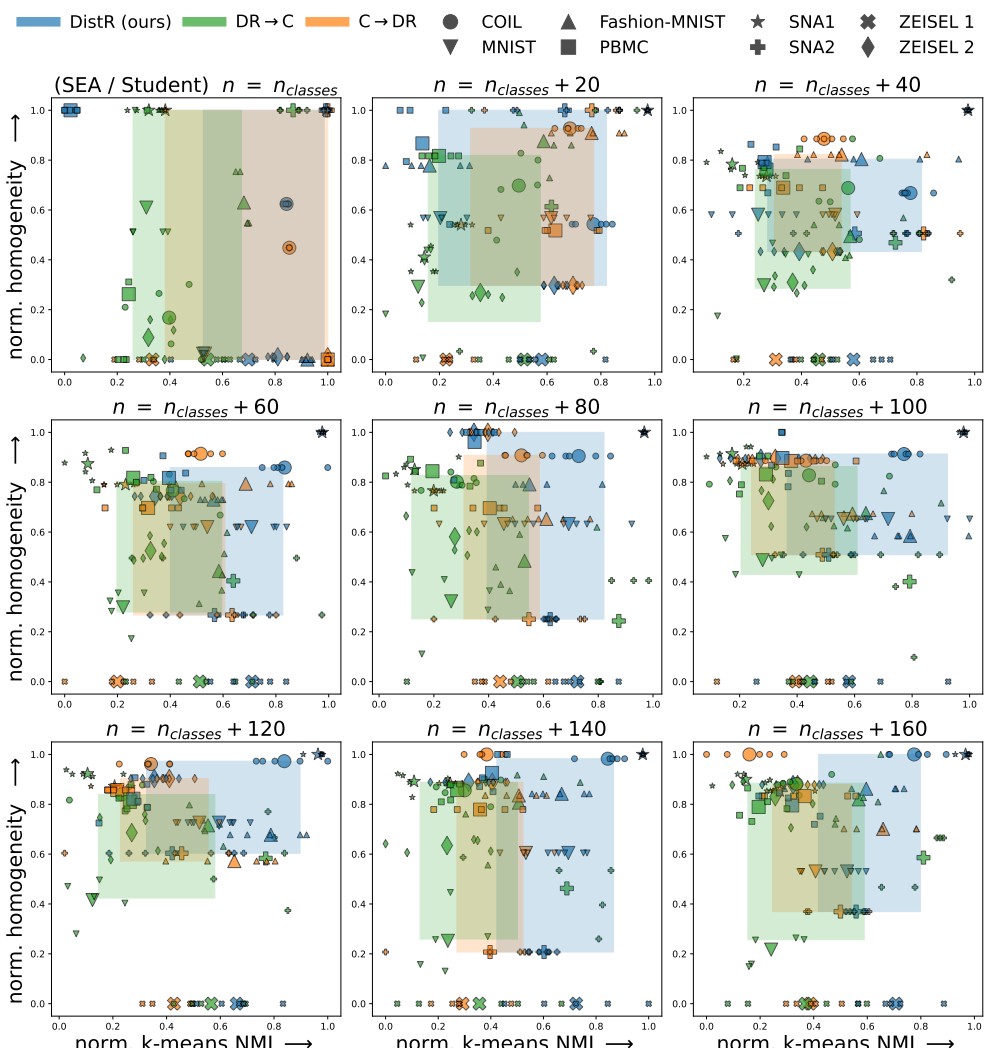

Figure 14: Trade-off between homogeneity vs silhouette scores using t-SNE model across various numbers of prototypes $n$. The illustration follows the same principal than fig. 3

## F.7 Benchmark between Spectral and Kmeans clustering

We compare in the following the performances of the joint DR/clustering algorithm across different clustering strategies. Namely for $C \rightarrow DR$ and $DR \rightarrow C$, we compare our default choice of the Spectral Clustering (SC) algorithm with a Kmeans algorithm. Then for DistR, we compare the use of both algorithms to initialize the transport plans. Performances for both PCA and t-SNE kernels with KL projection are reported in Figure 15. On average across datasets, we can observe that DistR (SC), the default for the method, achives a better trade-off w.r.t the silhouette and Kmeans NMI scores than DistR (Kmeans), while maintaining similar homogeneity scores. A similar behavior is observed for C→DR using PCA like kernels, whereas overall higher performances are observed for C→DR (SC) than C→DR (Kmeans). Therefore SC appears as a better clustering strategies to maximize performances for both DistR and C→DR, where DistR (SC) remains the most competitive methods. Finally, DR→C (Kmeans) consistently leads to higher homogeneity scores with comparable silhouette scores than DR→C (SC), but their respective trade-off between homogeneity and Kmeans NMI are more erratic.

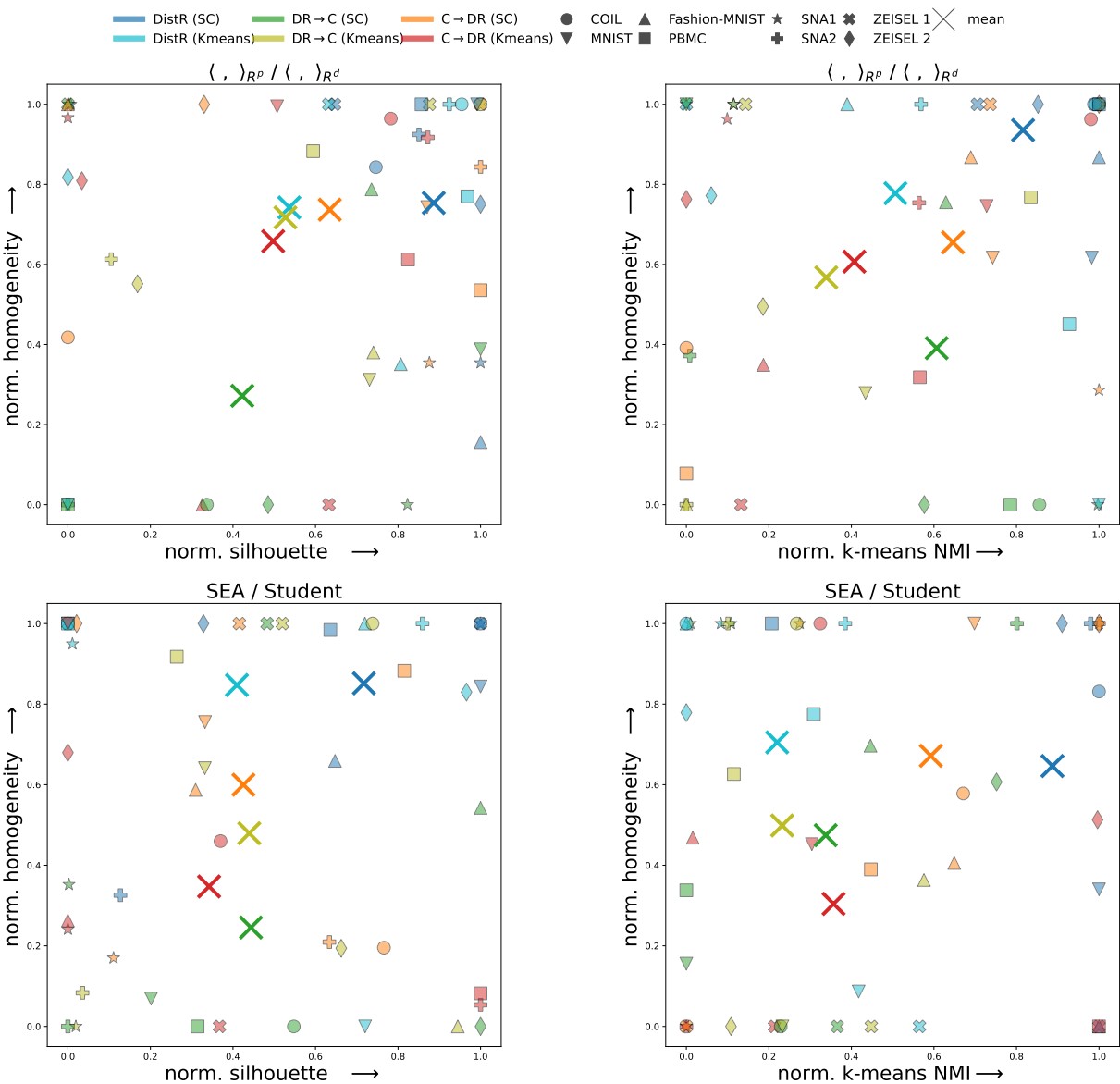

Figure 15: Best trade-off between homogeneity vs silhouette (left), and homogeneity vs NMI (right). Scores are normalized in $[0, 1]$ via min-max scaling over a dataset. Small markers represent scores averaged over 5 runs for a given dataset, while big ones are their mean over datasets. The illustration follows the same principal than fig. 3

## F.8 Sensitivity analysis w.r.t the embedding dimension

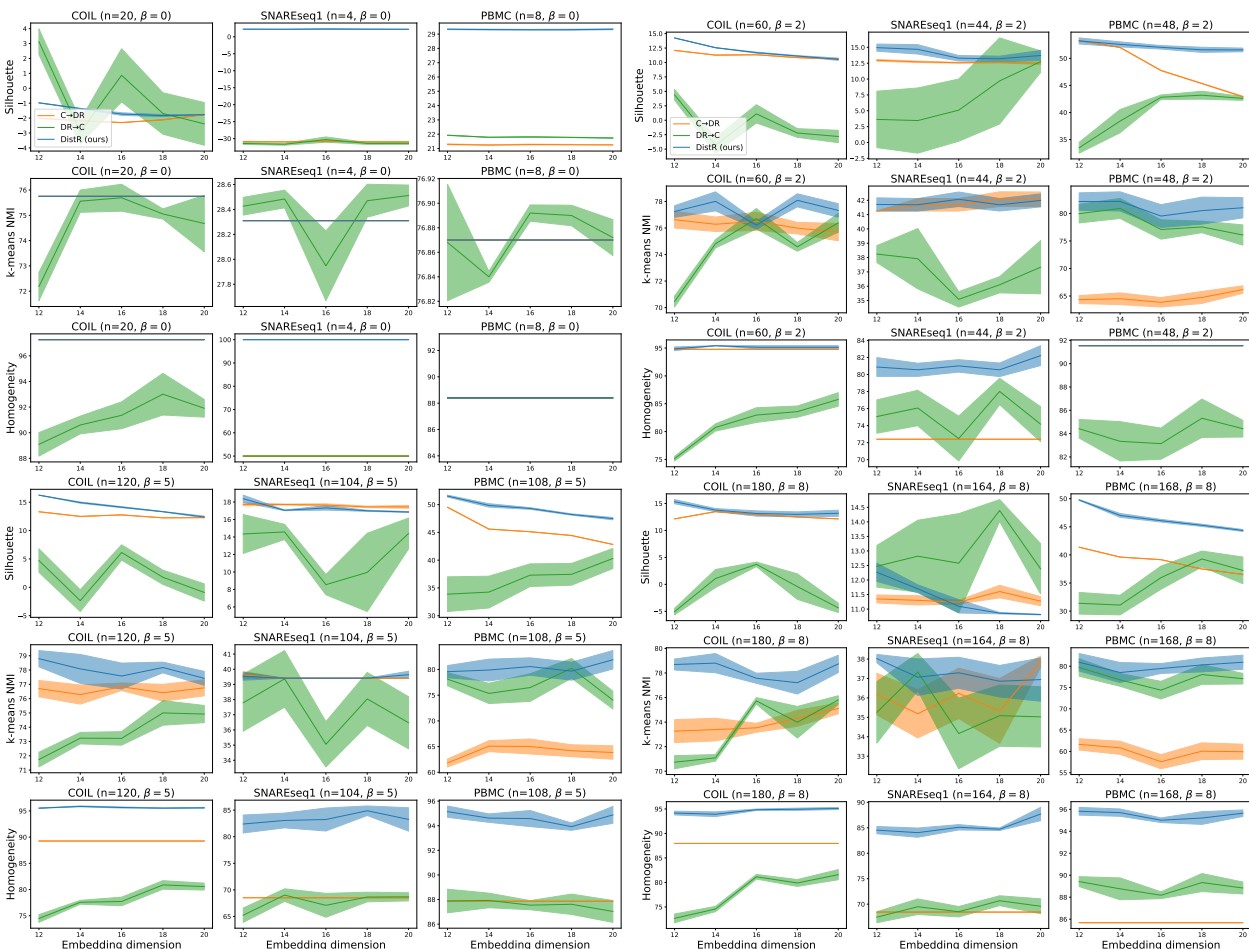

Figure 16: Sensitivity analysis w.r.t the embedding dimension $d$ for the spectral method analog to PCA with $C_X(\mathbf{X}) = \mathbf{X}\mathbf{X}^\top$ and $C_Z(\mathbf{Z}) = \mathbf{Z}\mathbf{Z}^\top$. Over three datasets, we make vary the embedding dimension from $d \in \{12, 14, ..., 20\}$.

.

## F.9 Computation time comparison

We compare in the following the computation time for all methods benchmarked in Table 1 when using a spectral method and a SNE method. For fair comparison across methods, we do not apply here the low-rank factorizations used for experiments of the main paper. All experiments were done on a server using a GPU (Tesla V100-SXM2-32GB) and composed of 18 cores Intel(R) Xeon(R) Gold 6240 CPU @ 2.60GHz. All DR steps benefit from our GPU compatible implementation, while spectral clustering (SC) was performed on CPU using scikit-learn implementation running on CPUs.

Notice that to run our experiments we precomputed and saved SC steps for the maximum number of prototypes ran on the input structure $C_X(\mathbf{X})$, for all benchmarked methods e.g CDR used for clustering or DistR and COOT used for initialization of the transport plans. As DRC performs SC over learned embeddings using $C_Z(\mathbf{Z})$ it cannot be precomputed, so we include the time of performing SC for all $n$ for all methods to achieve a fair comparison. Results for three datasets of increasing sample sizes SNA1, ZEISEL and MNIST (See Appendix E.1) are reported in Figure 18 with pre-computation and in Figure 17 without.

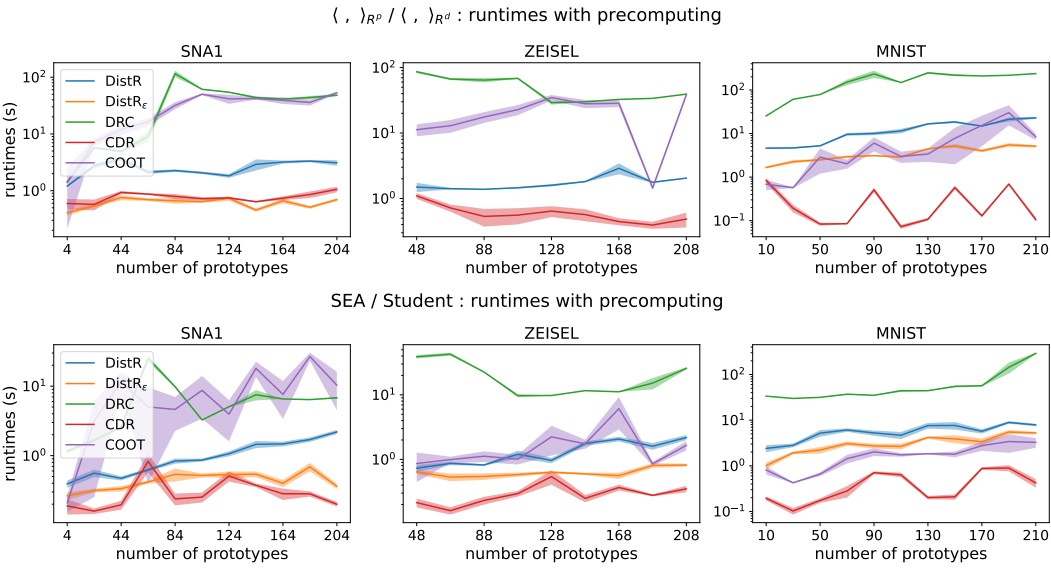

Figure 17: Computation time comparison depending on the number of prototypes $n$ for all methods over 3 datasets, for 5 different initializations, while precomputing spectral clustering on the input structure $C_X(\mathbf{X})$.

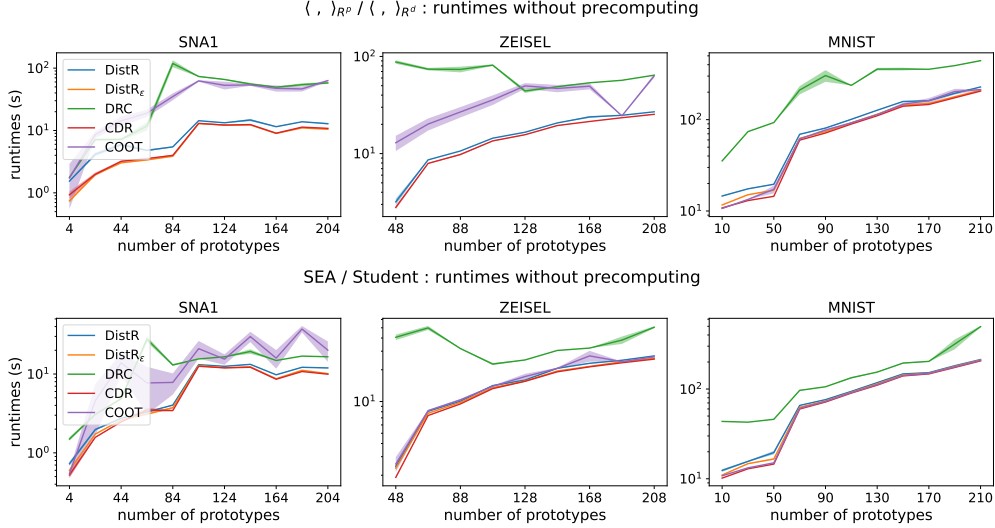

Figure 18: Computation time comparison depending on the number of prototypes $n$ for all methods over 3 datasets, for 5 different initializations, without precomputing spectral clustering on the input structure $C_X(\mathbf{X})$.

Overall our DistR models are competitive with all benchmarked methods in terms of computation time too and can be run (and further validated) in a few seconds using a GPU after precomputing the spectral embeddings for medium size dataset (N=10000 for MNIST), by efficiently leveraging low-rank kernels often used in the DR literature.

### F.10 Proofs of concepts with hyperbolic kernels

Hyperbolic spaces (Chami et al., 2021; Fan et al., 2022; Guo et al., 2022; Lin et al., 2023) are of particular interest as they can capture hierarchical structures more effectively than Euclidean spaces and mitigate the curse of dimensionality by producing representations with lower distortion rates. For instance, Guo et al. (2022) adapted t-SNE by using the Poincaré distance and by changing the Student's t-distribution with a more general hyperbolic Cauchy distribution. Notions of projection subspaces can also be adapted, *e.g.* Chami et al. (2021) use horospheres as one-dimensional subspaces. To match our experiments with the neighbor embeddings in Euclidean settings, we adapt the *Symmetric Entropic Affinity* (SEA) from Van Assel et al. (2023) for $\mathbf{C}_X$ and the scalar-normalized student similarity for $\mathbf{C}_Z$ (Van der Maaten and Hinton, 2008), by simply changing the Euclidean distance by an hyperbolic distance.

**Implementation details.**  Computations in Hyperbolic spaces are done with `Geoopt` (Kochurov et al., 2020) and the RAdam optimizer (Bécigneul and Ganea, 2018) replaces Adam. A Wrapped Normal distribution in Hyperbolic spaces (Nagano et al., 2019) is used to initialize $\mathbf{Z}$ in the hyperbolic setting. All the computations were conducted in the Lorentz model (Nickel and Kiela, 2018), which is less prone to numerical errors. We used the distance function from Nickel and Kiela (2018) to form $\mathbf{C}_Z$. Specifically, we employed a scalar-normalized Gaussian kernel, where the distances between points are computed as in Nickel and Kiela (2018, Equation 1). After optimization, results are projected back to the Poincaré ball for visualization purposes. In this hyperbolic context, we adopted the formulation of Guo et al. (2022) which generalizes Student's t-distribution by Hyperbolic Cauchy distributions (denoted as H-Student in the results). Notice that Guo et al. (2022) considered weighted sums of DR objective depending respectively on $L_2$ and $L_{KL}$, including 2 additional hyperparamaters, plus various validated curvature levels for the inner hyperbolic distances. In the following experiments, we only kept $L_{KL}$ for comparison with the Euclidean SNE-based methods illustrated in Section 5 and previous Sections of F, while validating the same hyperparameters and setting the space curvature to 1. The silhouette score was adapted to this kernel considering the Hyperbolic distance instead of the Euclidean one, and we implemented a Hyperbolic Kmeans whose barycenters are estimated using the RAdam optimizer to compute the NMI scores.

**Results.**  We first report in Figure 19 a relative comparison of the best trade-off between local and global metrics achieved by all methods. Similarly to visualizations of the main paper, we considered for each method and dataset, the model maximizing the sum of the two normalized metrics to account for their different ranges. DistR, being once again present on the top-right of all plots, provides on average the most discriminant low-dimensional representations endowed with a simple geometry, seconded by C→DR.

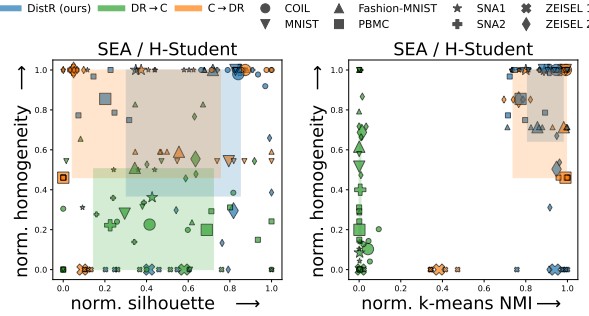

Figure 19: Best trade-off between homogeneity vs silhouette (2 first plots), and homogeneity vs NMI (2 last plots). Scores are normalized in $[0, 1]$ via min-max scaling over a dataset. Small markers represent scores for 5 runs for a given dataset, while big ones are their mean. For each method we illustrate the 20-80% percentiles of normalized scores as a colored surface.

Then we report in Figure 20 absolute performances for all methods and all datasets across various $n$, as done in Appendix F.4. We can observe that both DistR and C→DR achieve fairly high NMI and homogeneity scores across all settings, while DistR performs significantly better on average across the tested number of prototypes

*n*. However, DR→C struggles significantly to learn both globally and individually discriminant prototypes. Notice that DR→C's homogeneity scores are significantly lower on average than both benchmarked Euclidean kernels. This mitigates drastically the significance of the silhouette scores computed for this method, letting essentially DistR and C→DR to compare. Even though DistR outperforms consistently C→DR w.r.t silhouette scores, these scores remain significantly lower than with the other t-SNE kernels. As the latter is equivalent to a null curvature, this indicates that further fine-tuning of the curvature within these hyperbolic kernels could be beneficial. Nevertheless, these results confirm the versatility of our DistR approach, capable of operating on non-Euclidean geometries.

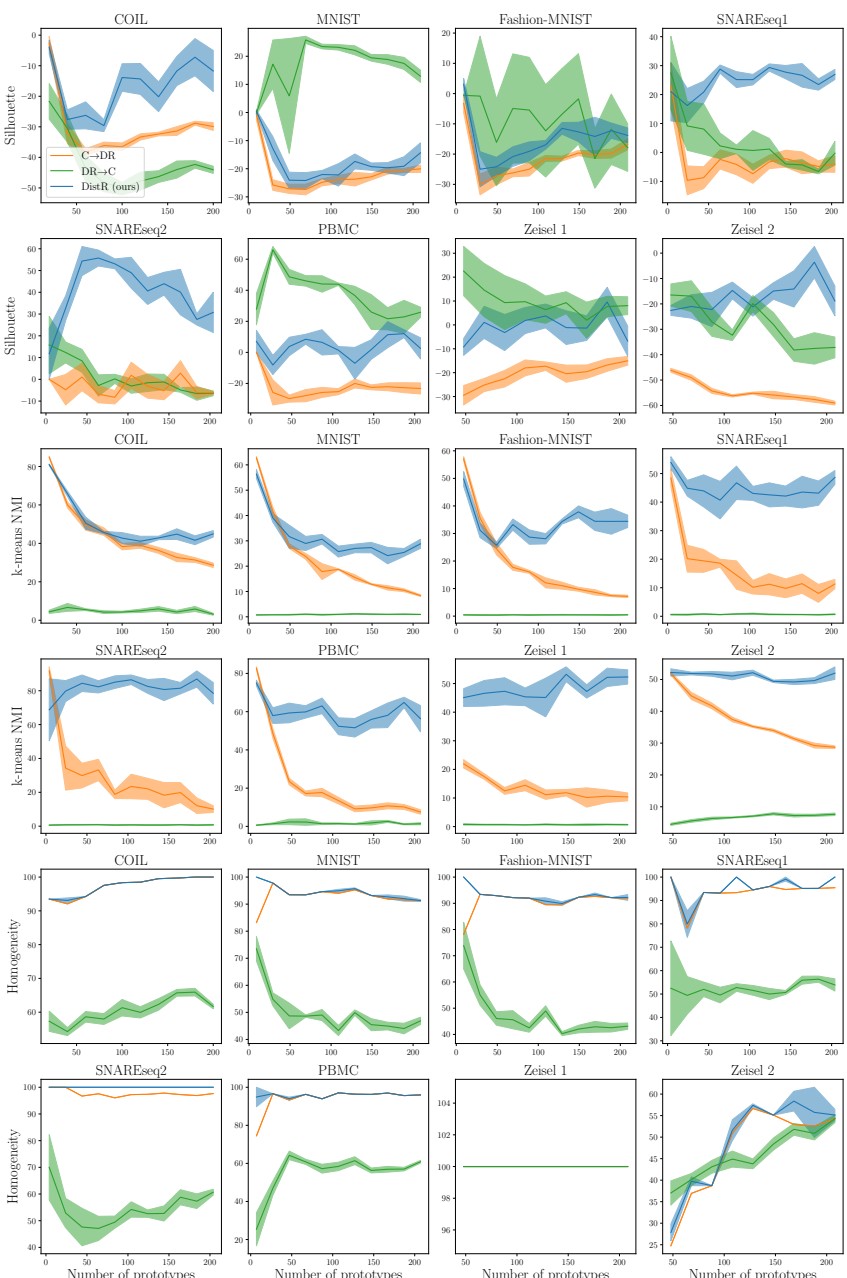

Figure 20: Scores ($\times 100$) with respect to the number of prototypes (in $\mathbb{R}^{10}$) produced by DistR using the Hyperbolic Student model.

