# OpenReview forum: "Distributional Reduction: Unifying Dimensionality Reduction and Clustering with Gromov-Wasserstein"
_TMLR — Accepted by TMLR_

### Review · Reviewer_YTGR · 2025-02-20

**Summary Of Contributions:**

This paper introduces a new framework, Distributional Reduction (DistR), which is able to integrate dimension reduction and clustering together based on the Gromov-Wasserstein (GW) optimal transport problem. This proves the researchers a new perspective, trying to jointly learn the low-dimensional embeddings and representative prototypes. The paper provides formal proofs establishing connections between GW transport, spectral embeddings, and prototype-based clustering. It used the block coordinate descents algorithm for the semi-relaxed GW divergence computation to guarantee convergence to the local optimum. From the evaluation perspective, the authors have compared eight different datasets to demonstrate its effectiveness compared to the DR->C and C->DR methods.

**Audience:**

Yes

**Broader Impact Concerns:**

I think the ethical claims that the authors might need to made for the broader impact statement would be mostly from the detection of the typical prototypes which is also the reason why I requested for a few changes for the large datasets.

**Claims And Evidence:**

Yes

**Requested Changes:**

To strengthen the paper and make it more compelling for both theoretical and practical applications, here are a few changes that I would like to recommend the authors to do.

- Improve Scalability Validation with Large Datasets

While the paper discusses low-rank factorization as a way to improve efficiency, there is no empirical validation demonstrating that DistR can scale to large datasets. The current experiments use a maximum of 10,000 samples, which is significantly smaller than datasets typically used in dimensionality reduction and clustering benchmarks. It would be better to conduct scalability experiments on larger datasets (e.g., 100,000+ samples) to demonstrate that DistR remains computationally feasible, and provide runtime vs. dataset size comparisons, similar to what is done in existing DR methods papers. Empirically validate the efficiency of low-rank factorization, showing how much computational and memory savings it provides in practice.

- Improving the convincing of the learned prototypes

The paper claims that DistR’s prototypes offer meaningful and structured representations, but no clear evidence is provided to confirm their interpretability. The homogeneity metric is used to evaluate the prototypes, but it does not prove that the prototypes represent distinct, meaningful classes rather than being artifacts of optimization. It would be better to conduct a stability analysis and an analysis on intra-clusters. For the stability analysis, I am wondering if the author can provide an evaluation on whether learned prototypes remain consistent across multiple training runs. For the intra-cluster analysis, aside from Figure 4, I am wondering if the authors can provide an analysis on the distribution of the intra-cluster variance to see if the clusters are truly distinct. This can also helps for explaining why the DistR would go over the true class labels in some sense.

- Provide Clearer Guidelines for Hyperparameter Selection

I would recommend the authors to discuss more on the hyperparameter sensitivity analysis explanations and the corresponding guidelines for how to select a proper hyperparameter with the metrics curve given. In the real world application, the instability of the trend in some of the datasets might lead to the confusion of the users. It would be better if the authors can have some discussion on this hyperparameter selection topic, so that the users are more capable of accepting this idea for combining the dimension reduction and clustering in the same place.

**Strengths And Weaknesses:**

## Strength

- Novel Theoretical Framework

The paper presents a new perspective that integrates dimensionality reduction (DR) and clustering into a single optimization problem using Gromov-Wasserstein (GW) optimal transport. The proposed framework is theoretically well-founded, with formal theorems and proofs that establish its validity. The framework can recover traditional DR methods (e.g., PCA, UMAP, t-SNE) and clustering methods (e.g., k-means, spectral clustering), making it highly flexible.

Different from the sequential approaches, DistR learns both simultaneously, so that everything will be optimized at the same time. The kind of idea is better in identifying representative prototypes in a way that reduces information loss, potentially leading to more meaningful cluster assignments.

- Efficient Optimization Approach

The paper uses an efficient computational strategy: Block Coordinates Descent (BCD) Optimization that allows for stable training. It uses semi-relaxed GW Divergence (srGW) that improves computational feasibility, making the problem more tractable.

- Interesting perspective for working on multi-view learning

In the Conclusion section, the authors explicitly mention that their approach naturally enables addressing multi-view settings with multiple unaligned inputs of different sizes​. The framework’s versatility allows for addressing multi-view settings, accommodating multiple unaligned inputs of different sizes. This seems to be a new direction for unsupervised learning on multi-views.

## Weakness

- Lack of scalability evidence for large datasets.

While there are couple of experiments for the scalability problem for different number of prototypes, there lacks sufficient evidences for DistR to applied in the high number of samples. Based on the Table 1 from Appendix F.1, the maximum number of samples used for validation is 10,000 samples. This is away smaller than most of the dimension reduction methods. For instance, the original MNIST Datasets contains 60,000 training points, and the 10,000 testing points, so I am wondering how the computational time would increase as the training dataset grows up (same for F-MNIST). I am interested to see how the relationship between the number of samples and the runtime would be like. Even though the authors have discussed about the low-rank factorization in their main paper, there is no empirical experiments to show how efficient it would improve.

- The interpretability and explainability of the prototypes

In the paper, the authors mentioned that the prototypes that DistR have offered is a discriminative representation of the input samples at various granularities, and from the numerical experiments perspective, they are trying to use the homogeneity as a kind of explanations. However, the metrics seems to be still unclear to show whether the prototypes are meaningful or just artifacts of optimization. A higher number of fixed prototypes generally increases the likelihood of achieving higher homogeneity, as each prototype can better capture distinct class structures, reducing class mixing within clusters. When you increase the number of prototypes, you allow for finer granularity in clustering, meaning each class can be better separated. If the number of prototypes significantly exceeds the number of true classes, homogeneity might not continue to improve, as some prototypes could start capturing sub-clusters within a single class, potentially reducing purity. Since the subclusters numbers are hard to evaluate at this point, I am a bit concerned about if those metrics are proving the prototypes are meaningful. Moreover, Figure 4 is showing the images that are medoids for each cluster, which is the argmax point for each cluster instead of the standard deviation or similarity within the cluster, so I am doubt that if the diversity within each group shrink (for example of the standard deviation within one subcluster, and the standard deviation between two cluster I assume should have statistically difference between each other?) This problem might lead to a doubt in showing whether the prototypes are just artifacts of the optimization or not?

- The stopping criteria/hyperparameter selection for the DistR

Based on the results shown within the Figure 5 and also for the Appendix F.4, the results for the DistR varies a lot when the number of prototypes changed, I am wondering when we are selecting the proper hyperparameters for the DistR, how we are able to select the hyperparameter for them. It would be more likely to be applicable if those information can be added within the paper.

---

> ### Author Response · Authors · 2025-05-07
> **Reply to Reviewer YTGR (1)**
>
> Thank you for your valuable comments, which help us improve the manuscript. All the changes made to the manuscript are highlighted in red.
>
> > Lack of scalability evidence for large datasets.
>
> Thank you for your insightful comment. First, **we would like to stress that large-scale applicability is not a primary claim of our paper and does not constitute its main objective**.
>
> We agree that there is indeed room for improvement in this regard (just as the original t-SNE, initially $O(N^2)$, underwent significant advancements to enhance scalability). This remains an interesting direction for future work.
>
> To address your concern more specifically, we refer you to the runtime experiments presented in Appendix F.8. These experiments demonstrate that our DistR models are competitive with all benchmarked methods in terms of computation time. Notably, they can be executed within a few seconds on a GPU for medium-sized datasets ($N=10000$ for MNIST), without even leveraging low-rank kernel structures or approximations commonly used in the DR literature.
>
> We also point out that computing the similarity matrix $C_X$ itself entails quadratic computational complexity in terms of the sample size $N$. This step is fundamental to most DR methods, meaning our approach does not introduce significant computational overhead compared to existing techniques.
> To give you some concrete order of magnitudes, storing $C_X(X)=XX^T$ with $N=10k$ requires 400 Mb and for $N=100k$ it would require 40 Gb without taking account the low-rank structures $C_X(X)$ but simply storing $X$ as done in our LR implementation requires only $20 Mb$ (with $d=50$ after the first PCA step used in DR papers). In the worse case where such low-rank properties are not satisfied, the recent package PyKeops (https://www.kernel-operations.io/keops/index.html) provides an extremely memory-efficient way to handle GPU memory allowing to potentially use DistR over millions of samples in future works.
>
> > The interpretability and explainability of the prototypes
>
> Thank you for this detailed analysis. Indeed, we agree that the homogeneity score tends to increase monotonically with the number of clusters. For this score to not be trivial, we enforced the number of prototypes $n$ to be significantly smaller than the number of samples $N$ (see Appendix F.1 for dataset statistics). Then we validated the number of prototypes such that $n \in \{ c + 20i \}_{i=0}^{10}$ where $c$ is the number of classes (cf L.410 in the paper), hence $c \leq n << N$ in all cases and does not appear to us as an overclustering in the context of joint DR & clustering. Our comprehensive results reported in Appendix F.4 clearly show that the evolution across $n$ of the homogeneity is not straight-forward and that DistR outperforms its competitors in most cases, even for methods admitting closed-form solutions like $C \rightarrow DR$ and $DR \rightarrow C$ with PCA kernels which cannot lead to "optimization artifacts".
>
> Moreover, as stressed in the paper, we consider that the homogeneity score has to be considered jointly with other globally discriminant metrics (silhouette & kmeans NMI) to guarantee the overall relevance of the prototypes. In particular, your concern about prototypes "collapsing" (i.e., whether the clusters are truly distinct) can be captured by our the silhouette score weighted by the transport plan. The method for computing this score is detailed in Appendix F.1. For each prototype, its silhouette coefficient is calculated as the difference between the (weighted) average distance to points within the same group and the (weighted) average distance to points in neighbouring groups. This score operates on a "the higher, the better" basis: if the difference is negative, the point is on average closer to a neighbouring group than to its own, indicating potential cluster collapse. Conversely, if the difference is positive, the point is closer on average to its own group than to a neighbouring group. As shown in many of our experimental results (Figure 3, Figure 5, and Appendix F.4), the silhouette scores for DistR are consistently higher compared to other methods, even as the number of clusters increases. We believe this demonstrates that cluster collapse does not occur in our approach. If you find this relevant, we can move this explanation into the main text to make it clearer.
>
> Finally, the choice of using medoids for image reconstruction was mainly to enhance clarity given the scale of the images, using centroids instead lead indeed to more blurred images but the medoids remain clearly visible, hence prototypes are not optimization artifacts.

---

> > ### Author Response · Authors · 2025-05-07
> > **Reply to Reviewer YTGR (2)**
> >
> > > The stopping criteria/hyperparameter selection for the DistR.
> > Based on the results shown within the Figure 5 and also for the Appendix F.4, the results for the DistR varies a lot when the number of prototypes changed, I am wondering when we are selecting the proper hyperparameters for the DistR, how we are able to select the hyperparameter for them. It would be more likely to be applicable if those information can be added within the paper.
> >
> > Thank you for the question. While hyperparameter selection is indeed a pertinent task, we believe that implementing a mechanism for automatic hyperparameter tuning falls slightly outside the scope of our current paper, acknowledging that these problems remain rather opened for most famous DR kernels considered in the paper. **Indeed the issue of hyperparameter selection is analogous in our context to that of any other DR and clustering methods** (and one can rely on BIC or ELBO rules on the GW loss for example).
> >
> > The primarly objective of the paper was more to demonstrate that DR methods can be formulated as a GW problem (Section 3) and to generalize them to perform joint clustering/DR while showing (in the experiments) that this generalization makes sense in practice. This is supported by our choice of experiments, which ensure a **fair comparison across all methods**. Furthermore, to strengthen our hyperparameter analysis, we conducted extensive sensitivity studies, revealing that DistR consistently leads to highly relevant embeddings compared to two-step approaches like DR-C and C-DR remains valid as the number of prototypes grows, as shown in Appendices F.6 and F.7.

---

### Review · Reviewer_3D1w · 2025-03-09

**Summary Of Contributions:**

The paper describes a unified framework for "data summarisation", within which both the problems of clustering and dimension reduction lie,
in terms of optimal transport and the Goromov-Wasserstein distance.
Some theoretical results are given which characterise existing models within this framework. In addition the authors show that a relaxation
of the problem, which does not tie the graph of the reduction to an explicit form, has a solution which induces a hard clustering.
An alternating algorithm for jointly reducing the dimension and granularity (the latter potentially for clustering) of the data is described.

Results from experiments comparing their approach with the alternatives of performing the dimension reduction and clustering tasks separately, in both orders (dimension reduction then clustering, and clustering then dimension reduction) show some promise of the proposed approach.

**Audience:**

Yes

**Broader Impact Concerns:**

None are apparent

**Claims And Evidence:**

No

**Requested Changes:**

-- within Theorem 3.2 the way $\alpha_Z, \beta_Z$ are introduced is strange. Perhaps replace the word "where" with "for some"
-- within the paragraph "DR as GW Minimization." the final sentence is not making a statement, suggest to reword as "... $\forall x \in \R^N$ s.t. $x^\top 1 = 0$ we have $x^\top C x \geq 0$..."
-- Having both notations C_X and C_{\bf X} is not advisable, they are too similar.
-- Can you elaborate on what is meant by "reasonable assumptions"?

-- Figure 1 is not that easy to understand without a bit of guidance from the authors in terms of what is being shown. Even when it is discussed in the text, it was not completely clear to me. Can you give more detailed description?

-- Some more discussion around Theorem 4.1 would be nice, especially whether the graph $\bar C$ is meaningful in its own right when $T$ induces a hard clustering.

-- I recognise the fact that using spectral clustering is the most "obvious" approach to use in DR-C and C-DR, however as mentioned above this will typically lead to meaningless results when producing a large number of "prototypes". Although arguably less natural, I strongly suggest using a more commonly used "prototyping" method. If the graph can be interpreted as/converted into a distance matrix then Partitioning Around Medoids (PAM) is the most obvious, otherwise, despite the fact that it doesn't directly connect with C_Z(Z) in DR-C, I would suggest kmeans.

**Strengths And Weaknesses:**

Strengths:
- For the most part the paper is well written and relatively easy to read.
- Although the connections between optimal transport and the problems of clustering and dimension reduction are not new, the paper does a good job of concisely characterising the relationships.
- Theorem 3.2 (although strangely worded; see below) has some nice generality, and if novel is a nice contribution in itself
- Theorem 4.1 (although needing some more discussion; see below) is also a nice contribution.

Weaknesses:
- There are some grammatical errors/quirks, some of which I indicate below in the following section.
- Unless I am missing something Lemma 3.1 is completely trivial.
- The experiments are not well designed, since "overclustering" using spectral clustering is nonsensical. In particular, the eigenvectors of the Laplacian(s) which are not associated with actual clusters will very easily lead to points arbitrarily far from one another being grouped together.

---

> ### Author Response · Authors · 2025-05-07
> **Reply to Reviewer 3D1w**
>
> Thank you for your valuable comments, which help us improve the manuscript. All the changes made to the manuscript are highlighted in red.
>
> > within Theorem 3.2 the way $\alpha_Z$ and $\beta_Z$ are introduced is strange. Perhaps replace the word "where" with "for some"
>
> Thank you we changed the formulation in the paper.
>
> > within the paragraph "DR as GW Minimization." the final sentence is not making a statement, suggest to reword as "... $\forall x \in \mathbb{R}^N$, s.t $x^\top 1 =0$ we have $x^\top C x \geq 0$ ..."
>
> Thank you we also changed this formulation.
>
> > Having both notations $C_X$ and $C_{\bf X}$ is not advisable, they are too similar.
>
> Yes thank you, it was typo we corrected it and kept everywhere $C_X$ as for the "function that generates a similarity matrix in the input space from a dataset"
>
> >   Can you elaborate on what is meant by "reasonable assumptions"?
>
> Thank you for pointing out this potential ambiguity. To clarify, by "reasonable assumption," we meant that the $C_Z$ function is permutation equivariant. This means that if we permute the samples in the input of $C_Z$ (i.e., the rows of $Z$), the resulting $C_Z(Z)$ matrix will be permuted accordingly. This property holds for any similarity matrix commonly encountered in practice.
> We changed the formulation in the paper.
>
> > Figure 1 is not that easy to understand without a bit of guidance from the authors in terms of what is being shown. Even when it is discussed in the text, it was not completely clear to me. Can you give more detailed description?
>
> We have changed the caption according to your suggestion.
>
> > Some more discussion around Theorem 4.1 would be nice, especially whether the graph $\bar{\mathbf{C}}$ is meaningful in its own right when $\mathbf{T}$ induces a hard clustering.
>
> We added some comments on this point, specifically noting that when $T$ represents a hard clustering, the $\bar{\mathbf{C}}$ matrix corresponds to the weights of a graph that results from a standard edge contraction according to $T$. Moreover, we detailed the implications of the equivalence between srGWB and weighted kernel kmeans for PSD matrices in terms of spectrum preservation.
>
> > I recognise the fact that using spectral clustering is the most "obvious" approach to use in DR-C and C-DR, however as mentioned above this will typically lead to meaningless results when producing a large number of "prototypes". Although arguably less natural, I strongly suggest using a more commonly used "prototyping" method. If the graph can be interpreted as/converted into a distance matrix then Partitioning Around Medoids (PAM) is the most obvious, otherwise, despite the fact that it doesn't directly connect with C_Z(Z) in DR-C, I would suggest kmeans.
>
> Thank you for these interesting comments. We recall that we validated the number of prototypes such that $n \in \{ c + 20i \}_{i=0}^{10}$ where $c$ is the number of classes (cf L.410 in the paper), hence $c \leq n << N$ in all cases and does not appear to us as an overclustering in the context of joint DR & clustering. We refer to our reply to Reviewer YTGR on homogeneity scores which support this idea. Nonetheless, faster initializations are of great interest for DistR and constitute an interesting direction for future works. **Hence we are currently conducting a comprehensive benchmarks of methods with PCA and t-SNE like kernels, while using kmeans clustering as initialization**. These experiments across all datasets will be finished by tomorrow and results will be reported in the manuscript. For now, intermediate results tend to show that, for all methods, using spectral clustering as initialization tends to lead to better prototypes than while using kmeans clustering. Moreover, rankings across methods seem to be preserved with DistR outperforming $C \rightarrow DR$ and $DR \rightarrow C$.  We will detail our answer as soon as possible.

---

> > ### Author Response · Authors · 2025-05-13
> > **Reply to Reviewer 3D1w (2)**
> >
> > We completed the benchmark mentioned in the last item above and reported the results in Appendix F.7 of the revised version.
> > We observe that on average across datasets, we can observe that DistR (SC), the default for the method, achieves a better trade-off w.r.t the silhouette and Kmeans NMI scores than DistR (Kmeans), while maintaining similar homogeneity scores. A similar behaviour is observed for C$\to$DR using PCA like kernels, whereas overall higher performances are observed for C$\to$DR (SC) than C$\to$DR (Kmeans). Therefore SC appears as a better clustering strategies to maximize performances for both DistR and C$\to$DR, where DistR (SC) remains the most competitive method. Finally, DR$\to$C (Kmeans) consistently leads to higher homogeneity scores with comparable silhouette scores than DR$\to$C (SC), but their respective trade-off between homogeneity and Kmeans NMI are more erratic.

---

### Review · Reviewer_QBZR · 2025-04-22

**Summary Of Contributions:**

Gromov-Wasserstein is able to define distance or compare data distributions over different (dimensional) domains, making it a good candidate for a fundamental technique in dimension reduction and clustering. The authors use GW to bridge many examples of clustering and dimension reduction, allowing for a joint dimension reduction and clustering technique. The approach (called distributional reduction) is broadly applicable and supported by theoretical results which recover special cases and other connections to existing methods. Demonstrations are provided on multiple popular datasets.

**Audience:**

Yes

**Broader Impact Concerns:**

This work presents general methodology and unifying theory, without a specific application in mind. The usual issues surrounding use of technology apply. As such, a broader impact statement is not necessary.

**Claims And Evidence:**

Yes

**Requested Changes:**

Please address my comments above. Namely,
- Discuss the ability of various methods to perform embeddings on test data
- Please clarify computational aspects of your approach. In what sense is srDistR simpler?
- Clarify text surrounding Theorem 4.1
- Could you comment on the bottom-right part of Figure 4?

Please address the minor points too (no need to respond here).

**Strengths And Weaknesses:**

Overall, the paper is well-written, with novel ideas, and what appears to be sound statements.

**comments (mixed strengths/weaknesses):**
- I like section 2.1. One thing I am missing is that in PCA, kernel PCA, etc., it is possible to reuse the mapping from high dimension to low dimension on new test points. However, in tSNE this is not possible AFAIK. What is the reason for this, and is this reason visible from the perspective of equations (DR) and (NE)?
- A nice direction for future work is to cast the widely popular t-SNE in terms of GW, using a result similar to Theorem 3.2 (ii). The authors note that t-SNE doesn't currently fit, because the pairwise similarity matrix in original dataspace is not CPD. The authors did a good job in pointing out the current limitation of their work.
- Dimension reduction (which optimises equally weighted dirac masses) can be generalised to distributional reduction, which additionally optimises the weights.
- DistR is a nonconvex problem, and the authors attempt to find local minima using a block coordinate descent, where one of the coordinate block steps is simply a semi relaxed DistR, which is "computationally simpler" (by the way, could you please indicate more precisely how it is computationally simpler, and what can be achieved?).
- The text before Theorem 4.1 seems to speak in very general terms, however the theorem requires $L=L_2$ (and more mildly, CPD/CND similarity in input space). Shouldn't you add some sentences to $L=L_2$ in the preceeding text?
- There is no direct link between DistR and and clustering, the authors do a good job in providing a connection between srGWB and hard clustering, and therefore provide an intuitive motivation for using GWB for clustering. The limitation but applicability of this result is appropriately discussed.
- Figure 3 takes a little while to understand, but is overall very clearly presented, with a good caption.
- I find it curious that the bottom-right subfigure in figure 4 has a constant homogeneity of 100 using distR. Could you explain this?



Minor:
- Figure 1 caption "two toys examples" should be "two toy examples"
- "the findings of (Chen et al., 2023), which" should be "the findings of Chen et al. (2023), which "

---

> ### Author Response · Authors · 2025-05-07
> **Reply to Reviewer QBZR (1)**
>
> Thank you for your valuable comments, which help us improve the manuscript. All the changes made to the manuscript are highlighted in red.
>
> > One thing I am missing is that in PCA, kernel PCA, etc., it is possible to reuse the mapping from high dimension to low dimension on new test points. However, in tSNE this is not possible AFAIK. What is the reason for this, and is this reason visible from the perspective of equations (DR) and (NE)?
>
> Thank you for this insightful comment, which raises the important question of the **out-of-sample properties of dimensionality reduction methods**.
>
> Kernel PCA benefits from a closed-form solution for $\mathbf{Z} \in \mathbb{R}^{N \times d}$ which depends on the eigenvectors of the similarity matrix $C_X(\mathbf{X})$ with $\mathbf{X} \in \mathbb{R}^p$, defining an orthogonal basis in $\mathbb{R}^{d}$, hence which can be reused to embed new points of $\mathbb{R}^p$ by orthogonal projections (inductive method). SNE variants do not benefit from such well-behaved closed-form solutions explaining why these are originally transductive methods **that can not embed new points**.
> The common workaround is to use a parametric family of functions $(f_θ)$, where $θ$ is learned to minimize the same dimension reduction objective, with embeddings represented as $Z = f_θ(X)$, often parameterized by neural networks, e.g. [1,2,3].
> However, these strategies are primarily driven by practical considerations rather than theoretical grounding: it is not known to what extent these strategies generalize well to unseen data.
>
> We believe that the connections between dimensionality reduction (DR) and Gromov-Wasserstein (GW) transport represent an important step toward finding functions that naturally extend methods like t-SNE to new data.
>
> Indeed, as shown in this work and in [4] (in the context of multidimensional scaling), when **the optimal transport plan is supported by a Monge map T, this T directly provides the optimal embeddings**.
>
> Therefore, the challenge reduces to understanding when GW admits a Monge map to discover these "generalizing functions", a difficult yet already explored problem, as discussed in [5].
>
> If relevant, we would be happy to incorporate this discussion into the conclusion of the paper.
>
> [1] L. Van Der Maaten, “Learning a parametric embedding by preserving local structure,” in Artificial intelligence and statistics, 2009.
>
> [2] A. Gisbrecht, A. Schulz, and B. Hammer, “Parametric nonlinear dimensionality reduction using kernel t-sne,” Neurocomputing, 2015.
>
> [3] E. Roman-Rangel and S. Marchand-Maillet, “Inductive t-sne via deep learning to visualize multi-label images,” Engineering Applications of Artificial Intelligence, vol. 81, 2019.
>
> [4] R. A. Clark, T. Needham, and T. Weighill, “Generalized dimension reduction using semi-relaxed gromov-wasserstein distance,” 2024.
>
> [5] T. Dumont, T. Lacombe, and F.-X. Vialard, “On the existence of monge maps for the gromov–wasserstein problem,” Foundations of Computational Mathematics, 2024.
>
> > DistR is a nonconvex problem, and the authors attempt to find local minima using a block coordinate descent, where one of the coordinate block steps is simply a semi relaxed DistR, which is "computationally simpler" (by the way, could you please indicate more precisely how it is computationally simpler, and what can be achieved?).
>
> Thank you for this comment, which relates to paper [6], where semi-relaxed Gromov-Wasserstein (srGW) was introduced. The problem becomes "simpler" due to the semi-relaxed constraint, as solving it only requires addressing **independent line-wise problems** on the transport matrix $T$. This property allows for more efficient implementation compared to the GW problem, as highlighted in [Section 3.2, 6].
> For instance, the direction-finding step of the exact Frank-Wolfe solver for srGW can be solved with linear complexity w.r.t $N$ (see Appendix E.2), whereas for GW a linear OT problem has to be solved e.g via common network flow solvers admitting a cubic complexity. Moreover the srGW solver can be easily implemented on GPU (going 100 times faster than on CPU in our experiments), whereas the GW one still doesn't benefit from efficient GPU implementations.
>
> While the theoretical complexity remains the same for both problems, the practical runtime is significantly improved, srGW is approximately an order of magnitude faster than GW in many scenarios while running on CPU and even more on GPU.
>
>
> [6] Semi-relaxed Gromov-Wasserstein divergence with applications on graphs
> Cédric Vincent-Cuaz, Rémi Flamary, Marco Corneli, Titouan Vayer, Nicolas Courty, ICLR, 2022.
>
> > The text before Theorem 4.1 seems to speak in very general terms, however the theorem requires $L=L_2$ (and more mildly, CPD/CND similarity in input space). Shouldn't you add some sentences to $L=L_2$ in the preceeding text?
>
> Thank you for the remark, we corrected it in the paper.

---

> > ### Author Response · Authors · 2025-05-07
> > **Reply to Reviewer QBZR (2)**
> >
> > > I find it curious that the bottom-right subfigure in figure 4 has a constant homogeneity of 100 using distR. Could you explain this?
> >
> > Thank you for this observation. In most scenarios (see complete scores in Appendix F.4), we can observe that DistR outperforms its competitors in terms of homogeneity with high performances across most kernels and datasets, which suggest that its joint optimization scheme benefits to clustering input samples via the OT plan. However, **performance variations across kernels for a given dataset are quite hard to interpret for now**. For instance for the SNAREseq1 dataset you mentioned, the homogeneity across tested number of propototypes is always 100% for t-SNE like kernels using both entropic and euclidean projections, whereas it varies between 86-100% with UMAP and 80-100% with PCA.
> >
> > We emphasize that the choice and existence of an optimal pair of kernels (DR) is out of the scope of our DistR paper. However, our various sensitivity analyses reveal that the tested spectral method consistently underperforms compared to its SNE-based competitors, and the latter are also closer from one another. To summarize this, we report here for each method the performances over our 3 metrics averaged across datasets, selecting models similarly than in Figure 3 taking into account all metrics simultaneously. We can observe that tSNE (KL proj.) leads to the best averaged kmeans NMI and homogeneity scores, while ranking second in terms of silhouette score. Whereas UMAP leads to the best averaged silhouette score, while ranking second on the other metrics. These results seem consistent with those reported in the extensive quantitative analysis of the paper introducing the tSNE method with KL projection. **Overall, this analysis suggests that improvements w.r.t the DR model or its optimization will lead to improvements for our DistR framework**.
> >
> > |    averaged metrics   | tSNE (KL proj.) | tSNE (L2 proj.) |  UMAP  |
> > |:------------:|:---------------:|:---------------:|:------:|
> > | silhouette |       41.2      |       38.9      | **44** |
> > | kmeans NMI |       **71.2**      |       66.3      | 70.3 |
> > | homogeneity |        **97.0**       |       95.4      | 95.6 |

---

### Decision · Action_Editor_DNws · 2025-06-16

**Recommendation:** Accept as is

**Audience:**

Yes

**Audience Explanation:**

DR and clustering two classical and important problems in ML. Both are of strong interest to the TMLR audience, and this paper offers a unifying approach that addresses both tasks.

**Claims And Evidence:**

Yes

**Claims Explanation:**

The claims of the paper are well supported theoretically and through extensive experimentation. The authors have effectively addressed the reviewers' concerns. While a few minor issues remain, they can be readily resolved in the final revision.